# Improving the Worst-Case Bidirectional Communication Complexity for Nonconvex Distributed Optimization under Function Similarity

**Kaja Gruntkowska**
KAUST*

**Alexander Tyurin**
KAUST,* AIRI,† Skoltech‡

**Peter Richtárik**
KAUST*

## Abstract

Effective communication between the server and workers plays a key role in distributed optimization. In this paper, we focus on optimizing communication, uncovering inefficiencies in prevalent downlink compression approaches. Considering first the pure setup where the uplink communication costs are negligible, we introduce MARINA-P, a novel method for downlink compression, employing a collection of correlated compressors. Theoretical analysis demonstrates that MARINA-P with permutation compressors can achieve a server-to-worker communication complexity improving with the number of workers, thus being provably superior to existing algorithms. We further show that MARINA-P can serve as a starting point for extensions such as methods supporting bidirectional compression: we introduce M3, a method combining MARINA-P with uplink compression and a momentum step, achieving bidirectional compression with provable improvements in total communication complexity as the number of workers increases. Theoretical findings align closely with empirical experiments, underscoring the efficiency of the proposed algorithms.

## 1   Introduction

In federated learning (McMahan et al., 2017; Konečný et al., 2016) and large-scale machine learning (Ramesh et al., 2021; OpenAI, 2023), a typical environment consists of multiple devices working together to train a model. Facilitating this collaborative process requires the transmission of substantial information (e.g., gradients, current model) between these devices. In the centralized framework, communication takes place via a server. As a result, practical challenges arise due to the large size of machine learning models and network speed limitations, potentially creating a communication bottleneck (Kairouz et al., 2021; Wang et al., 2023a). One possible strategy to reduce this communication burden is to use *lossy compression* (Seide et al., 2014; Alistarh et al., 2017). Our paper focuses on this research direction.

We consider the following nonconvex distributed optimization task:

$$\min_{x \in \mathbb{R}^d} \left\{ f(x) := \frac{1}{n} \sum_{i=1}^{n} f_i(x) \right\}, \tag{1}$$

where $x \in \mathbb{R}^d$ is the vector of parameters of the model, $n$ is the number of workers and $f_i : \mathbb{R}^d \to \mathbb{R}$, $i \in [n] := \{1, \dots, n\}$ are smooth nonconvex functions. We investigate the scenario where the functions $f_i$ are stored on $n$ distinct workers, each directly connected to the server via some

---

*King Abdullah University of Science and Technology, Thuwal, Saudi Arabia

†AIRI, Moscow, Russia

‡Skolkovo Institute of Science and Technology, Moscow, Russia

38th Conference on Neural Information Processing Systems (NeurIPS 2024).

communication port (Kairouz et al., 2021). At present, we operate under the following generic assumptions:

**Assumption 1.1.** The function $f$ is $L$–smooth, i.e., $\|\nabla f(x) - \nabla f(y)\| \leq L \|x - y\| \; \forall x, y \in \mathbb{R}^d$.

**Assumption 1.2.** There exists $f^* \in \mathbb{R}$ such that $f(x) \geq f^* \; \forall x \in \mathbb{R}^d$.

In the nonconvex world, our goal is to find a (possibly) random point $\bar{x}$ such that $\mathbb{E}[\|\nabla f(\bar{x})\|^2] \leq \varepsilon$. We refer to such a point an $\varepsilon$–stationary point.

## 1.1 Related Work

Before we discuss more advanced optimization methods, let us consider the simplest baseline: the gradient descent (GD) (Lan, 2020), which iteratively performs updates $x^{t+1} = x^t - \gamma \nabla f(x^t) = x^t - \gamma/n \sum_{i=1}^n \nabla f_i(x^t)$. In the distributed setting, the method can be implemented as follows: each worker calculates $\nabla f_i(x^t)$ and sends it to the server where the gradients are aggregated, after which the server takes the step and broadcasts $x^{t+1}$ back to the workers. With step size $\gamma = 1/L$, GD finds an $\varepsilon$–stationary point after $\mathcal{O}\left(\delta^0 L / \varepsilon\right)$ steps, where $\delta^0 := f(x^0) - f^*$ for a starting point $x^0$. Since at each step the workers and the server send $\Theta(d)$ coordinates/bits, the worker-to-server (w2s, uplink) and server-to-worker (s2w, downlink) communication costs are

$$\mathcal{O}\left(\frac{d\delta^0 L}{\varepsilon}\right). \tag{2}$$

**Definition 1.3.** The *worker-to-server (w2s)* and *server-to-worker (s2w)* communication complexities of a method are the expected number of coordinates/floats that a worker sends to the server and that the server sends to a worker, respectively, to find an $\varepsilon$–solution. The *total communication complexity* is the sum of these complexities.

**Unbiased compressors.** In this work, to perform lossy compression, we employ mappings from the following family:

**Definition 1.4.** A stochastic mapping $\mathcal{C} : \mathbb{R}^d \to \mathbb{R}^d$ is an *unbiased compressor* if there exists $\omega \geq 0$ such that

$$\mathbb{E}\left[\mathcal{C}(x)\right] = x, \; \mathbb{E}\left[\|\mathcal{C}(x) - x\|^2\right] \leq \omega \|x\|^2 \; \forall x \in \mathbb{R}^d. \tag{3}$$

We denote the family of such mappings by $\mathbb{U}(\omega)$. A canonical example is the $\text{Rand}K \in \mathbb{U}(d/K - 1)$ sparsifier, which preserves $K$ random coordinates of a vector scaled by $d/K$ (Beznosikov et al., 2020). More examples can be found in Wangni et al. (2018); Beznosikov et al. (2020); Szlendak et al. (2021); Horváth et al. (2022). A larger family of compressors, called *biased compressors*, also exists (see Section B). In this paper, we implicitly assume that compressors are mutually independent *across iterations* of algorithms.

**Worker-to-server compression scales with $n$.** Many previous works ignore the s2w communication costs and focus solely on w2s compression, assuming that *broadcasting is free*. For nonconvex objective functions, the current state-of-the-art w2s communication complexities are achieved by the MARINA and DASHA methods (Gorbunov et al., 2021; Szlendak et al., 2021; Tyurin and Richtárik, 2023b). Here, two additional assumptions are needed:

**Assumption 1.5.** The function $f_i$ is $L_i$–smooth. We define $\widehat{L}^2 := \frac{1}{n} \sum_{i=1}^n L_i^2$ and $L_{\max} := \max_{i \in [n]} L_i$.

**Assumption 1.6.** For all $\mathcal{C} \in \mathbb{U}(\omega)$, all calls of $\mathcal{C}$ are mutually independent.[4]

Under Assumptions 1.1, 1.2, 1.5, 1.6, and considering the $\text{Rand}K$ compressor with $K \leq d/\sqrt{n}$ as an example, the w2s communication complexity of both methods is

$$\underbrace{K}_{\text{\# of sent coord.}} \times \underbrace{\mathcal{O}\left(\frac{\delta^0}{\varepsilon}(L + \frac{\omega}{\sqrt{n}}\widehat{L})\right)}_{\text{\# of iterations}} = \mathcal{O}\left(\frac{d\delta^0 \widehat{L}}{\sqrt{n}\varepsilon}\right), \tag{4}$$

---

[4]This assumptions means that if an algorithm calls a compressor $\mathcal{C}$ at some points $x_1, \ldots, x_m$, then $\mathcal{C}(x_1), \ldots, \mathcal{C}(x_m)$ are *i.i.d.*

where we use the facts that $L \leq \widehat{L}$ and $\omega = d/K - 1$ for RandK. The key observation is that when comparing (2) and (4), one sees that (4) can be $\sqrt{n}$ times smaller if $\widehat{L} \approx L$. Consequently, the communication complexity of MARINA/DASHA scales with the number of workers $n$, and can *provably* improve the *worker-to-server* communication complexity $\mathcal{O}\left(d\delta^0 L/\varepsilon\right)$ achieved by GD.

**Server-to-worker compression does not scale with $n$.** In certain applications, the significance of s2w communication cannot be ignored. In 4G LTE and 5G networks, w2s and s2w communication speeds can be almost the same (Huang et al., 2012) or differ by at most a factor of 10 (Narayanan et al., 2021). Although important, this issue is often overlooked and that is why it is the s2w communication that this work places a central emphasis on.

There exist many papers using communication compression techniques to reduce the s2w communication (Zheng et al., 2019; Liu et al., 2020; Philippenko and Dieuleveut, 2021; Fatkhullin et al., 2021; Gruntkowska et al., 2023; Tyurin and Richtárik, 2023a). However, to the best of our knowledge, under Assumptions 1.1, 1.2, 1.5, and 1.6, in the worst case, all previous theoretical s2w communication guarantees *are greater or equal* to (2). As an example, let us consider the result from Gruntkowska et al. (2023)[Theorem E.3]. If the server employs operators from $\mathbb{U}(\omega)$ and we ignore w2s compression, the method from Gruntkowska et al. (2023) converges in $\mathcal{O}\left((\omega+1)\delta^0 L/\varepsilon\right)$ iterations. Thus, with RandK, the s2w communication complexity is $\mathcal{O}\left(K \times (\omega+1)\delta^0 L/\varepsilon\right) = \mathcal{O}\left(d\delta^0 L/\varepsilon\right)$. Another method, called CORE, proposed by Yue et al. (2023), achieves s2w and w2s communication complexities equal to $\mathcal{O}\left(r_1(f)\delta^0 L/\varepsilon\right)$, where $r_1(f)$ is a uniform upper bound of the trace of the Hessian. When $r_1(f) \leq dL$, CORE can improve on GD. However, this complexity does not scale with $n$ and requires an additional assumption about the Hessian of $f$.

## 2 Contributions

In our work, we aim to investigate whether the *server-to-worker and total communication complexities* (2) of the vanilla GD method can be improved. We make the following contributions:

1. We start by proving the impossibility of devising a method where the server communicates with the workers using unbiased compressors $\mathbb{U}(\omega)$ (or biased compressors from Section B) and achieves an *iteration rate* faster than $\Omega\left((\omega+1)L\delta^0/\varepsilon\right)$ (Theorem 3.1) under Assumptions 1.1, 1.2 and 1.6. This result provides a lower bound for any method that applies such compressors to vectors sent from the server to the workers in every iteration. Moreover, we prove a more general iteration lower bound of $\Omega\left((\omega+1)L\delta^0/\varepsilon\right)$ for all methods where the server zeroes out a coordinate with probability $1/(\omega+1)$ (see Remark 3.2).

2. In view of this result, it is clear that an extra assumption is needed to break the lower bound $\Omega\left((\omega+1)L\delta^0/\varepsilon\right)$. In response, we introduce a novel assumption termed "Functional $(L_A, L_B)$ Inequality" (see Assumption 4.2). We prove that this assumption is relatively weak and holds, for instance, under the local smoothness of the functions $f_i$ (see Assumption 1.5).

3. We develop a new method for downlink compression, MARINA-P, and show that under our new assumption, along with Assumptions 1.1 1.2, and 1.6, it can achieve the iteration rate of

$$\mathcal{O}\left(\frac{\delta^0 L}{\varepsilon} + \frac{\delta^0 L_A(\omega+1)}{\varepsilon} + \frac{\delta^0 L_B(\omega+1)}{\sqrt{n}\varepsilon}\right)$$

(see Theorem D.1 with $p = 1/(\omega+1)$ + Lemma A.5). Notably, when $L_A$ is small and $n \gg 1$, this complexity is *provably* superior to $\Theta((\delta^0 L(\omega+1))/\varepsilon)$ and the complexities of the previous compressed methods. In this context, $L_A$ serves as a measure of the similarity between the functions $f_i$, and can be bounded by the "variance" of the Hessians of the functions $f_i$ (see Theorem 4.8). Thus, MARINA-P is *the first method whose iteration complexity can provably improve with the number of workers $n$.*

4. Moreover, MARINA-P can achieve the s2w communication complexity of

$$\mathcal{O}\left(\frac{d\delta^0 L}{n\varepsilon} + \frac{d\delta^0 L_A}{\varepsilon}\right).$$

When $L_A$ is small and $n \gg 1$, this communication complexity is provably superior to (2) and the communication complexities of the previous compressed methods.

5. Our theoretical improvements can be combined with techniques enhancing the *w2s communication complexities*. In particular, by combining MARINA-P with MARINA (Gorbunov et al., 2021) and adding the crucial momentum step, we develop a new method, M3, that guarantees a *total communication complexity* (s2w + w2s) of

$$\mathcal{O}\left(\frac{d\delta^0 L_{\max}}{n^{1/3}\varepsilon} + \frac{d\delta^0 L_A}{\varepsilon}\right).$$

When $n \gg 1$ and in the close-to-homogeneous regime, i.e., when $L_A$ is small, this complexity is better than (2) and the complexities of the previous bidirectionally compressed methods.

6. Our theoretical results are supported by numerical experiments (see Section F).

## 3  Lower Bound under Smoothness

Let us first investigate the possibility of improving the iteration complexity $\mathcal{O}\left(\frac{(\omega+1)\delta^0 L}{\varepsilon}\right)$ under Assumptions 1.1,1.2 and 1.6. In Section G, we consider a family of methods that include those proposed in Zheng et al. (2019); Liu et al. (2020); Philippenko and Dieuleveut (2021); Fatkhullin et al. (2021); Gruntkowska et al. (2023), where the server communicates with workers using unbiased/biased compressors, and establish that

**Theorem 3.1** (Slightly Less Formal Reformulation of Theorem G.5). *Under Assumptions 1.1, 1.2 and 1.6, all methods in which the server communicates with clients using different and independent unbiased compressors from $\mathbb{U}(\omega)$ and sends one compressed vector to each worker cannot converge before $\Omega\left(\frac{(\omega+1)L\delta^0}{\varepsilon}\right)$ iterations.*

*Remark* 3.2. The theorem remains applicable to biased compressors $\mathbb{B}(\alpha)$ (see Section B) with a lower bound of $\Theta\left(\frac{L\delta^0}{\alpha\varepsilon}\right)$. This is because if $\mathcal{C} \in \mathbb{U}(\omega)$, then $(\omega+1)^{-1}\mathcal{C} \in \mathbb{B}\left((\omega+1)^{-1}\right)$. We also establish a more general result (Theorem G.4): "all methods in which the server zeroes out a coordinate with probability $\leq p$ independently *across iterations* cannot converge before $\Omega\left(\frac{L\delta^0}{p\varepsilon}\right)$ iterations."

This lower bound is tight up to a constant factor. For instance, under exactly the same assumptions, the EF21-P mechanism from Gruntkowska et al. (2023) converges after $\Theta\left(\frac{(\omega+1)L\delta^0}{\varepsilon}\right)$ iterations. Unlike (4), this convergence rate does not scale with $n$, and Theorem 3.1 leaves no room for improvement. Consequently, breaking the lower bound requires an additional assumption about the structure of the problem. Before presenting our candidate assumption, we first introduce the ingredients needed to leverage it to the fullest extent: our novel downlink compression method and the type of compressors we shall employ.

## 4  The MARINA-P Method

Let us first recall the MARINA method (Gorbunov et al., 2021; Szlendak et al., 2021):

$$
\begin{aligned}
x^{t+1} &= x^t - \gamma g^t, \qquad c^t \sim \text{Bernoulli}(p) \\
g_i^{t+1} &= \begin{cases} \nabla f_i(x^{t+1}) & \text{if } c^t = 1, \\ g^t + \mathcal{C}_i^t(\nabla f_i(x^{t+1}) - \nabla f_i(x^t)) & \text{if } c^t = 0 \end{cases} \quad \text{for all } i \in [n], \\
g^{t+1} &= \frac{1}{n}\sum_{i=1}^n g_i^{t+1},
\end{aligned}
\tag{5}
$$

where $g^0 = \nabla f(x^0)$. Motivated by MARINA, we design its primal counterpart, MARINA-P (Algorithm 1), operating in the primal space of the model parameters, as outlined in (6).

At each iteration of MARINA-P, the workers calculate $\nabla f_i(w_i^t)$ and transmit it to the server. The server then averages the gradients and updates the global model $x^t$. Subsequently, with some (typically small) probability $p$, the master sends the non-compressed vector $x^{t+1}$ to all workers. Otherwise, the $i^{\text{th}}$ worker receives a compressed vector $\mathcal{C}_i^t(x^{t+1} - x^t)$. Each worker then uses the received message to compute $w_i^{t+1}$ locally. Importantly, $\mathcal{C}_1^t(x^{t+1} - x^t), \ldots, \mathcal{C}_n^t(x^{t+1} - x^t)$ *can differ*, and this distinction will form the basis of our forthcoming advancements.

The MARINA-P Method:

Initialize vectors $x_0, w_1^0, \ldots, w_n^0 \in \mathbb{R}^d$, step size $\gamma > 0$, probability $0 < p \leq 1$ and compressors $\mathcal{C}_1^t, \ldots, \mathcal{C}_n^t \in \mathbb{U}(\omega_P)$ for all $t \geq 0$. The method iterates

$$g^t = \frac{1}{n} \sum_{i=1}^{n} \nabla f_i(w_i^t),$$

$$x^{t+1} = x^t - \gamma g^t,$$

$$c^t \sim \text{Bernoulli}(p),$$

$$w_i^{t+1} = \begin{cases} x^{t+1} & \text{if } c^t = 1, \\ w_i^t + \mathcal{C}_i^t(x^{t+1} - x^t) & \text{if } c^t = 0 \end{cases}$$

for all $i \in [n]$.

We denote $w^t := {}^1\!/n \sum_{i=1}^{n} w_i^t$. See the implementation in Algorithm 1.

Comparing (5) and (6), MARINA-P and MARINA are dual methods: both learn control variables ($w_i^t$ and $g_i^t$), compress the differences ($x^{t+1} - x^t$ and $\nabla f_i(x^{t+1}) - \nabla f_i(x^t)$), and with some probability $p$ send non-compressed vectors ($x^{t+1}$ and $\nabla f_i(x^{t+1})$). However, unlike MARINA, which compresses vectors sent *from workers to server* and operates in the *dual* space of gradients, MARINA-P compresses messages sent *from server to workers* and operates in the *primal* space of arguments.

Let us take $\text{Rand}K \in \mathbb{U}(d/K - 1)$ as an example. If we set $p = (\omega + 1)^{-1} = K/d$ to balance heavy communications of $x^{t+1}$ and light communications of $\mathcal{C}_i^t$ in (6), MARINA-P averages sending $pd + (1 - p)K \leq 2K$ coordinates per iteration. Then, the lower bound from Theorem G.4 implies that at least $\Omega\left((\omega+1)\delta^0 L/\varepsilon\right)$ iterations of the algorithm are needed.

At first glance, it seems that MARINA-P does not offer any extra benefits compared to previous methods, and that is true – we could not expect to break the lower bound. However, as we shall soon see, under an extra assumption, MARINA-P can achieve a communication complexity that improves with $n$.

## 4.1 Three ways to compress

Existing algorithms performing s2w compression share a common characteristic: at each iteration, the server broadcasts *the same* message to all workers (Zheng et al., 2019; Liu et al., 2020; Fatkhullin et al., 2021; Gruntkowska et al., 2023; Tyurin and Richtárik, 2023a).[5] In contrast, in w2s compression methods, each worker sends to the server a *different* message, specific to the data stored on that particular device. An analogous approach can be taken in the s2w communication: intuitively, sending $n$ distinct messages would convey more information, potentially leading to theoretical improvements. This indeed proves to be the case. While the usual approach of the server broadcasting the same vector to all clients does not lead to an improvement over (2), allowing these vectors to differ enables a well-crafted method to achieve communication complexity that improves with $n$ (see Corollary D.4).

In Appendix A we provide a detailed discussion of the topic and compare the theoretical complexities of MARINA-P when the server employs three different compression techniques: a) uses *one compressor* and sends the same vector to all clients, b) uses a *collection of independent compressors*, or c) uses a *collection of correlated compressors*. We now turn to presenting the technique that gives the best theoretical s2w communication complexity out of these, namely the use of a set of *correlated compressors*.

---

[5]A notable exception form this rule is the MCM method (Philippenko and Dieuleveut, 2021) - see Appendix A.

## 4.2 Recap: permutation compressors Perm$K$

Szlendak et al. (2021) propose compressors that will play a key role in our new theory. For clarity of presentation, we shall assume that $d \geq n$ and $n|d$.[6]

**Definition 4.1** (Perm$K$ (for $d \geq n$ and $n|d$)). Assume that $d \geq n$ and $d = qn$, where $q \in \mathbb{N}_{>0}$. Let $\pi = (\pi_1, \ldots, \pi_d)$ be a random permutation of $\{1, \ldots, d\}$. For all $x \in \mathbb{R}^d$ and each $i \in \{1, 2, \ldots, n\}$, we define

$$\mathcal{C}_i(x) := n \times \sum_{j=q(i-1)+1}^{qi} x_{\pi_j} e_{\pi_j}.$$

Unpacking this definition: when the server compresses a vector using a Perm$K$ compressor, it randomly partitions its coordinates across the workers, so that each client receives a sparse vector containing a random subset of entries of the input vector. Like Rand$K$, Perm$K$ is also a sparsifier. However, unlike Rand$K$, it does not allow flexibility in choosing $K$, as it is fixed to $d/n$. Furthermore, it can be shown (Lemma A.6) that $\mathcal{C}_i \in \mathbb{U}(n-1)$ for all $i \in [n]$.

An appealing property of Perm$K$ is the fact that

$$\frac{1}{n} \sum_{i=1}^{n} \mathcal{C}_i(x) = x \tag{7}$$

for all $x \in \mathbb{R}^d$ deterministically. Here, it is important to note that by design, compressors $\mathcal{C}_i$ from Definition 4.1 are *correlated*, and do not satisfy Assumption 1.6. This correlation proves advantageous – Szlendak et al. (2021) show that MARINA with Perm$K$ compressors performs provably better than with i.i.d. Rand$K$ compressors.

## 4.3 Warmup: homogeneous quadratics

We are finally ready to present our first result showing that the s2w communication complexity can scale with the number of workers $n$. To explain the intuition behind our approach, let us consider the simplest (and somewhat impractical) choice of functions $f_i$ – the homogeneous quadratics:

$$f_i(x) = \tfrac{1}{2} x^\top \mathbf{A} x + b^\top x + c, \quad i \in [n], \tag{8}$$

where $\mathbf{A} \in \mathbb{R}^{d \times d}$ is a symmetric but not necessarily positive semidefinite matrix, $b \in \mathbb{R}^d$ and $c \in \mathbb{R}$. We now investigate the operation of MARINA-P with Perm$K$ compressors. With probability $p$, we have $w^{t+1} = x^{t+1}$. Otherwise $w^{t+1} = w^t + \frac{1}{n} \sum_{i=1}^n \mathcal{C}_i^t(x^{t+1} - x^t) \overset{(7)}{=} x^{t+1} + (w^t - x^t)$. Hence, if we initialize $w_i^0 = x^0$ for all $i \in [n]$, an inductive argument shows that $w^t = x^t$ deterministically for all $t \geq 0$. Then, substituting the gradients of $f_i$ to (6), one gets

$$g^t = \tfrac{1}{n} \sum_{i=1}^{n} (\mathbf{A} w_i^t + b) = \mathbf{A} w^t + b = \mathbf{A} x^t + b = \nabla f(x^t)$$

for all $t \geq 0$. Therefore, MARINA-P with Perm$K$ compressor in this setting is essentially a smart implementation of vanilla GD! Indeed, for $p \leq 1/n$, MARINA-P with Perm$K$ sends on average $\leq 2d/n$ coordinates to each worker, so the s2w communication complexity is

$$\frac{2d}{n} \times \underbrace{\mathcal{O}\left(\frac{\delta^0 L}{\varepsilon}\right)}_{\text{GD rate}} = \mathcal{O}\left(\frac{d\delta^0 L}{n\varepsilon}\right),$$

which is $n$ times smaller than in (2)!

## 4.4 Functional $(L_A, L_B)$ Inequality

From the discussion in Section 3, we know that to improve (2), an extra assumption about the structure of the problem is needed. Building on the example from Section 4.3, we introduce the *Functional $(L_A, L_B)$ Inequality*.

---

[6]The general definition of Perm$K$ for $d \bmod n \neq 0$ is presented in (Szlendak et al., 2021)[App. I].

Table 1: **The worst case *communication complexities* to find an $\varepsilon$–stationary point.** For simplicity, we compare the complexities with non-homogeneous quadratics: $f_i(x) = \frac{1}{2}x^\top \mathbf{A}_i x + b_i^\top x + c_i$, where $\mathbf{A}_i \in \mathbb{R}^{d \times d}$ is symmetric but not necessarily positive semidefinite, $b_i \in \mathbb{R}^d$ and $c_i \in \mathbb{R}$ for $i \in [n]$. We denote $\mathbf{A} = \frac{1}{n}\sum_{i=1}^n \mathbf{A}_i$.

| Server-to-Workers Communication Complexities (s2w) | | Total Communication Complexities (s2w + w2s) | |
|---|---|---|---|
| **Method** | **Complexity** | **Method** | **Complexity** |
| GD
and other compressed methods[a] | $\geq \frac{d\delta^0\|\mathbf{A}\|}{\varepsilon}$ | GD
and other compressed methods[a] | $\geq \frac{d\delta^0\|\mathbf{A}\|}{\varepsilon}$ |
| CORE
(Yue et al., 2023) | $\frac{\delta^0 \mathrm{tr}\mathbf{A}}{\varepsilon}$ | CORE
(Yue et al., 2023) | $\frac{\delta^0 \mathrm{tr}\mathbf{A}}{\varepsilon}$ |
| MARINA-P
with independent Rand$K$[b]
(Corollary D.4) | $\frac{d\delta^0 \frac{1}{n}\sum_{i=1}^n \|\mathbf{A}_i\|}{\sqrt{n}\varepsilon} + \frac{d\delta^0 \max_{i\in[n]}\|\mathbf{A}_i - \mathbf{A}\|}{\varepsilon}$ | M3
with Perm$K$ and Rand$K$[b]
(Theorem 5.1) | $\frac{d\delta^0 \max_{i\in[n]}\|\mathbf{A}_i\|}{n^{1/3}\varepsilon} + \frac{d\delta^0 \max_{i\in[n]}\|\mathbf{A}_i - \mathbf{A}\|}{\varepsilon}$ |
| MARINA-P with Perm$K$[b]
(Corollary 4.7) | $\frac{d\delta^0\|\mathbf{A}\|}{n\varepsilon} + \frac{d\delta^0 \max_{i\in[n]}\|\mathbf{A}_i - \mathbf{A}\|}{\varepsilon}$ | | |

The complexities of MARINA-P and M3 with Perm$K$ are better when $n > 1$ and in close-to-homogeneous regimes, i.e., when $\max_{i\in[n]}\|\mathbf{A}_i - \mathbf{A}\|$ is small.
[a] including EF21-P (Gruntkowska et al., 2023), dist-EF-SGD (Zheng et al., 2019), DORE (Liu et al., 2020), MCM (Philippenko and Dieuleveut, 2021), and EF21-BC (Fatkhullin et al., 2021).
[b] This table only showcases the results for Rand$K$ and Perm$K$. A more general result for all compressors is provided in Sections D and E.

**Assumption 4.2** (Functional $(L_A, L_B)$ Inequality). There exist constants $L_A, L_B \geq 0$ such that

$$\left\| \frac{1}{n}\sum_{i=1}^n \left(\nabla f_i(x + u_i) - \nabla f_i(x)\right) \right\|^2 \leq L_A^2 \left(\frac{1}{n}\sum_{i=1}^n \|u_i\|^2\right) + L_B^2 \left\| \frac{1}{n}\sum_{i=1}^n u_i \right\|^2 \qquad (9)$$

for all $x, u_1, \ldots, u_n \in \mathbb{R}^d$.

*Remark* 4.3. A similar assumption, termed "Heterogeneity-driven Lipschitz Condition on Averaged Gradients", is proposed in Wang et al. (2023b). Our assumption aligns with theirs when $L_B = 0$. However, our formulation proves to be more powerful. The possibility that $L_B > 0$ becomes instrumental in driving the enhancements we introduce.

Assumption 4.2 is defined for all functions together, and intuitively, it tries to capture the similarities between the functions $f_i$. For $n = 1$, inequality (9) reduces to

$$\|\nabla f(x) - \nabla f(y)\|^2 \leq \left(L_A^2 + L_B^2\right) \|x - y\|^2 \ \forall x, y \in \mathbb{R}^d,$$

equivalent to standard $L$-smoothness (Assumption 1.1) with $L^2 = L_A^2 + L_B^2$. The Functional $(L_A, L_B)$ Inequality is reasonably weak also for $n > 1$, as the next theorem shows.

**Theorem 4.4.** *For all $i \in [n]$, assume that the functions $f_i$ are $L_i$–smooth (Assumption 1.5). Then, Assumption 4.2 holds with $L_A = L_{\max}$ and $L_B = 0$.*

Therefore, Assumption 4.2 holds whenever the functions $f_i$ are smooth, which is a standard assumption in the literature. Now, returning to the example from Section 4.3,

**Theorem 4.5.** *For all $i \in [n]$, assume that the functions $f_i$ are homogeneous quadratics defined in (8). Then, Assumption 4.2 holds with $L_A = 0$ and $L_B = \|\mathbf{A}\|$.*

Under Assumption 1.5, no information about the similarity of the functions $f_i$ is available, yielding $L_B = 0$ and $L_A > 0$ in Theorem 4.4. However, once we have some information limiting heterogeneity, $L_A$ can decrease. Notably, $L_A = 0$ for homogeneous quadratics. As we shall see in Section 4.5, the values $L_A$ and $L_B$ significantly influence the s2w communication complexity of MARINA-P, with lower $L_A$ values leading to greatly improved performance.

### 4.5 The Convergence Theory of MARINA-P with Perm$K$

We are ready to present our main convergence result, focusing on the Perm$K$ compressor from Section 4.2. This choice simplifies the presentation, but our approach generalizes to a much larger class of compression operators. The full theoretical framework, covering all unbiased compressors, is detailed in Appendix D.

**Theorem 4.6.** *Let Assumptions 1.1, 1.2 and 4.2 be satisfied. Set $w_i^0 = x^0$ for all $i \in [n]$. Take PermK as $\mathcal{C}_i^t$ and $\gamma = \left( L + L_A \sqrt{\omega_P(1/p - 1)} \right)^{-1}$, where $\omega_P = n - 1$ (Lemma A.6). Then, MARINA-P finds an $\varepsilon$–stationary point after*

$$\mathcal{O}\left( \tfrac{\delta^0}{\varepsilon} \left( L + L_A \sqrt{\omega_P/p} \right) \right)$$

*iterations.*

**Corollary 4.7.** *Let $p = K/d \equiv 1/n$. Then, in the view of Theorem 4.6, the average s2w communication complexity of MARINA-P with PermK compressor is*

$$\mathcal{O}\left( \tfrac{d\delta^0 L}{n\varepsilon} + \tfrac{d\delta^0 L_A}{\varepsilon} \right). \tag{10}$$

The key observation is that (10) is independent of $L_B$, and only depends on $L_A$. This particular property is specific to *correlated compressors* with parameter $\theta = 0$ (defined in Appendix A), such as PermK. A similar result holds for *independent* RandK compressors (see Corollary D.4), but the convergence rate is worse and depends on $L_B$. Nevertheless, this dependence improves with $n$.

When $L_A = 0$, which is the case for homogeneous quadratics, the step size bound from Theorem 4.6 simplifies to $\gamma \leq 1/L$, the standard GD stepsize (recall that in this case our method reduces to GD). Most importantly, (10) scales with the number of workers $n$! Even when $L_A > 0$, for sufficiently big $n$, (10) can improve (2) to $\mathcal{O}\left( d\delta^0 L_A/\varepsilon \right)$.

Let us now investigate how the constants $L_A$ and $L_B$ change in the general case.

### 4.6 Estimating $L_A$ and $L_B$ in the General Case

It is clear from Corollary 4.7 that MARINA-P with PermK shines when $L_A$ is small. To gain further insights into what values $L_A$ may take, we now provide an analysis based on the Hessians of the functions $f_i$.

**Theorem 4.8.** *Assume that the functions $f_i$ are twice continuously differentiable, $L_i$–smooth (Assumption 1.5), and that there exist $D_i \geq 0$ such that*

$$\sup_{z_1,\ldots,z_n \in \mathbb{R}^d} \left\| \nabla^2 f_i(z_i) - \tfrac{1}{n} \sum_{j=1}^n \nabla^2 f_j(z_j) \right\| \leq D_i \tag{11}$$

*for all $i \in [n]$. Then, Assumption 4.2 holds with $L_A = \sqrt{2} \max_{i \in [n]} D_i \leq 2\sqrt{2} \max_{i \in [n]} L_i$ and $L_B = \sqrt{2} \left( \tfrac{1}{n} \sum_{i=1}^n L_i \right)$.*

Intuitively, (11) measures the similarity between the functions $f_i$. The above theorem yields a more refined result than Theorem 4.4: it is always true that $\max_{i \in [n]} D_i \leq 2 \max_{i \in [n]} L_i$, and, in fact, $\max_{i \in [n]} D_i$ can be much smaller, as the next result shows.

**Theorem 4.9.** *Assume that $f_i(x) = \tfrac{1}{2} x^\top \mathbf{A}_i x + b_i^\top x + c_i$, where $\mathbf{A}_i \in \mathbb{R}^{d \times d}$ is symmetric but not necessarily positive semidefinite, $b_i \in \mathbb{R}^d$ and $c_i \in \mathbb{R}$ for $i \in [n]$. Define $\mathbf{A} = \tfrac{1}{n} \sum_{i=1}^n \mathbf{A}_i$. Then, Assumption 4.2 holds with $L_A = \sqrt{2} \max_{i \in [n]} \|\mathbf{A}_i - \mathbf{A}\|$ and $L_B = \sqrt{2} \left( \tfrac{1}{n} \sum_{i=1}^n \|\mathbf{A}_i\| \right)$.*

Thus, $L_A$ is less than or equal to $\sqrt{2} \max_{i \in [n]} \|\mathbf{A}_i - \mathbf{A}\|$, which serves as a measure of similarity between the matrices. The smaller the values of $\|\mathbf{A}_i - \mathbf{A}\|$ (indicating greater similarity among the functions $f_i$), the smaller the $L_A$ value.

In the view of this theorem, the s2w communication complexity of MARINA-P with PermK on non-homogeneous quadratics is

$$\mathcal{O}\left( \tfrac{d\delta^0 \|\mathbf{A}\|}{n\varepsilon} + \tfrac{d\delta^0 \max_{i \in [n]} \|\mathbf{A}_i - \mathbf{A}\|}{\varepsilon} \right). \tag{12}$$

Since the corresponding complexity of GD is

$$\mathcal{O}\left( \tfrac{d\delta^0 \|\mathbf{A}\|}{\varepsilon} \right), \tag{13}$$

in the close-to-homogeneous regimes (i.e., when $\max_{i \in [n]} \|\mathbf{A}_i - \mathbf{A}\|$ is small), the complexity (12) can be *provably* much smaller than (13). The same reasoning applies to the general case when the





The M3 Method

(M3 = MARINA-P + Momentum + MARINA):



Initialize vectors $x_0, w_i^0, g_i^0, z_i^0 \in \mathbb{R}^d$ for all $i \in [n]$, step size $\gamma > 0$, probabilities $0 < p_P, p_D \leq 1$ and compressors $\mathcal{C}_1^t, \ldots, \mathcal{C}_n^t \in \mathbb{U}(\omega_P) \cap \mathbb{P}(\theta)^a$, $\mathcal{Q}_1^t, \ldots, \mathcal{Q}_n^t \in \mathbb{U}(\omega_D)$ for all $t \geq 0$. The method iterates

$$x^{t+1} = x^t - \gamma g^t,$$

$$w_i^{t+1} = \begin{cases} x^{t+1} & \text{with probability } p_P, \\ w_i^t + \mathcal{C}_i^t(x^{t+1} - x^t) & \text{with probability } 1 - p_P, \end{cases}$$

$$z_i^{t+1} = \beta w_i^{t+1} + (1 - \beta) z_i^t \qquad \text{(Momentum)} \tag{14}$$

$$g_i^{t+1} = \begin{cases} \nabla f_i(z_i^{t+1}) & \text{with probability } p_D, \\ g_i^t + \mathcal{Q}_i^t(\nabla f_i(z_i^{t+1}) - \nabla f_i(z_i^t)) & \text{with probability } 1 - p_D \end{cases}$$

for all $i \in [n]$,

where the probabilistic decisions are the same for all $i \in [n]$, i.e., one coin is tossed for all workers (as in (5) and (6)), and the coins for the first and second probabilistic decisions with $p_P$ and $p_D$ are independent. We denote $w^t := \frac{1}{n} \sum_{i=1}^n w_i^t$, $g^t := \frac{1}{n} \sum_{i=1}^n g_i^t$, $z^t := \frac{1}{n} \sum_{i=1}^n z_i^t$. See the implementation in Algorithm 2.

---

$^a$By $\mathbb{P}(\theta)$ we denote a family of *correlated compressors* (defined in Appendix A). It includes, among others, PermK compressors.



functions $f_i$ are not quadratics: MARINA-P improves with the number of workers $n$ in the regimes when $D_i$ are small (see Theorem 4.8).

Let us note that there is another method, CORE, by Yue et al. (2023), that can also provably outperform GD, achieving the s2w communication complexity of $\Omega\left(\delta^0 \mathrm{tr} \mathbf{A}/\varepsilon\right)$ on non-homogeneous quadratics. Neither their method nor ours universally provides the best possible communication guarantees. Our method excels in the close-to-homogeneous regimes: for example, if we take $\mathbf{A}_i = L_i \mathbf{I}$ for all $i \in [n]$, and define $L = \frac{1}{n} \sum_{i=1}^n L_i$, then the complexity of CORE is $\Omega\left(d\delta^0 L/\varepsilon\right)$, while ours is $\mathcal{O}\left(\frac{d\delta^0 L}{n\varepsilon} + \frac{d\delta^0 \max_{i \in [n]}|L_i - L|}{\varepsilon}\right)$. Hence, our guarantees are superior in regimes where $\max_{i \in [n]}|L_i - L| \ll L$. One interesting research direction is to develop a universally better method combining the benefits of both approaches.

## 5 M3: A New Bidirectional Method

In the previous sections, we introduce a new method that provably improves the *server-to-worker* communication, but ignores the *worker-to-server* communication overhead. Our aim now is to treat MARINA-P as a starting point for developing methods applicable to more practical scenarios, by combining it with techniques that compress in the opposite direction. Since the theoretical state-of-the-art w2s communication complexity is obtained by MARINA (see Section 1.1), our next research step was to combine the two and analyze "MARINA + MARINA-P", but this naive approach did not yield communication complexity guarantees surpassing (2) in any regime. It became apparent that some "buffer" step between these two techniques is needed, and this step turned out to be the momentum. Our new method, M3 (Algorithm 2), is described in (14).

M3 combines (5), (6), and the momentum step $z_i^{t+1} = \beta w_i^{t+1} + (1 - \beta) z_i^t$, which is the key to our improvements. A similar technique is used to reduce the variance in Fatkhullin et al. (2023). Let us explain how M3 works in practice. First, the server calculates $x^{t+1}$. Depending on the first probabilistic decision, it sends either $x^{t+1}$ or $\mathcal{C}_i^t(x^{t+1} - x^t)$ to the workers, who then calculate $w_i^{t+1}$ locally. Next, the workers compute $z_i^{t+1}$, and depending on the second probabilistic decision, they send either $\nabla f_i(z_i^{t+1})$ or $\mathcal{Q}_i^t(\nabla f_i(z_i^{t+1}) - \nabla f_i(z_i^t))$ back to the server. The server aggregates the received vectors and calculates $g^{t+1}$. As in MARINA, $p_P$ and $p_D$ are chosen in such a way that the

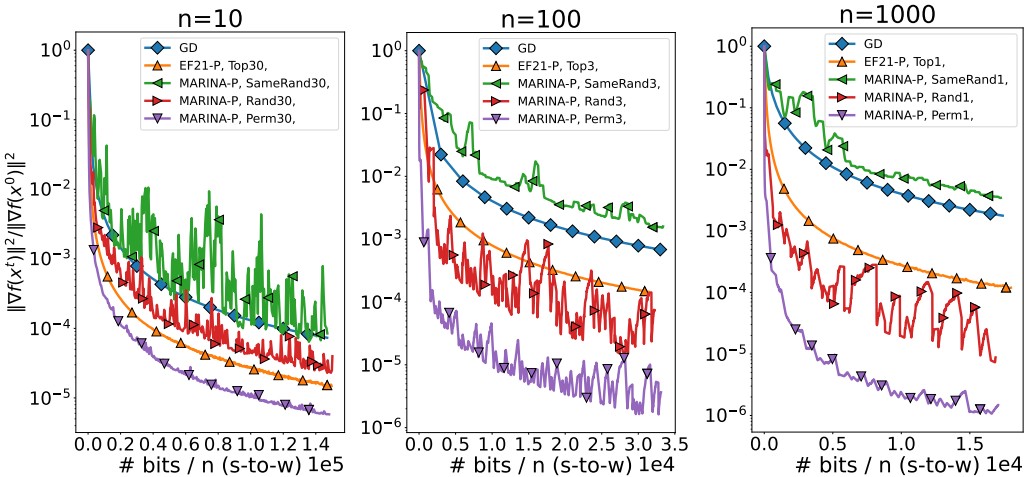

Figure 1: Experiments on the quadratic optimization problem from Section 6. We plot the norm of the gradient w.r.t. # of coordinates sent from the server to the workers.

non-compressed communication does not negatively affect the communication complexity. Therefore, the method predominantly transmits compressed information, with only a marginal probability of sending uncompressed vectors.

### 5.1 The Convergence Theory of M3

For simplicity, we consider $\text{Perm}K$ in the role of $\mathcal{C}_i^t$ and $\text{Rand}K$ in the role of $\mathcal{Q}_i^t$. The general theory for all unbiased compressors is presented in Section E.

**Theorem 5.1.** *Let Assumptions 1.1, 1.2, 1.5 and 4.2 be satisfied. Take $\gamma = \left(L + 34\left(nL_A + n^{2/3}L_B + n^{2/3}L_{\max}\right)\right)^{-1}$, $p_D = p_P = 1/n$, $\beta = n^{-2/3}$, $w_i^0 = z_i^0 = x^0$ and $g_i^0 = \nabla f_i(x^0)$ for all $i \in [n]$. Then $\text{MARINA-P}$ with $\mathcal{C}_i^t = \text{Perm}K$ and $\mathcal{Q}_i^t = \text{Rand}K$ with $K = d/n$ finds an $\varepsilon$–stationary point after $\mathcal{O}\left(\frac{\delta^0}{\varepsilon}\left(n^{2/3}L_{\max} + nL_A\right)\right)$ iterations. The total communication complexity is*

$$\mathcal{O}\left(\frac{d\delta^0 L_{\max}}{n^{1/3}\varepsilon} + \frac{d\delta^0 L_A}{\varepsilon}\right). \tag{15}$$

Once again, we observe improvement with the number of workers $n$, and the obtained complexity (15) can be provably smaller than (2). Indeed, in scenarios like federated learning, where the number of workers (e.g., mobile phones) is typically large (Kairouz et al., 2021; Chowdhery et al., 2023), the first term can be significantly smaller than $d\delta^0 L/\varepsilon$. The second term can also be small in close-to-homogeneous regimes (see Section 4).

## 6 Experimental Highlights

This section presents insights from the experiments, with further details and additional results in Appendix F. The experiment aims to empirically test the theoretical results from Section 4. We consider a quadratic optimization problem, where the functions $f_i$ are as defined in Theorem 4.9 and $\mathbf{A}_i \in \mathbb{R}^{300 \times 300}$. We compare GD, MARINA-P sending the same message compressed using a single $\text{Rand}K$ compressor to all workers ("SameRand$K$" from Appendix A), MARINA-P with independent $\text{Rand}K$ compressors, MARINA-P with $\text{Perm}K$ compressors, and EF21-P with Top$K$ compressor. We consider $n \in \{10, 100, 1000\}$ and fine-tune the step size for each algorithm. The results, presented in Figure 1, align closely with the theory, with MARINA-P using $\text{Perm}K$ compressors consistently performing best. Moreover, the convergence rate of MARINA-P with $\text{Perm}K$ and independent $\text{Rand}K$ compressors improves with $n$. Since this is not the case for EF21-P, even though it outperforms MARINA-P with independent $\text{Rand}K$ compressors for $n = 10$, it falls behind for $n \in \{100, 1000\}$.

## Acknowledgments and Disclosure of Funding

The research reported in this publication was supported by funding from King Abdullah University of Science and Technology (KAUST): i) KAUST Baseline Research Scheme, ii) Center of Excellence for Generative AI, under award number 5940, iii) SDAIA-KAUST Center of Excellence in Artificial Intelligence and Data Science. The work of A.T. was partially supported by the Analytical center under the RF Government (subsidy agreement 000000D730321P5Q0002, Grant No. 70-2021-00145 02.11.2021).

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

# Contents

# A  Three unbiased ways to compress

The main focus of this paper is handling the server-to-worker communication costs. To explain better where the improvements outlined in the main part of this paper come from, let us first consider the scenario where uplink communication cost is negligible but downilnk communication cost is not. While we do no necessarily say that this is a realistic setup, examining it first enables us to understand how downlink compression should be performed, capturing all the intricacies.

Existing algorithms with lossy s2w and w2s communication have a certain common feature. The compression mechanism employed on the clients is very different from the one used on the server: while each client transmits to the server a different message, specific to the data stored on each device, the server broadcasts the same update to all clients. We want to question this algorithmic step and suggest to the reader that if compression is applied multiple times and each worker receives its individual update, then intuitively more information can be transmitted. A well-designed algorithm should be able to take advantage of this.

One can depart from the usual approach of sending the same update to all workers in two ways: a) compress the update $n$ times *independently*, or b) produce $n$ such updates in a *correlated* way. Either way, the server broadcasts $n$ different compressed messages rather than one, and sends a *different* update to each worker. The key discovery here is that both a) and b) are mathematically *provably better* than the prevalent approach of sending the same update to all clients.

This is a crucial improvement in a system where the above setup is a good approximation of reality. And even if it is not, and the current model is not perfectly capturing the reality, we can accept it for now, as it allows us to focus on the novel aspects of the approach. With that said, these considerations can serve as a starting point for thinking about *bidirectional compression*: having focused on the simplified setup and equipped with knowledge on how the compression on the master should be performed, we employ this mechanism in more complex scenarios (see Section 5).

Let us now describe the three possible ways to perform compression on the server.

**"Same" compressors.** The prevalent approach in downlink compression is to transmit the same update to all workers. To illustrate this, let us call a collection $\mathcal{C}_1, \ldots, \mathcal{C}_n$ of compressors "SameRand$K$" if for all $i \in [n]$ we have $\mathcal{C}_i = \mathcal{C}$ for some Rand$K$ compressor $\mathcal{C}$. Now, consider one iteration $t$ of MARINA-P with SameRand$K$ compressor. The server calculates $\mathcal{C}_i^t(x^{t+1} - x^t)$ for $i \in [n]$, but in this case, $\mathcal{C}_1^t(x^{t+1} - x^t) = \ldots = \mathcal{C}_n^t(x^{t+1} - x^t) = \mathcal{C}^t(x^{t+1} - x^t)$. Thus, applying a collection of SameRand$K$ compressors to some vector $x \in \mathbb{R}^d$ is equivalent to using a single Rand$K$ compression operator and transmitting the same message $\mathcal{C}(x)$ to all workers.

**Independent Compressors.** Rather than setting $\mathcal{C}_i(x) = \mathcal{C}(x)$ for all $i \in [n]$, one can break the dependency between the messages and allow the compressors to differ. For illustrational purposes, suppose that $\mathcal{C}_i, i \in [n]$ are independent Rand$K$ compressors (Assumption 1.6). Then, applying such a collection of mappings to the vector of interest $x \in \mathbb{R}^d$, one obtains $n$ distinct and independent sparse vectors $\mathcal{C}_1(x), \ldots, \mathcal{C}_n(x)$.

*Remark* A.1. We are aware of only one method that uses $n$ distinct compressors in downlink compression, Rand-MCM by Philippenko and Dieuleveut (2021). Given the absence of results in the non-convex case, let us compare the communication complexities of Rand-MCM and M3 under the Polyak-Łojasiewicz condition (Assumption D.9), which holds under strong convexity. In the strongly convex case, the proved iteration complexity of Rand-MCM is

$$\Omega\left(\frac{L_{\max}}{\mu}\left(\omega_P^{3/2} + \frac{\omega_P \omega_D^{1/2}}{\sqrt{n}} + \frac{\omega_D}{n}\right)\log\frac{\delta^0}{\varepsilon}\right).$$

Assuming for simplicity that the server and the workers use Rand$K$ compressors with $K = d/n$, this gives the total communication complexity of

$$\Omega\left(\frac{d}{n} \times \frac{L_{\max}}{\mu}\left(\omega_P^{3/2} + \frac{\omega_P \omega_D^{1/2}}{\sqrt{n}} + \frac{\omega_D}{n}\right)\log\frac{\delta^0}{\varepsilon}\right) = \Omega\left(\frac{d\sqrt{n}L_{\max}}{\mu}\log\frac{\delta^0}{\varepsilon}\right),$$

which is getting *worse* as the number of workers $n$ increases. Meanwhile, by Corollary E.10, the total communication complexity of M3 (where $\mathcal{C}_i^t$ are the Perm$K$ compressors and $\mathcal{Q}_i^t$ are independent

Rand$K$ compressors, both with $K = d/n$) under the Polyak-Łojasiewicz condition is

$$\mathcal{O}\left(\left(\frac{dL_{\max}}{n^{1/3}\mu} + \frac{dL_A}{\mu} + d\right)\log\frac{\delta^0}{\varepsilon}\right).$$

Since $n$ is typically large, the total communication complexity of M3 can be much better than that of Rand-MCM.

**Correlated Compressors.** In their work, Szlendak et al. (2021) introduce an alternative class of compressors, which satisfy the following condition:

**Definition A.2** (AB-inequality (Szlendak et al., 2021)). There exist constants $A, B \geq 0$ such that the random operators $\mathcal{C}_1, \ldots \mathcal{C}_n$ satisfy

$$\mathbb{E}\left[\mathcal{C}_i(x)\right] = x,$$

$$\mathbb{E}\left[\left\|\frac{1}{n}\sum_{i=1}^{n}\mathcal{C}_i(x_i) - \frac{1}{n}\sum_{i=1}^{n}x_i\right\|^2\right] \leq A\frac{1}{n}\sum_{i=1}^{n}\|x_i\|^2 - B\left\|\frac{1}{n}\sum_{i=1}^{n}x_i\right\|^2 \tag{16}$$

for all $x, x_1, \ldots, x_n \in \mathbb{R}^d$. If these conditions hold, we write $\{\mathcal{C}_i\}_{i=1}^{n} \in \mathbb{U}(A, B)$.

Following on this idea, we introduce the concept of a *collection of correlated compressors*.

**Definition A.3** (Collection of Correlated Compressors). There exists a constant $\theta \geq 0$ such that the random operators $\mathcal{C}_1, \ldots \mathcal{C}_n$ satisfy:

$$\mathbb{E}\left[\mathcal{C}_i(x)\right] = x$$

$$\mathbb{E}\left[\left\|\frac{1}{n}\sum_{i=1}^{n}\mathcal{C}_i(x) - x\right\|^2\right] \leq \theta\|x\|^2 \tag{17}$$

for all $x \in \mathbb{R}^d$. If these conditions hold, we write $\{\mathcal{C}_i\}_{i=1}^{n} \in \mathbb{P}(\theta)$.

Definition A.3 will play a key role in our upcoming advancements. But what makes this assumption reasonable?

First, it is easy to note that condition (17) is weaker than (16). Indeed, if $\{\mathcal{C}_i\}_{i=1}^{n} \in \mathbb{U}(A, B)$, then inequality (17) holds with $\theta := A - B$. It turns out that it is in fact strictly weaker, as the following example shows.

*Example* A.4. Let $n = 2, d = 1$. Let $\{\zeta_x : x \in \mathbb{R}\}$ be a collection of independent Cauchy variables indexed by real numbers. Define $\mathcal{C}_1(u) = u + \zeta_u$, and $\mathcal{C}_2(u) = u - \zeta_u$. Then

$$\frac{1}{2}\left(\mathcal{C}_1(u) + \mathcal{C}_2(u)\right) = u,$$

so $\mathcal{C}_1(u)$ and $\mathcal{C}_2(u)$ satisfy Definition A.3 with $\theta = 0$. However, for $u_1 \neq u_2$, by the properties of Cauchy distribution we have

$$\mathbb{E}\left[\left(\frac{1}{2}\left(\mathcal{C}_1(u) + \mathcal{C}_2(u)\right) - \frac{1}{2}\left(u_1 + u_2\right)\right)^2\right] = \mathbb{E}\left[\left(\frac{1}{2}\left(\zeta_1 + \zeta_2\right)\right)^2\right]$$

$$= \frac{1}{4}\mathbb{E}\left[\zeta_1^2 + \zeta_2^2 + 2\zeta_1\zeta_2\right] = \infty.$$

Thus, $\mathcal{C}_1(u)$ and $\mathcal{C}_2(u)$ do not satisfy Definition A.2.

In fact, the condition specified in Definition A.3 does not impose any restrictions on the compressor class when working with unbiased compressors. This is because, for any set of compressors $\mathcal{C}_1, \ldots, \mathcal{C}_n \in \mathbb{U}(\omega)$, there exists $\theta \geq 0$ such that $\{\mathcal{C}_i\}_{i=1}^{n} \in \mathbb{P}(\theta)$, as shown in the following lemma.

**Lemma A.5.**

    1. *Let $\mathcal{C}_1, \ldots, \mathcal{C}_n$ be a collection of compressors such that $\mathcal{C}_i \in \mathbb{U}(\omega)$ for all $i \in [n]$. Then $\{\mathcal{C}_i\}_{i=1}^{n} \in \mathbb{P}(\omega)$.*

2. Let us further assume that $\mathcal{C}_1, \ldots, \mathcal{C}_n$ are independent (Assumption 1.6). Then $\{\mathcal{C}_i\}_{i=1}^n \in \mathbb{P}(\omega/n)$.

*Proof.*    1. Jensen's inequality gives

$$\mathbb{E}\left[\left\|\frac{1}{n}\sum_{i=1}^n \mathcal{C}_i(u) - u\right\|^2\right] \overset{(50)}{\leq} \frac{1}{n}\sum_{i=1}^n \mathbb{E}\left[\|\mathcal{C}_i(u) - u\|^2\right] \overset{\text{Def.1.4}}{\leq} \frac{1}{n}\sum_{i=1}^n \omega\|u\|^2 = \omega\|u\|^2,$$

so $\theta = \omega$.

2. Using independence of compressors, we have

$$\mathbb{E}\left[\left\|\frac{1}{n}\sum_{i=1}^n \mathcal{C}_i(u) - u\right\|^2\right] = \frac{1}{n^2}\sum_{i=1}^n \mathbb{E}\left[\|\mathcal{C}_i(u) - u\|^2\right] \overset{\text{Def.1.4}}{\leq} \frac{1}{n^2}\sum_{i=1}^n \omega\|u\|^2 = \frac{\omega}{n}\|u\|^2.$$

Thus $\theta = \omega/n$.

$\square$

However, the true advantages of employing correlated compressors become apparent when the definition holds with $\theta = 0$, as in the case of PermK compressors.

**Lemma A.6.** *Let $\mathcal{C}_1, \ldots, \mathcal{C}_n$ be a collection of a) SameRandK, b) independent RandK, c) PermK compressors. Then*

a) *$\{\mathcal{C}_i\}_{i=1}^n \in \mathbb{P}(\omega)$ and $\mathcal{C}_i \in \mathbb{U}(\omega)$ where $\omega = d/K - 1$ for all $i \in [n]$,*

b) *$\{\mathcal{C}_i\}_{i=1}^n \in \mathbb{P}(\omega/n)$ and $\mathcal{C}_i \in \mathbb{U}(\omega)$ where $\omega = d/K - 1$ for all $i \in [n]$,*

c) *$\{\mathcal{C}_i\}_{i=1}^n \in \mathbb{P}(0)$ and $\mathcal{C}_i \in \mathbb{U}(\omega)$ where $\omega = n - 1$ for all $i \in [n]$.*

*Proof.* Unbiasedness follows easily from definitions of compressors (the proof for PermK compressors can be found in Szlendak et al. (2021)). That RandK $\in \mathbb{U}(d/K - 1)$ (and hence trivially SameRandK $\in \mathbb{U}(d/K - 1)$) is a well-known fact. Next, the fact that a) $\{\mathcal{C}_i\}_{i=1}^n \in \mathbb{P}(\omega)$ for SameRandK compressors and b) $\{\mathcal{C}_i\}_{i=1}^n \in \mathbb{P}(\omega/n)$ for independent RandK compressors follows directly from Lemma A.5.

To compute $\omega$ for PermK compressor, first assume that $d \geq n$. Then $\mathbb{E}\left[\|\mathcal{C}_i(x)\|^2\right] = n\|x\|^2$ (Szlendak et al., 2021), so

$$\mathbb{E}\left[\|\mathcal{C}_i(x) - x\|^2\right] \overset{(48)}{=} \mathbb{E}\left[\|\mathcal{C}_i(x)\|^2\right] - \|x\|^2 = (n-1)\|x\|^2.$$

Similarly, suppose that $d \leq n$, and write $n$ as $n = qd + r$, where $q \in \mathbb{N}_{>0}$ and $0 \leq r < d$. Then $\mathbb{E}\left[\|\mathcal{C}_i(x)\|^2\right] = n/q\|x\|^2$ (Szlendak et al., 2021), and hence

$$\mathbb{E}\left[\|\mathcal{C}_i(x) - x\|^2\right] \overset{(48)}{=} \mathbb{E}\left[\|\mathcal{C}_i(x)\|^2\right] - \|x\|^2 = \left(\frac{n}{q} - 1\right)\|x\|^2 \leq (n-1)\|x\|^2.$$

In both cases $\omega = n - 1$.

Finally, by construction of PermK, we have $\frac{1}{n}\sum_{i=1}^n \mathcal{C}_i(x) = x$, implying $\theta = 0$.    $\square$

In what follows, when considering the PermK compressor, we shall assume for simplicity that $d \geq n$. The results for $d < n$ are analogous.

## B   Biased Compressors

In addition to unbiased compressors (Definition 1.4), the literature of compressed methods distinguishes another class of mappings:

**Algorithm 1** MARINA-P

---

1: **Input:** initial model $x_0 \in \mathbb{R}^d$ (stored on the server), initial model shifts $w_1^0 = \ldots = w_n^0 = x^0$ (stored on the workers), step size $\gamma > 0$, probability $0 < p \leq 1$, compressors $\mathcal{C}_1^t, \ldots, \mathcal{C}_n^t \in \mathbb{U}(\omega_P)$

2: **for** $t = 0, \ldots, T$ **do**
3:     **for** $i = 1, \ldots, n$ in parallel **do**
4:         Calculate $\nabla f_i(w_i^t)$ and send it to the server          *Workers evaluate the gradients at the current model estimate*
5:     **end for**
    **On the server:**
6:     $g^t = \frac{1}{n} \sum_{i=1}^n \nabla f_i(w_i^t)$          *Server averages the messages received from the workers*
7:     $x^{t+1} = x^t - \gamma g^t$          *Server takes a gradient-type step to update the global model*
8:     Sample $c^t \sim \text{Bernoulli}(p)$
9:     **if** $c^t = 0$ **then**
10:        Send $\mathcal{C}_i^t(x^{t+1} - x^t)$ to worker $i$ for $i \in [n]$      *Server sends compressed messages to all workers w.p. $1 - p$*
11:     **else**
12:        Send $x^{t+1}$ to worker $i$ for $i \in [n]$      *Server sends the same uncompressed message to all workers w.p. $p$*
13:     **end if**
    **On the workers:**
14:     **for** $i = 1, \ldots, n$ in parallel **do**
15:        $w_i^{t+1} = \begin{cases} x^{t+1} & \text{if } c^t = 1, \\ w_i^t + \mathcal{C}_i^t(x^{t+1} - x^t) & \text{if } c^t = 0 \end{cases}$      *Worker $i$ updates its local model shift*
16:     **end for**
17: **end for**

---

**Definition B.1.** A stochastic mapping $\mathcal{C} : \mathbb{R}^d \to \mathbb{R}^d$ is a *biased compressor* if there exists $\alpha \in (0, 1]$ such that

$$\mathbb{E}\left[\|\mathcal{C}(x) - x\|^2\right] \leq (1 - \alpha)\|x\|^2 \quad \forall x \in \mathbb{R}^d. \tag{18}$$

The family of such compressors is denoted by $\mathbb{B}(\alpha)$. It is well-known that if $\mathcal{C} \in \mathbb{U}(\omega)$, then $(\omega + 1)^{-1}\mathcal{C} \in \mathbb{B}\left((\omega + 1)^{-1}\right)$, meaning that the family of biased compressors is broader. A canonical example is the Top$K \in \mathbb{B}(K/d)$ compressor, which preserves the $K$ largest in magnitude coordinates of the input vector (Beznosikov et al., 2020).

# C   Properties of $L_A$ and $L_B$

We first prove the results from Section 4, starting with calculating the constants $L_A$ and $L_B$ from Assumption 4.2 in some special cases.

**Theorem 4.4.** *For all $i \in [n]$, assume that the functions $f_i$ are $L_i$–smooth (Assumption 1.5). Then, Assumption 4.2 holds with $L_A = L_{\max}$ and $L_B = 0$.*

*Proof.* From Assumption 1.5 it follows that

$$\left\|\frac{1}{n}\sum_{i=1}^n (\nabla f_i(x + u_i) - \nabla f_i(x))\right\|^2 \overset{(50)}{\leq} \frac{1}{n}\sum_{i=1}^n \|\nabla f_i(x + u_i) - \nabla f_i(x)\|^2$$

$$\overset{\text{Ass.1.5}}{\leq} \frac{1}{n}\sum_{i=1}^n L_i^2 \|u_i\|^2$$

$$\leq L_{\max}^2 \left(\frac{1}{n}\sum_{i=1}^n \|u_i\|^2\right),$$

so Assumption 4.2 holds with $L_A = L_{\max}$ and $L_B = 0$. $\qquad\square$

**Theorem 4.5.** *For all $i \in [n]$, assume that the functions $f_i$ are homogeneous quadratics defined in (8). Then, Assumption 4.2 holds with $L_A = 0$ and $L_B = \|\mathbf{A}\|$.*

*Proof.* It is easy to verify that

$$\left\| \frac{1}{n} \sum_{i=1}^{n} (\nabla f_i(x + u_i) - \nabla f_i(x)) \right\|^2 = \left\| \frac{1}{n} \sum_{i=1}^{n} (\mathbf{A}(x + u_i) + b - (\mathbf{A}x + b)) \right\|^2$$

$$= \left\| \mathbf{A} \left( \frac{1}{n} \sum_{i=1}^{n} u_i \right) \right\|^2$$

$$\leq \|\mathbf{A}\|^2 \left\| \frac{1}{n} \sum_{i=1}^{n} u_i \right\|^2,$$

meaning that Assumption 4.2 holds with $L_A = 0$ and $L_B = \|\mathbf{A}\|$. □

**Lemma C.1.** *Let Assumption 1.5 hold. Then, there exist constants $L_A, L_B \geq 0$ such that Assumption 4.2 holds and $L_A^2 + L_B^2 \leq L_{\max}^2$.*

*Proof.* Assumption 1.5 gives

$$\left\| \frac{1}{n} \sum_{i=1}^{n} (\nabla f_i(x + u_i) - \nabla f_i(x)) \right\|^2 \overset{(50)}{\leq} \frac{1}{n} \sum_{i=1}^{n} \|\nabla f_i(x + u_i) - \nabla f_i(x)\|^2$$

$$\overset{\text{Ass.1.5}}{\leq} \frac{1}{n} \sum_{i=1}^{n} L_i^2 \|u_i\|^2$$

$$\leq L_{\max}^2 \left( \frac{1}{n} \sum_{i=1}^{n} \|u_i\|^2 \right),$$

and hence Assumption 4.2 holds with $L_A^2 = L_{\max}^2$ and $L_B^2 = 0$. □

*Remark* C.2. Under Assumption 4.2 we have

$$\left\| \frac{1}{n} \sum_{i=1}^{n} (\nabla f_i(x + u_i) - \nabla f_i(x)) \right\|^2 \overset{\text{Ass.4.2}}{\leq} L_A^2 \left( \frac{1}{n} \sum_{i=1}^{n} \|u_i\|^2 \right) + L_B^2 \left\| \frac{1}{n} \sum_{i=1}^{n} u_i \right\|^2$$

$$\overset{\text{Ass.1.5}}{\leq} \left( L_A^2 + L_B^2 \right) \left( \frac{1}{n} \sum_{i=1}^{n} \|u_i\|^2 \right),$$

so, in principle, one could always set $L_B^2 = 0$. However, the bound could be tightened by decreasing $L_A$ and increasing $L_B$. The smaller $L_A$, the better the performance of our algorithms (see Corollaries D.4 and E.3).

Now, we proceed to prove the result that relates the values of $L_A$ and $L_B$ to the Hessians of the functions $f_i$.

**Theorem 4.8.** *Assume that the functions $f_i$ are twice continuously differentiable, $L_i$–smooth (Assumption 1.5), and that there exist $D_i \geq 0$ such that*

$$\sup_{z_1,\ldots,z_n \in \mathbb{R}^d} \left\| \nabla^2 f_i(z_i) - \frac{1}{n} \sum_{j=1}^{n} \nabla^2 f_j(z_j) \right\| \leq D_i \tag{11}$$

*for all $i \in [n]$. Then, Assumption 4.2 holds with $L_A = \sqrt{2} \max_{i \in [n]} D_i \leq 2\sqrt{2} \max_{i \in [n]} L_i$ and $L_B = \sqrt{2} \left( \frac{1}{n} \sum_{i=1}^{n} L_i \right)$.*

*Proof.* By the fundamental theorem of calculus,

$$\nabla f_i(x + u_i) - \nabla f_i(x) = \int_0^1 \nabla^2 f_i(x + tu_i) u_i dt = \left( \int_0^1 \nabla^2 f_i(x + tu_i) dt \right) u_i = \mathbf{Q}_i u_i,$$

where $\mathbf{Q}_i = \int_0^1 \nabla^2 f_i(x + tu_i)dt$. Letting $\mathbf{Q} = \frac{1}{n}\sum_{i=1}^n \mathbf{Q}_i$, we can write

$$\left\| \frac{1}{n}\sum_{i=1}^n \left(\nabla f_i(x+u_i) - \nabla f_i(x)\right) \right\|^2 = \left\| \frac{1}{n}\sum_{i=1}^n \mathbf{Q}_i u_i \right\|^2$$

$$= \left\| \frac{1}{n}\sum_{i=1}^n \left(\mathbf{Q}_i - \mathbf{Q}\right)u_i + \mathbf{Q}\left(\frac{1}{n}\sum_{i=1}^n u_i\right) \right\|^2$$

$$\overset{(44)}{\leq} 2\left\| \frac{1}{n}\sum_{i=1}^n \left(\mathbf{Q}_i - \mathbf{Q}\right)u_i \right\|^2 + 2\left\| \mathbf{Q}\left(\frac{1}{n}\sum_{i=1}^n u_i\right) \right\|^2$$

$$\overset{(50)}{\leq} 2\frac{1}{n}\sum_{i=1}^n \left\|\left(\mathbf{Q}_i - \mathbf{Q}\right)u_i\right\|^2 + 2\left\|\mathbf{Q}\right\|^2 \left\| \frac{1}{n}\sum_{i=1}^n u_i \right\|^2$$

$$\leq 2\frac{1}{n}\sum_{i=1}^n \left\|\mathbf{Q}_i - \mathbf{Q}\right\|^2 \left\|u_i\right\|^2 + 2\left\|\mathbf{Q}\right\|^2 \left\| \frac{1}{n}\sum_{i=1}^n u_i \right\|^2.$$

Further,

$$\left\|\mathbf{Q}_i - \mathbf{Q}\right\| = \left\| \int_0^1 \nabla^2 f_i(x+tu_i)dt - \frac{1}{n}\sum_{j=1}^n \int_0^1 \nabla^2 f_j(x+tu_j)dt \right\|$$

$$= \left\| \int_0^1 \nabla^2 f_i(x+tu_i)dt - \int_0^1 \frac{1}{n}\sum_{j=1}^n \nabla^2 f_j(x+tu_j)dt \right\|$$

$$= \left\| \int_0^1 \frac{1}{n}\sum_{j=1}^n \left(\nabla^2 f_i(x+tu_i) - \nabla^2 f_j(x+tu_j)\right)dt \right\|$$

$$\leq \int_0^1 \left\| \nabla^2 f_i(x+tu_i) - \frac{1}{n}\sum_{j=1}^n \nabla^2 f_j(x+tu_j) \right\| dt$$

$$\leq \int_0^1 D_i dt = D_i,$$

and

$$\left\|\mathbf{Q}\right\| = \left\| \frac{1}{n}\sum_{j=1}^n \int_0^1 \nabla^2 f_j(x+tu_j)dt \right\| = \left\| \int_0^1 \frac{1}{n}\sum_{j=1}^n \nabla^2 f_j(x+tu_j)dt \right\|$$

$$\leq \int_0^1 \left\| \frac{1}{n}\sum_{j=1}^n \nabla^2 f_j(x+tu_j) \right\| dt \overset{(50)}{\leq} \int_0^1 \frac{1}{n}\sum_{j=1}^n \left\|\nabla^2 f_j(x+tu_j)\right\| dt$$

$$\leq \int_0^1 \frac{1}{n}\sum_{j=1}^n L_j dt = \frac{1}{n}\sum_{j=1}^n L_j.$$

By combining the above, we get

$$\left\| \frac{1}{n}\sum_{i=1}^n \left(\nabla f_i(x+u_i) - \nabla f_i(x)\right) \right\|^2 \leq 2\frac{1}{n}\sum_{i=1}^n D_i^2 \left\|u_i\right\|^2 + 2\left(\frac{1}{n}\sum_{j=1}^n L_j\right)^2 \left\| \frac{1}{n}\sum_{i=1}^n u_i \right\|^2$$

$$\leq 2\left(\max_i D_i^2\right)\left\|u_i\right\|^2 + 2\left(\frac{1}{n}\sum_{j=1}^n L_j\right)^2 \left\| \frac{1}{n}\sum_{i=1}^n u_i \right\|^2,$$

which means that Assumption 4.2 holds with $L_A^2 = 2(\max_i D_i^2)$ and $L_B^2 = 2\left(\frac{1}{n}\sum_{i=1}^n L_i\right)^2$. $\qquad\square$

*Remark* C.3. Clearly, if Assumption 1.5 holds, i.e., if there exists $L_i \geq 0$ such that $\sup_{z_i \in \mathbb{R}^d} \left\| \nabla^2 f_i(z_i) \right\| \leq L_i$ for all $i \in [n]$, then there exists $D_i$ such that $\sup_{z_1,\ldots,z_n \in \mathbb{R}^d} \left\| \nabla^2 f_i(z_i) - \frac{1}{n} \sum_{j=1}^n \nabla^2 f_j(z_j) \right\| \leq D_i$, which means that this latter condition is not restrictive. Indeed,

$$\left\| \nabla^2 f_i(z_i) - \frac{1}{n} \sum_{j=1}^n \nabla^2 f_j(z_j) \right\| \leq \left\| \nabla^2 f_i(z_i) \right\| + \left\| \frac{1}{n} \sum_{j=1}^n \nabla^2 f_j(z_j) \right\|$$

$$\leq \left\| \nabla^2 f_i(z_i) \right\| + \frac{1}{n} \sum_{j=1}^n \left\| \nabla^2 f_j(z_j) \right\|$$

$$\leq L_i + \frac{1}{n} \sum_{j=1}^n L_j.$$

However, $D_i$ can be small even if the constants $\{L_i\}$ are large, as the next theorem shows.

**Theorem 4.9.** *Assume that $f_i(x) = \frac{1}{2} x^\top \mathbf{A}_i x + b_i^\top x + c_i$, where $\mathbf{A}_i \in \mathbb{R}^{d \times d}$ is symmetric but not necessarily positive semidefinite, $b_i \in \mathbb{R}^d$ and $c_i \in \mathbb{R}$ for $i \in [n]$. Define $\mathbf{A} = \frac{1}{n} \sum_{i=1}^n \mathbf{A}_i$. Then, Assumption 4.2 holds with $L_A = \sqrt{2} \max_{i \in [n]} \|\mathbf{A}_i - \mathbf{A}\|$ and $L_B = \sqrt{2} \left( \frac{1}{n} \sum_{i=1}^n \|\mathbf{A}_i\| \right).$*

*Proof.* In this case $\nabla^2 f_i(z_i) \equiv \mathbf{A}_i$, and the result easily follows from Theorem 4.8. $\square$

# D  Convergence of MARINA-P in the General Case

## D.1  Main Results

As promised, we now present a result generalizing Theorem 4.6 to all unbiased compressors.

**Theorem D.1.** *Let Assumptions 1.1, 1.2 and 4.2 be satisfied and suppose that $\{\mathcal{C}_i^t\}_{i=1}^n \in \mathbb{P}(\theta)$ (Def. A.3) and $\mathcal{C}_i^t \in \mathbb{U}(\omega_P)$ (Def. 1.4) for all $i \in [n]$. Let*

$$0 < \gamma \leq \frac{1}{L + \sqrt{(L_A^2 \omega_P + L_B^2 \theta) \left( \frac{1}{p} - 1 \right)}}.$$

*Letting*

$$\Psi^t = f(x^t) - f^* + \frac{\gamma L_A^2}{2p} \frac{1}{n} \sum_{i=1}^n \left\| w_i^t - x^t \right\|^2 + \frac{\gamma L_B^2}{2p} \left\| w^t - x^t \right\|^2,$$

*for each $T \geq 1$ we have*

$$\sum_{t=0}^{T-1} \frac{1}{T} \mathbb{E} \left[ \left\| \nabla f(x^t) \right\|^2 \right] \leq \frac{2 \Psi^0}{\gamma T}.$$

Let us provide some important examples:

**Theorem D.2.** *Let Assumptions 1.1, 1.2 and 4.2 be satisfied. Choose*

$$\gamma = \begin{cases} \left( L + \sqrt{(L_A^2 + L_B^2) \, \omega_P / p} \right)^{-1} & \text{for SameRandK } \textit{compressors} \\ \left( L + \sqrt{(L_A^2 + L_B^2/n) \, \omega_P / p} \right)^{-1} & \text{for independent RandK } \textit{compressors} \\ \left( L + L_A \sqrt{\omega_P / p} \right)^{-1} & \text{for PermK } \textit{compressors} \end{cases}$$

*and set $w_i^0 = x^0$ for all $i \in [n]$. Then* MARINA-P *finds an $\varepsilon$–stationary point after*

$$
\bar{T} = \begin{cases}
\mathcal{O}\left(\dfrac{\delta^0\left(L+\sqrt{(L_A^2+L_B^2)\omega_P/p}\right)}{\varepsilon}\right) & \text{for SameRandK } compressors \\[3ex]
\mathcal{O}\left(\dfrac{\delta^0\left(L+\sqrt{(L_A^2+L_B^2/n)\omega_P/p}\right)}{\varepsilon}\right) & \text{for independent RandK } compressors \\[3ex]
\mathcal{O}\left(\dfrac{\delta^0\left(L+L_A\sqrt{\omega_P/p}\right)}{\varepsilon}\right) & \text{for PermK } compressors
\end{cases}
$$

*iterations.*

*Remark* D.3.

- The result for Perm$K$ compressors proves Theorem 4.6.

- The above theorem demonstrates the complexities for a) SameRand$K$, b) independent Rand$K$ and c) Perm$K$ compressors. However, the result applies to any families of compressors such that for all $t \geq 0$ we have a) $\mathcal{C}_1^t = \ldots = \mathcal{C}_n^t = \mathcal{C}^t \in \mathbb{U}(\omega_P)$, b) $\mathcal{C}_1^t, \ldots, \mathcal{C}_n^t \in \mathbb{U}(\omega_P)$ are independent, and c) $\mathcal{C}_1^t, \ldots, \mathcal{C}_n^t \in \mathbb{U}(\omega_P) \cap \mathbb{P}(\theta)$, respectively.

We now derive the communication complexities:

**Corollary D.4.** *Let us take $p = 1/n$ and set $K = d/n$ (corresponding to the sparsification level of a Perm$K$ compressor). Then, in the view of Theorem D.2, the average s2w communication complexity of* MARINA-P *is*

$$
\begin{cases}
\mathcal{O}\left(\dfrac{d\delta^0 L}{n\varepsilon} + \dfrac{d\delta^0}{\varepsilon}\sqrt{L_A^2 + L_B^2}\right) & \text{for SameRandK } compressors \\[2.5ex]
\mathcal{O}\left(\dfrac{d\delta^0 L}{n\varepsilon} + \dfrac{d\delta^0}{\varepsilon}\sqrt{L_A^2 + \dfrac{L_B^2}{n}}\right) & \text{for independent RandK } compressors \\[2.5ex]
\mathcal{O}\left(\dfrac{d\delta^0 L}{n\varepsilon} + \dfrac{d\delta^0}{\varepsilon}L_A\right) & \text{for PermK } compressors
\end{cases}
\tag{19}
$$

*Remark* D.5.

- The result for Perm$K$ compressors proves Corollary 4.7.

- The key observation from (19) is the dependence on $L_A$ and $L_B$. In particular, if $L_A \approx 0$ (which is the case, e.g., for homogeneous quadratics), the above communication complexities are

$$
\begin{cases}
\mathcal{O}\left(\dfrac{\delta^0}{\varepsilon}d\left(\dfrac{L}{n} + L_B\right)\right) & \text{for SameRandK compressors,} \\[2.5ex]
\mathcal{O}\left(\dfrac{\delta^0}{\varepsilon}d\left(\dfrac{L}{n} + \dfrac{L_B}{\sqrt{n}}\right)\right) & \text{for independent RandK compressors,} \\[2.5ex]
\mathcal{O}\left(\dfrac{\delta^0}{\varepsilon}d\dfrac{L}{n}\right) & \text{for PermK compressors.}
\end{cases}
$$

Hence, only by sending different messages to different clients, one obtains complexities improving with $n$. In particular, for Perm$K$, the complexity scales linearly with the number of workers.

## D.2 Proofs

To prove the results from the previous section, we first establish several identities and inequalities satisfied by the sequences $\{w_1^t, \ldots, w_n^t\}_{t \geq 0}$. We start by studying the evolution of the quantity $\|w_i^t - x^t\|^2$. In what follows, $\mathbb{E}_t[\cdot]$ denotes the expectation conditioned on the first $t$ iterations.

**Lemma D.6.** *Let $\mathcal{C}_i^t \in \mathbb{U}(\omega_P)$ for all $i \in [n]$. Then*

$$
\frac{1}{n}\sum_{i=1}^n \mathbb{E}\left[\|w_i^{t+1} - x^{t+1}\|^2\right] \leq (1-p)\frac{1}{n}\sum_{i=1}^n \mathbb{E}\left[\|w_i^t - x^t\|^2\right] + (1-p)\omega_P \mathbb{E}\left[\|x^{t+1} - x^t\|^2\right].
$$

*Proof.* In the view of definition of $w^{t+1}$, we get

$$\mathbb{E}_t \left[ \left\| w_i^{t+1} - x^{t+1} \right\|^2 \right]$$

$$= (1-p)\mathbb{E}_t \left[ \left\| w_i^t + \mathcal{C}_i^t(x^{t+1} - x^t) - x^{t+1} \right\|^2 \right]$$

$$\overset{(48)}{=} (1-p)\mathbb{E}_t \left[ \left\| \mathcal{C}_i^t(x^{t+1} - x^t) - (x^{t+1} - x^t) \right\|^2 \right] + (1-p)\mathbb{E}_t \left[ \left\| w_i^t - x^t \right\|^2 \right]$$

$$\overset{\text{Def.1.4}}{\leq} (1-p)\omega_P \left\| x^{t+1} - x^t \right\|^2 + (1-p) \left\| w_i^t - x^t \right\|^2.$$

Averaging, taking expectation and using the tower property, we get the result. $\qquad\square$

This lemma is less powerful: it is *not* an identity, and hence some information is lost. Moreover, it focuses on a single client $i$, and is therefore not able to take advantage of the correlation among the compressors. On the other hand, it can be used in the convergence analysis without any need to restrict the function class.

Next, we study the evolution of the quantity $\|w^t - x^t\|^2$.

**Lemma D.7.** *Let $\{\mathcal{C}_i^t\}_{i=1}^n \in \mathbb{P}(\theta)$. Then*

$$\mathbb{E} \left[ \left\| w^{t+1} - x^{t+1} \right\|^2 \right] \leq (1-p)\mathbb{E} \left[ \left\| w^t - x^t \right\|^2 \right] + (1-p)\theta\mathbb{E} \left[ \left\| x^{t+1} - x^t \right\|^2 \right].$$

*Proof.* In the view of definition of $w^{t+1}$, we get

$$\mathbb{E}_t \left[ \left\| w^{t+1} - x^{t+1} \right\|^2 \right]$$

$$= \mathbb{E}_t \left[ \left\| \frac{1}{n} \sum_{i=1}^n w_i^{t+1} - x^{t+1} \right\|^2 \right]$$

$$= (1-p)\mathbb{E}_t \left[ \left\| \frac{1}{n} \sum_{i=1}^n (w_i^t + \mathcal{C}_i^t(x^{t+1} - x^t)) - x^{t+1} \right\|^2 \right]$$

$$= (1-p)\mathbb{E}_t \left[ \left\| \frac{1}{n} \sum_{i=1}^n \mathcal{C}_i^t(x^{t+1} - x^t) - (x^{t+1} - x^t) - x^t + w^t \right\|^2 \right]$$

$$\overset{(48)}{=} (1-p)\mathbb{E}_t \left[ \left\| \frac{1}{n} \sum_{i=1}^n \mathcal{C}_i^t(x^{t+1} - x^t) - (x^{t+1} - x^t) \right\|^2 \right] + (1-p)\mathbb{E}_t \left[ \left\| w^t - x^t \right\|^2 \right]$$

$$\overset{\text{Def.}A.3}{\leq} (1-p)\theta \left\| x^{t+1} - x^t \right\|^2 + (1-p) \left\| w^t - x^t \right\|^2.$$

Taking expectation and using the tower property, we get the result. $\qquad\square$

This lemma is more powerful since it is able to take advantage of the correlation among the compressors. Indeed, if $\theta = 0$ (as in the case of $\mathrm{Perm}K$ compressors), then it becomes an identity:

$$\mathbb{E} \left[ \left\| w^{t+1} - x^{t+1} \right\|^2 \right] = (1-p)\mathbb{E} \left[ \left\| w^t - x^t \right\|^2 \right].$$

We now prove convergence of MARINA-P in the general case.

**Theorem D.1.** *Let Assumptions 1.1, 1.2 and 4.2 be satisfied and suppose that $\{\mathcal{C}_i^t\}_{i=1}^n \in \mathbb{P}(\theta)$ (Def. A.3) and $\mathcal{C}_i^t \in \mathbb{U}(\omega_P)$ (Def. 1.4) for all $i \in [n]$. Let*

$$0 < \gamma \leq \frac{1}{L + \sqrt{(L_A^2 \omega_P + L_B^2 \theta)\left(\frac{1}{p} - 1\right)}}.$$

*Letting*

$$\Psi^t = f(x^t) - f^* + \frac{\gamma L_A^2}{2p} \frac{1}{n} \sum_{i=1}^n \left\| w_i^t - x^t \right\|^2 + \frac{\gamma L_B^2}{2p} \left\| w^t - x^t \right\|^2,$$

*for each $T \geq 1$ we have*

$$\sum_{t=0}^{T-1} \frac{1}{T} \mathbb{E}\left[\left\|\nabla f(x^t)\right\|^2\right] \leq \frac{2\Psi^0}{\gamma T}.$$

*Proof.* First, combining the inequalities in Lemmas D.6 and D.7, we get

$$\frac{\gamma L_A^2}{2p} \frac{1}{n} \sum_{i=1}^n \mathbb{E}\left[\left\| w_i^{t+1} - x^{t+1} \right\|^2\right] + \frac{\gamma L_B^2}{2p} \mathbb{E}\left[\left\| w^{t+1} - x^{t+1} \right\|^2\right]$$

$$\leq \frac{\gamma L_A^2}{2p}(1-p)\frac{1}{n} \sum_{i=1}^n \mathbb{E}\left[\left\| w_i^t - x^t \right\|^2\right] + \frac{\gamma L_A^2}{2p}(1-p)\omega_P \mathbb{E}\left[\left\| x^{t+1} - x^t \right\|^2\right]$$

$$+ \frac{\gamma L_B^2}{2p}(1-p)\mathbb{E}\left[\left\| w^t - x^t \right\|^2\right] + \frac{\gamma L_B^2}{2p}(1-p)\theta\mathbb{E}\left[\left\| x^{t+1} - x^t \right\|^2\right]$$

$$= \frac{\gamma L_A^2}{2p}(1-p)\frac{1}{n} \sum_{i=1}^n \mathbb{E}\left[\left\| w_i^t - x^t \right\|^2\right] + \frac{\gamma L_B^2}{2p}(1-p)\mathbb{E}\left[\left\| w^t - x^t \right\|^2\right]$$

$$+ \frac{\gamma}{2p}\left(L_A^2 \omega_P + L_B^2 \theta\right)(1-p)\mathbb{E}\left[\left\| x^{t+1} - x^t \right\|^2\right]. \tag{20}$$

Next, using Assumption 4.2, we have

$$\mathbb{E}\left[\left\| g^t - \nabla f(x^t) \right\|^2\right] \quad = \quad \mathbb{E}\left[\left\| \frac{1}{n} \sum_{i=1}^n \left(\nabla f_i(w_i^t) - \nabla f_i(x^t)\right) \right\|^2\right]$$

$$\overset{\text{Ass.4.2}}{\leq} \quad L_A^2 \frac{1}{n} \sum_{i=1}^n \mathbb{E}\left[\left\| w_i^t - x^t \right\|^2\right] + L_B^2 \mathbb{E}\left[\left\| \frac{1}{n} \sum_{i=1}^n w_i^t - x^t \right\|^2\right].$$

Combining the above inequality with Lemma H.1 gives

$$\mathbb{E}\left[\delta^{t+1}\right] \leq \mathbb{E}\left[\delta^t\right] - \frac{\gamma}{2}\mathbb{E}\left[\left\|\nabla f(x^t)\right\|^2\right] - \left(\frac{1}{2\gamma} - \frac{L}{2}\right)\mathbb{E}\left[\left\| x^{t+1} - x^t \right\|^2\right] + \frac{\gamma}{2}\mathbb{E}\left[\left\| g^t - \nabla f(x^t) \right\|^2\right]$$

$$\leq \mathbb{E}\left[\delta^t\right] - \frac{\gamma}{2}\mathbb{E}\left[\left\|\nabla f(x^t)\right\|^2\right] - \left(\frac{1}{2\gamma} - \frac{L}{2}\right)\mathbb{E}\left[\left\| x^{t+1} - x^t \right\|^2\right]$$

$$+ \frac{\gamma}{2}\left(L_A^2 \frac{1}{n} \sum_{i=1}^n \mathbb{E}\left[\left\| w_i^t - x^t \right\|^2\right] + L_B^2 \mathbb{E}\left[\left\| w^t - x^t \right\|^2\right]\right). \tag{21}$$

By adding inequalities (20) and (21), we get

$$\mathbb{E}\left[\Psi^{t+1}\right] = \mathbb{E}\left[\delta^{t+1}\right] + \frac{\gamma L_A^2}{2p} \frac{1}{n} \sum_{i=1}^n \mathbb{E}\left[\left\| w_i^{t+1} - x^{t+1} \right\|^2\right] + \frac{\gamma L_B^2}{2p} \mathbb{E}\left[\left\| w^{t+1} - x^{t+1} \right\|^2\right]$$

$$\leq \mathbb{E}\left[\delta^t\right] - \frac{\gamma}{2}\mathbb{E}\left[\left\|\nabla f(x^t)\right\|^2\right] - \left(\frac{1}{2\gamma} - \frac{L}{2}\right)\mathbb{E}\left[\left\| x^{t+1} - x^t \right\|^2\right]$$

$$+ \frac{\gamma}{2}\left(L_A^2 \frac{1}{n} \sum_{i=1}^n \mathbb{E}\left[\left\| w_i^t - x^t \right\|^2\right] + L_B^2 \mathbb{E}\left[\left\| w^t - x^t \right\|^2\right]\right)$$

$$+ \frac{\gamma L_A^2}{2p}(1-p)\frac{1}{n} \sum_{i=1}^n \mathbb{E}\left[\left\| w_i^t - x^t \right\|^2\right] + \frac{\gamma L_B^2}{2p}(1-p)\mathbb{E}\left[\left\| w^t - x^t \right\|^2\right]$$

$$+ \frac{\gamma}{2p}\left(L_A^2 \omega_P + L_B^2 \theta\right)(1-p)\mathbb{E}\left[\left\| x^{t+1} - x^t \right\|^2\right]$$

$$= \mathbb{E}\left[\Psi^t\right] - \frac{\gamma}{2}\mathbb{E}\left[\left\|\nabla f(x^t)\right\|^2\right]$$
$$- \left(\frac{1}{2\gamma} - \frac{L}{2} - \frac{\gamma}{2p}\left(L_A^2\omega_P + L_B^2\theta\right)(1-p)\right)\mathbb{E}\left[\left\|x^{t+1} - x^t\right\|^2\right]$$
$$\leq \mathbb{E}\left[\Psi^t\right] - \frac{\gamma}{2}\mathbb{E}\left[\left\|\nabla f(x^t)\right\|^2\right],$$

where in the last line we use the assumption on the step size and Lemma H.2. Summing up the above inequality for $t = 0, 1, \dots, T-1$ and rearranging the terms, we get

$$\frac{1}{T}\sum_{t=0}^{T-1}\mathbb{E}\left[\left\|\nabla f(x^t)\right\|^2\right] \leq \frac{2}{\gamma T}\left(\mathbb{E}\left[\Psi^0\right] - \mathbb{E}\left[\Psi^T\right]\right) \leq \frac{2\Psi^0}{\gamma T}.$$

$\square$

With the above result, we can establish the iteration and communication complexities of MARINA-P for three different compression schemes described in Appendix A. First, let us prove a result when independent compressors are used.

**Theorem D.8.** *Let Assumptions 1.1, 1.2 and 4.2 be satisfied and suppose that $\mathcal{C}_1^t, \dots, \mathcal{C}_n^t$ is a collection of independent compressors (Assumption 1.6) such that $\mathcal{C}_i^t \in \mathbb{U}(\omega)$ for all $i \in [n]$, $t \in \mathbb{N}$. Choose*

$$\gamma = \left(L + \sqrt{\left(L_A^2 + L_B^2/n\right)\omega_P/p}\right)^{-1}$$

*and set $w_i^0 = x^0$ for all $i \in [n]$. Then MARINA-P finds an $\varepsilon-$stationary point after*

$$\bar{T} = \mathcal{O}\left(\frac{\delta^0\left(L + \sqrt{\left(L_A^2 + L_B^2/n\right)\omega_P/p}\right)}{\varepsilon}\right)$$

*iterations.*

*Proof.* In view of Theorem D.1, the step size satisfies the inequality

$$\gamma \leq \frac{1}{L + \sqrt{\left(L_A^2\omega_P + L_B^2\theta\right)\left(\frac{1}{p} - 1\right)}}.$$

Since by Lemma A.5, when the compressors are independent we have $\theta = \omega_P/n$, the algorithm converges in

$$\bar{T} = \frac{\Psi^0}{\varepsilon}\left(L + \sqrt{\left(L_A^2\omega_P + L_B^2\frac{\omega_P}{n}\right)\left(\frac{1}{p} - 1\right)}\right)$$
$$= \mathcal{O}\left(\frac{\Psi^0}{\varepsilon}\left(L + \sqrt{\frac{\omega_P}{p}\left(L_A^2 + \frac{L_B^2}{n}\right)}\right)\right) \tag{22}$$

iterations. $\square$

**Theorem D.2.** *Let Assumptions 1.1, 1.2 and 4.2 be satisfied. Choose*

$$\gamma = \begin{cases} \left(L + \sqrt{(L_A^2 + L_B^2)\,\omega_P/p}\right)^{-1} & \text{for SameRandK } \textit{compressors} \\ \left(L + \sqrt{\left(L_A^2 + L_B^2/n\right)\omega_P/p}\right)^{-1} & \text{for independent RandK } \textit{compressors} \\ \left(L + L_A\sqrt{\omega_P/p}\right)^{-1} & \text{for PermK } \textit{compressors} \end{cases}$$

*and set $w_i^0 = x^0$ for all $i \in [n]$. Then* MARINA-P *finds an $\varepsilon$–stationary point after*

$$\bar{T} = \begin{cases} \mathcal{O}\left( \dfrac{\delta^0 \left( L + \sqrt{\left( L_A^2 + L_B^2 \right) \omega_P / p} \right)}{\varepsilon} \right) & \textit{for } \mathsf{SameRandK} \textit{ compressors} \\[3ex] \mathcal{O}\left( \dfrac{\delta^0 \left( L + \sqrt{\left( L_A^2 + L_B^2 / n \right) \omega_P / p} \right)}{\varepsilon} \right) & \textit{for independent } \mathsf{RandK} \textit{ compressors} \\[3ex] \mathcal{O}\left( \dfrac{\delta^0 \left( L + L_A \sqrt{\omega_P / p} \right)}{\varepsilon} \right) & \textit{for } \mathsf{PermK} \textit{ compressors} \end{cases}$$

*iterations.*

*Proof.* In view of Theorem D.1, the step size is such that

$$\gamma \leq \frac{1}{L + \sqrt{\left( L_A^2 \omega_P + L_B^2 \theta \right) \left( \frac{1}{p} - 1 \right)}}.$$

We now use Lemma A.6 and substitute the vales of $\theta$ specific to each compression type.

For SameRand$K$, we have $\theta = \omega_P$, so the algorithm converges after

$$\bar{T} = \frac{\Psi^0}{\varepsilon} \left( L + \sqrt{\left( L_A^2 \omega_P + L_B^2 \omega_P \right) \left( \frac{1}{p} - 1 \right)} \right) = \mathcal{O}\left( \frac{\Psi^0}{\varepsilon} \left( L + \sqrt{\frac{\omega_P}{p} \left( L_A^2 + L_B^2 \right)} \right) \right) \quad (23)$$

iterations. Following the same reasoning as in the proof of Theorem D.8, for Rand$K$ we have

$$\bar{T} = \frac{\Psi^0}{\varepsilon} \left( L + \sqrt{\left( L_A^2 \omega_P + L_B^2 \frac{\omega_P}{n} \right) \left( \frac{1}{p} - 1 \right)} \right) = \mathcal{O}\left( \frac{\Psi^0}{\varepsilon} \left( L + \sqrt{\frac{\omega_P}{p} \left( L_A^2 + \frac{L_B^2}{n} \right)} \right) \right). \tag{24}$$

Finally, for Perm$K$ we have $\theta = 0$, so

$$\bar{T} = \frac{\Psi^0}{\varepsilon} \left( L + \sqrt{L_A^2 \omega_P \left( \frac{1}{p} - 1 \right)} \right) = \mathcal{O}\left( \frac{\Psi^0}{\varepsilon} \left( L + L_A \sqrt{\frac{\omega_P}{p}} \right) \right). \tag{25}$$

The result follows from the fact that $w_i^0 = x^0$ for all $i \in [n]$. $\qquad \square$

**Corollary D.4.** *Let us take $p = 1/n$ and set $K = d/n$ (corresponding to the sparsification level of a* Perm$K$ *compressor). Then, in the view of Theorem D.2, the average s2w communication complexity of* MARINA-P *is*

$$\begin{cases} \mathcal{O}\left( \dfrac{d \delta^0 L}{n \varepsilon} + \dfrac{d \delta^0}{\varepsilon} \sqrt{L_A^2 + L_B^2} \right) & \textit{for } \mathsf{SameRandK} \textit{ compressors} \\[3ex] \mathcal{O}\left( \dfrac{d \delta^0 L}{n \varepsilon} + \dfrac{d \delta^0}{\varepsilon} \sqrt{L_A^2 + \dfrac{L_B^2}{n}} \right) & \textit{for independent } \mathsf{RandK} \textit{ compressors} \\[3ex] \mathcal{O}\left( \dfrac{d \delta^0 L}{n \varepsilon} + \dfrac{d \delta^0}{\varepsilon} L_A \right) & \textit{for } \mathsf{PermK} \textit{ compressors} \end{cases} \tag{19}$$

*Proof.* The expected number of floats a server is relaying to each client at each iteration of MARINA-P is

$$pd + (1-p)k = \frac{d}{n} + \frac{n-1}{n} k \leq \frac{2d}{n}.$$

Next, using the results from Lemma A.6, our choice of compressors and parameters gives $\omega_P = d/K - 1 = n - 1$ in each of the three cases. Hence, substituting $p = 1/n$ in (23), (24) and (25), we obtain the following server-to-worker communication complexities:

1. for SameRand$K$ compressors:

$$\frac{d}{n} \times \frac{\delta^0}{\varepsilon} \left( L + \sqrt{\omega_P \left( L_A^2 + L_B^2 \right) \left( \frac{1}{p} - 1 \right)} \right) = \frac{\delta^0}{\varepsilon} \left( \frac{d}{n} L + \frac{d}{n} \sqrt{\left( L_A^2 + L_B^2 \right) (n-1)^2} \right)$$
$$= \mathcal{O} \left( \frac{\delta^0}{\varepsilon} \left( \frac{d}{n} L + d \sqrt{L_A^2 + L_B^2} \right) \right),$$

2. for Rand$K$ compressors:

$$\frac{d}{n} \times \frac{\delta^0}{\varepsilon} \left( L + \sqrt{\omega_P \left( L_A^2 + \frac{L_B^2}{n} \right) \left( \frac{1}{p} - 1 \right)} \right) = \frac{\delta^0}{\varepsilon} \left( \frac{d}{n} L + \frac{d}{n} \sqrt{\left( L_A^2 + \frac{L_B^2}{n} \right) (n-1)^2} \right)$$
$$= \mathcal{O} \left( \frac{\delta^0}{\varepsilon} \left( \frac{d}{n} L + d \sqrt{L_A^2 + \frac{L_B^2}{n}} \right) \right),$$

3. for Perm$K$ compressors:

$$\frac{d}{n} \times \frac{\delta^0}{\varepsilon} \left( L + \sqrt{L_A^2 \omega_P \left( \frac{1}{p} - 1 \right)} \right) = \frac{\delta^0}{\varepsilon} \left( \frac{d}{n} L + \frac{d}{n} L_A \sqrt{(n-1)^2} \right)$$
$$= \mathcal{O} \left( \frac{\delta^0}{\varepsilon} \left( \frac{d}{n} L + d L_A \right) \right).$$

$\square$

## D.3 Polyak-Łojasiewicz condition

### D.3.1 Main Results

To complete the theory, we now establish a convergence result for MARINA-P under the Polyak-Łojasiewicz assumption.

**Assumption D.9** (Polyak-Łojasiewicz condition). The function $f$ satisfies Polyak-Łojasiewicz (PŁ) condition with parameter $\mu$, i.e., for all $x \in \mathbb{R}^d$ there exists $x^* \in \arg\min_{x \in \mathbb{R}^d} f(x)$ such that

$$2\mu \left( f(x) - f(x^*) \right) \leq \| \nabla f(x) \|^2. \tag{26}$$

**Theorem D.10.** *Let Assumptions 1.1, 1.2, 4.2 and D.9 be satisfied and suppose that $\{\mathcal{C}_i^t\}_{i=1}^n \in \mathbb{P}(\theta)$ and $\mathcal{C}_i^t \in \mathbb{U}(\omega_P)$ for all $i \in [n]$. Take*

$$0 < \gamma \leq \min \left\{ \frac{1}{L + \sqrt{2 \left( L_A^2 \omega_P + L_B^2 \theta \right) \left( \frac{1}{p} - 1 \right)}}, \frac{p}{2\mu} \right\}. \tag{27}$$

*Letting*

$$\Psi^t = f(x^t) - f^* + \frac{\gamma L_A^2}{p} \frac{1}{n} \sum_{i=1}^n \mathbb{E} \left[ \| w_i^t - x^t \|^2 \right] + \frac{\gamma L_B^2}{p} \mathbb{E} \left[ \left\| \frac{1}{n} \sum_{i=1}^n w_i^t - x^t \right\|^2 \right], \tag{28}$$

*for each $T \geq 1$ we have*

$$\mathbb{E} \left[ \Psi^T \right] \leq (1 - \gamma\mu)^T \Psi^0.$$

**Corollary D.11.** *Let $\mathcal{C}_i^t \in \mathbb{P}(0)$ for all $i \in [n]$ (e.g. PermK), choose $p = 1/(\omega_P + 1)$. Then, in the view of Theorem D.10, Algorithm 1 ensures that $\mathbb{E} \left[ f(x^T) - f^* \right] \leq \varepsilon$ after*

$$\mathcal{O} \left( \max \left\{ \frac{L + L_A \omega_P}{\mu}, \omega_P + 1 \right\} \log \frac{\Psi^0}{\varepsilon} \right)$$

*iterations.*

**Corollary D.12.** *Let $\mathcal{C}_i^t$ be the PermK compressors ($K = d/n$). Then, in the view of Corollary D.11, the s2w communication complexity of* MARINA-P *with PermK is*

$$\mathcal{O}\left(\left(\frac{dL}{n\mu} + \frac{dL_A}{\mu} + d\right)\log\frac{\Psi^0}{\varepsilon}\right).$$

### D.3.2 Proofs

**Theorem D.10.** *Let Assumptions 1.1, 1.2, 4.2 and D.9 be satisfied and suppose that $\{\mathcal{C}_i^t\}_{i=1}^n \in \mathbb{P}(\theta)$ and $\mathcal{C}_i^t \in \mathbb{U}(\omega_P)$ for all $i \in [n]$. Take*

$$0 < \gamma \le \min\left\{\frac{1}{L + \sqrt{2\left(L_A^2\omega_P + L_B^2\theta\right)\left(\frac{1}{p} - 1\right)}}, \frac{p}{2\mu}\right\}. \tag{27}$$

*Letting*

$$\Psi^t = f(x^t) - f^* + \frac{\gamma L_A^2}{p}\frac{1}{n}\sum_{i=1}^n \mathbb{E}\left[\left\|w_i^t - x^t\right\|^2\right] + \frac{\gamma L_B^2}{p}\mathbb{E}\left[\left\|\frac{1}{n}\sum_{i=1}^n w_i^t - x^t\right\|^2\right], \tag{28}$$

*for each $T \ge 1$ we have*

$$\mathbb{E}\left[\Psi^T\right] \le (1 - \gamma\mu)^T \Psi^0.$$

*Proof.* We proceed similarly as in the proof of Theorem D.1. Combining the inequalities in Lemmas D.6 and D.7 gives

$$\frac{\gamma L_A^2}{p}\frac{1}{n}\sum_{i=1}^n \mathbb{E}\left[\left\|w_i^{t+1} - x^{t+1}\right\|^2\right] + \frac{\gamma L_B^2}{p}\mathbb{E}\left[\left\|w^{t+1} - x^{t+1}\right\|^2\right]$$

$$\le \frac{\gamma L_A^2}{p}(1-p)\frac{1}{n}\sum_{i=1}^n \mathbb{E}\left[\left\|w_i^t - x^t\right\|^2\right] + \frac{\gamma L_A^2}{p}(1-p)\omega_P\mathbb{E}\left[\left\|x^{t+1} - x^t\right\|^2\right]$$

$$+ \frac{\gamma L_B^2}{p}(1-p)\mathbb{E}\left[\left\|w^t - x^t\right\|^2\right] + \frac{\gamma L_B^2}{p}(1-p)\theta\mathbb{E}\left[\left\|x^{t+1} - x^t\right\|^2\right]$$

$$= \frac{\gamma L_A^2}{p}(1-p)\frac{1}{n}\sum_{i=1}^n \mathbb{E}\left[\left\|w_i^t - x^t\right\|^2\right] + \frac{\gamma L_B^2}{p}(1-p)\mathbb{E}\left[\left\|w^t - x^t\right\|^2\right]$$

$$+ \frac{\gamma}{p}\left(L_A^2\omega_P + L_B^2\theta\right)(1-p)\mathbb{E}\left[\left\|x^{t+1} - x^t\right\|^2\right]. \tag{29}$$

By adding inequalities (21) and (29), we get

$$\mathbb{E}\left[\Psi^{t+1}\right] \quad = \quad \mathbb{E}\left[\delta^{t+1}\right] + \frac{\gamma L_A^2}{p}\frac{1}{n}\sum_{i=1}^n \mathbb{E}\left[\left\|w_i^{t+1} - x^{t+1}\right\|^2\right] + \frac{\gamma L_B^2}{p}\mathbb{E}\left[\left\|w^{t+1} - x^{t+1}\right\|^2\right]$$

$$\le \quad \mathbb{E}\left[\delta^t\right] - \frac{\gamma}{2}\mathbb{E}\left[\left\|\nabla f(x^t)\right\|^2\right] - \left(\frac{1}{2\gamma} - \frac{L}{2}\right)\mathbb{E}\left[\left\|x^{t+1} - x^t\right\|^2\right]$$

$$+ \frac{\gamma}{2}\left(L_A^2\frac{1}{n}\sum_{i=1}^n \mathbb{E}\left[\left\|w_i^t - x^t\right\|^2\right] + L_B^2\mathbb{E}\left[\left\|w^t - x^t\right\|^2\right]\right)$$

$$+ \frac{\gamma L_A^2}{p}(1-p)\frac{1}{n}\sum_{i=1}^n \mathbb{E}\left[\left\|w_i^t - x^t\right\|^2\right] + \frac{\gamma L_B^2}{p}(1-p)\mathbb{E}\left[\left\|w^t - x^t\right\|^2\right]$$

$$+ \frac{\gamma}{p}\left(L_A^2\omega_P + L_B^2\theta\right)(1-p)\mathbb{E}\left[\left\|x^{t+1} - x^t\right\|^2\right]$$

$$= \quad \mathbb{E}\left[\delta^t\right] - \frac{\gamma}{2}\mathbb{E}\left[\left\|\nabla f(x^t)\right\|^2\right]$$

$$-\left(\frac{1}{2\gamma} - \frac{L}{2} - \frac{\gamma}{p}\left(L_A^2\omega_P + L_B^2\theta\right)(1-p)\right)\mathbb{E}\left[\left\|x^{t+1} - x^t\right\|^2\right]$$

$$+\gamma L_A^2\left(\frac{1}{p} - \frac{1}{2}\right)\frac{1}{n}\sum_{i=1}^n \mathbb{E}\left[\left\|w_i^t - x^t\right\|^2\right] + \gamma L_B^2\left(\frac{1}{p} - \frac{1}{2}\right)\mathbb{E}\left[\left\|w^t - x^t\right\|^2\right]$$

$$\overset{\text{Ass.}D.9,(27)}{\leq} (1-\gamma\mu)\mathbb{E}\left[\delta^t\right] + \frac{\gamma L_A^2}{p}(1-\gamma\mu)\frac{1}{n}\sum_{i=1}^n \mathbb{E}\left[\left\|w_i^t - x^t\right\|^2\right]$$

$$+\frac{\gamma L_B^2}{p}(1-\gamma\mu)\mathbb{E}\left[\left\|w^t - x^t\right\|^2\right]$$

$$= (1-\gamma\mu)\mathbb{E}\left[\Psi^t\right],$$

where the last inequality follows from the Polyak-Łojasiewicz condition, Lemma H.2 and our choice of $\gamma$. Applying the above inequality iteratively, we finish the proof. $\square$

**Corollary D.11.** *Let $\mathcal{C}_i^t \in \mathbb{P}(0)$ for all $i \in [n]$ (e.g. PermK), choose $p = 1/(\omega_P + 1)$. Then, in the view of Theorem D.10, Algorithm 1 ensures that $\mathbb{E}\left[f(x^T) - f^*\right] \leq \varepsilon$ after*

$$\mathcal{O}\left(\max\left\{\frac{L + L_A\omega_P}{\mu}, \omega_P + 1\right\}\log\frac{\Psi^0}{\varepsilon}\right)$$

*iterations.*

*Proof.* In view of Theorem D.10, the step size satisfies

$$\gamma \leq \min\left\{\frac{1}{L + \sqrt{2\left(L_A^2\omega_P + L_B^2\theta\right)\left(\frac{1}{p} - 1\right)}}, \frac{p}{2\mu}\right\}.$$

Therefore, since $\theta = 0$ and $p = 1/(\omega_P + 1)$, the algorithm converges after

$$\bar{T} = \max\left\{\frac{L + \sqrt{2\left(L_A^2\omega_P + L_B^2\theta\right)\left(\frac{1}{p} - 1\right)}}{\mu}, \frac{2}{p}\right\}\log\frac{\Psi^0}{\varepsilon}$$

$$= \mathcal{O}\left(\max\left\{\frac{L + L_A\omega_P}{\mu}, \omega_P + 1\right\}\log\frac{\Psi^0}{\varepsilon}\right)$$

iterations. $\square$

**Corollary D.12.** *Let $\mathcal{C}_i^t$ be the PermK compressors ($K = {}^d/n$). Then, in the view of Corollary D.11, the s2w communication complexity of* MARINA-P *with PermK is*

$$\mathcal{O}\left(\left(\frac{dL}{n\mu} + \frac{dL_A}{\mu} + d\right)\log\frac{\Psi^0}{\varepsilon}\right).$$

*Proof.* For PermK, $\omega_P = n - 1$. Therefore, the iteration complexity is

$$\mathcal{O}\left(\max\left\{\frac{L + L_A n}{\mu}, n\right\}\log\frac{\Psi^0}{\varepsilon}\right).$$

Since the expected number of floats the server is relaying to each client is

$$pd + (1-p)k = \frac{d}{n} + \frac{n-1}{n}k \leq \frac{2d}{n},$$

the server-to-worker communication complexity is

$$\mathcal{O}\left(\max\left\{\frac{\frac{d}{n}L + dL_A}{\mu}, d\right\}\log\frac{\Psi^0}{\varepsilon}\right).$$

$\square$

# E Convergence of M3 in the General Case

We now move on to the bidirectionally compressed method. Below is a generalization of Theorem 5.1 to all unbiased compressors.

## E.1 Main Results

**Theorem E.1.** *Let Assumptions 1.1, 1.2, 1.5 and 4.2 hold and suppose that the compressors $\mathcal{Q}_i^t \in \mathbb{U}(\omega_D)$ satisfy Assumption 1.6, $\{\mathcal{C}_i^t\}_{i=1}^n \in \mathbb{P}(\theta)$ and $\mathcal{C}_i^t \in \mathbb{U}(\omega_P)$ for all $i \in [n]$. Let $\gamma > 0$ be such that*

$$\gamma \leq \left( L + \sqrt{288 \left( \left( \frac{\theta}{p_P} + \frac{1 + \theta p_P}{\beta^2} \right) L_B^2 + \left( \frac{\omega_P}{p_P} + \frac{1 + \omega_P p_P}{\beta^2} \right) L_A^2 + \left( \frac{\omega_D \omega_P \beta}{n p_D} + \frac{\omega_D (1 + \omega_P p_P)}{n p_D} \right) L_{\max}^2 \right)} \right)^{-1}.$$

*Letting*

$$\Psi^t = \delta^t + \kappa \left\| g^t - \frac{1}{n} \sum_{i=1}^n \nabla f_i(z_i^t) \right\|^2 + \eta \left\| z^t - w^t \right\|^2 + \nu \frac{1}{n} \sum_{i=1}^n \left\| z_i^t - w_i^t \right\|^2$$

$$+ \rho \left\| w^t - x^t \right\|^2 + \mu \frac{1}{n} \sum_{i=1}^n \left\| w_i^t - x^t \right\|^2,$$

*where $\kappa = \frac{\gamma}{p_D}$, $\eta = \frac{4 \gamma L_B^2}{\beta}$, $\nu = \frac{4 \gamma L_A^2}{\beta} + \frac{6 \gamma \omega_D \beta L_{\max}^2}{n p_D}$, $\rho = 32 \gamma L_B^2 \left( \frac{1}{p_P} + \frac{p_P}{\beta^2} \right)$ and $\mu = 32 \gamma L_A^2 \left( \frac{1}{p_P} + \frac{p_P}{\beta^2} \right) + \frac{48 \gamma \omega_D L_{\max}^2}{n p_D} (\beta + p_P)$, M3 ensures that*

$$\frac{1}{T} \sum_{t=0}^{T-1} \mathbb{E} \left[ \left\| \nabla f(x^t) \right\|^2 \right] = \mathcal{O} \left( \frac{\Psi^0}{\gamma T} \right).$$

We now simplify the above result by considering $\theta = 0$.

**Corollary E.2.** *Let $\mathcal{C}_i^t \in \mathbb{P}(0)$ for all $i \in [n]$ (e.g. PermK), choose $p_P = 1/(\omega_P + 1)$, $p_D = 1/(\omega_D + 1)$ and*

$$\beta = \min \left\{ \left( \frac{n}{\omega_D \omega_P (\omega_D + 1)} \right)^{1/3}, 1 \right\}.$$

*Then, in the view of Theorem E.1, the iteration complexity is*

$$\mathcal{O} \left( \frac{\Psi^0}{\varepsilon} \left( L_{\max} + \left( \frac{\omega_D \omega_P (\omega_D + 1)}{n} \right)^{1/3} L_{\max} + \sqrt{\frac{\omega_D (\omega_D + 1)}{n}} L_{\max} + \sqrt{\omega_P (\omega_P + 1)} L_A \right) \right).$$

We now give the bound for the total communication complexity of M3.

**Corollary E.3.** *Let $\mathcal{C}_i^t$ be the PermK compressors and $\mathcal{Q}_i^t$ be the independent (Assumption 1.6) RandK compressors, both with $K = d/n$. Then, in the view of Corollary E.2, the iteration complexity is*

$$\mathcal{O} \left( \frac{\Psi^0}{\varepsilon} \left( n^{2/3} L_{\max} + n L_A \right) \right),$$

*and the total communication complexity is*

$$\mathcal{O} \left( \frac{\Psi^0}{\varepsilon} \left( \frac{d L_{\max}}{n^{1/3}} + d L_A \right) \right).$$

*Remark* E.4. The above result proves the complexities from Theorem 5.1.

**Algorithm 2** M3

1: **Input:** initial model $x_0 \in \mathbb{R}^d$ (stored on the server), initial model shifts $w_i^0 = z_i^0 = x^0$,
   $i \in [n]$ (stored on the workers), initial gradient estimators $g^0 = \nabla f(x^0)$ (stored on the sever),
   step size $\gamma > 0$, probabilities $0 < p_P, p_D \le 1$, compressors $\mathcal{C}_1^t, \dots, \mathcal{C}_n^t \in \mathbb{U}(\omega_P) \cap \mathbb{P}(\theta)$,
   $\mathcal{Q}_1^t, \dots, \mathcal{Q}_n^t \in \mathbb{U}(\omega_D)$ for all $t \ge 0$.
2: **for** $t = 0, \dots, T$ **do**
3:     $x^{t+1} = x^t - \gamma g^t$                    Server takes a gradient-type step to update the global model
4:     Sample $c_P^t \sim \text{Bernoulli}(p_P), c_D^t \sim \text{Bernoulli}(p_D)$
5:     For $i \in [n]$, send $\mathcal{C}_i^t(x^{t+1} - x^t)$ to worker $i$ if $c_P^t = 0$ and $x^{t+1}$ otherwise
6:     **for** $i = 1, \dots, n$ in parallel **do**
7:        $w_i^{t+1} = \begin{cases} x^{t+1} & \text{if } c_P^t = 1, \\ w_i^t + \mathcal{C}_i^t(x^{t+1} - x^t) & \text{if } c_P^t = 0, \end{cases}$     Worker $i$ updates its local model shift
8:        $z_i^{t+1} = \beta w_i^{t+1} + (1 - \beta) z_i^t$                    Worker $i$ takes the momentum step
9:        Send $\mathcal{Q}_i^t(\nabla f_i(z_i^{t+1}) - \nabla f_i(z_i^t))$ to the server if $c_D^t = 0$ and $\nabla f_i(z_i^{t+1})$ otherwise
10:    **end for**
11:    **if** $c_D^t = 1$ **then**
12:       $g^{t+1} = \frac{1}{n} \sum_{i=1}^n f_i(z_i^{t+1})$
13:    **else**
14:       $g^{t+1} = g^t + \frac{1}{n} \sum_{i=1}^n \mathcal{Q}_i^t(\nabla f_i(z_i^{t+1}) - \nabla f_i(z_i^t))$
15:    **end if**
16: **end for**

(maintaining only the sequence $g^t$ in the implementation is sufficient; the sequences $g_i^t$ from (14) are *virtual*)

## E.2 Proofs

Similar to our approach from the previous section, we start by establishing several inequalities satisfied by the sequences $\{w_1^t, \dots, w_n^t\}_{t \ge 0}$, $\{z_1^t, \dots, z_n^t\}_{t \ge 0}$ and $\{g_1^t, \dots, g_n^t\}_{t \ge 0}$.

**Lemma E.5.** *Let $\mathcal{C}_i^t \in \mathbb{U}(\omega_P)$ for all $i \in [n]$ and $\{\mathcal{C}_i^t\}_{i=1}^n \in \mathbb{P}(\theta)$. Then*

$$\mathbb{E}_t \left[ \left\| w_i^{t+1} - z_i^t \right\|^2 \right] \le \left\| (x^{t+1} - x^t) - p_P(w_i^t - x^t) + (w_i^t - z_i^t) \right\|^2$$
$$+ p_P \left\| w_i^t - x^t \right\|^2 + \omega_P \left\| x^{t+1} - x^t \right\|^2$$

*for all $i \in [n]$, and*

$$\mathbb{E}_t \left[ \left\| w^{t+1} - z^t \right\|^2 \right] \le \left\| (x^{t+1} - x^t) - p_P(w^t - x^t) + (w^t - z^t) \right\|^2$$
$$+ p_P \left\| w^t - x^t \right\|^2 + \theta \left\| x^{t+1} - x^t \right\|^2.$$

*Proof.* Using the definition of $w_i^{t+1}$, we have

$$\mathbb{E}_t \left[ w_i^{t+1} \right] = x^{t+1} + (1 - p_P)(w_i^t - x^t)$$

and hence

$$\mathbb{E}_t \left[ \left\| w_i^{t+1} - z_i^t \right\|^2 \right] \stackrel{(48)}{=} \left\| x^{t+1} + (1 - p_P)(w_i^t - x^t) - z_i^t \right\|^2$$
$$+ \mathbb{E}_t \left[ \left\| w_i^{t+1} - (x^{t+1} + (1 - p_P)(w_i^t - x^t)) \right\|^2 \right]$$
$$= \left\| (x^{t+1} - x^t) - p_P(w_i^t - x^t) + (w_i^t - z_i^t) \right\|^2$$
$$+ \mathbb{E}_t \left[ \left\| w_i^{t+1} - (x^{t+1} + (1 - p_P)(w_i^t - x^t)) \right\|^2 \right].$$

Using the definition of $w_i^{t+1}$ again, we get

$$\mathbb{E}_t \left[ \left\| w_i^{t+1} - z_i^t \right\|^2 \right]$$

$$= \left\| (x^{t+1} - x^t) - p_P(w_i^t - x^t) + (w_i^t - z_i^t) \right\|^2$$
$$+ p_P \left\| x^{t+1} - (x^{t+1} + (1 - p_P)(w_i^t - x^t)) \right\|^2$$
$$+ (1 - p_P)\mathbb{E}_t \left[ \left\| w_i^t + \mathcal{C}_i^t(x^{t+1} - x^t) - (x^{t+1} + (1 - p_P)(w_i^t - x^t)) \right\|^2 \right]$$
$$= \left\| (x^{t+1} - x^t) - p_P(w_i^t - x^t) + (w_i^t - z_i^t) \right\|^2 + p_P(1 - p_P)^2 \left\| w_i^t - x^t \right\|^2$$
$$+ (1 - p_P)\mathbb{E}_t \left[ \left\| \mathcal{C}_i^t(x^{t+1} - x^t) - (x^{t+1} - x^t) + p_P(w_i^t - x^t) \right\|^2 \right]$$
$$\overset{(48)}{=} \left\| (x^{t+1} - x^t) - p_P(w_i^t - x^t) + (w_i^t - z_i^t) \right\|^2 + p_P(1 - p_P)^2 \left\| w_i^t - x^t \right\|^2$$
$$+ (1 - p_P)p_P^2 \left\| w_i^t - x^t \right\|^2 + (1 - p_P)\mathbb{E}_t \left[ \left\| \mathcal{C}_i^t(x^{t+1} - x^t) - (x^{t+1} - x^t) \right\|^2 \right]$$
$$\overset{\text{Def.}1.4}{\leq} \left\| (x^{t+1} - x^t) - p_P(w_i^t - x^t) + (w_i^t - z_i^t) \right\|^2 + p_P \left\| w_i^t - x^t \right\|^2 + \omega_P \left\| x^{t+1} - x^t \right\|^2 .$$

Using the same reasoning, we now prove the second inequality:

$$\mathbb{E}_t \left[ \left\| w^{t+1} - z^t \right\|^2 \right]$$
$$\overset{(48)}{=} \left\| x^{t+1} + (1 - p_P)(w^t - x^t) - z^t \right\|^2 + \mathbb{E}_t \left[ \left\| w^{t+1} - (x^{t+1} + (1 - p_P)(w^t - x^t)) \right\|^2 \right]$$
$$= \left\| (x^{t+1} - x^t) - p_P(w^t - x^t) + (w^t - z^t) \right\|^2$$
$$+ \mathbb{E}_t \left[ \left\| w^{t+1} - (x^{t+1} + (1 - p_P)(w^t - x^t)) \right\|^2 \right]$$
$$= \left\| (x^{t+1} - x^t) - p_P(w^t - x^t) + (w^t - z^t) \right\|^2 + p_P(1 - p_P)^2 \left\| w^t - x^t \right\|^2$$
$$+ (1 - p_P)\mathbb{E}_t \left[ \left\| w^t + \frac{1}{n}\sum_{i=1}^n \mathcal{C}_i^t(x^{t+1} - x^t) - (x^{t+1} + (1 - p_P)(w^t - x^t)) \right\|^2 \right]$$
$$= \left\| (x^{t+1} - x^t) - p_P(w^t - x^t) + (w^t - z^t) \right\|^2 + p_P(1 - p_P)^2 \left\| w^t - x^t \right\|^2$$
$$+ (1 - p_P)\mathbb{E}_t \left[ \left\| \frac{1}{n}\sum_{i=1}^n \mathcal{C}_i^t(x^{t+1} - x^t) - (x^{t+1} - x^t) + p_P(w^t - x^t) \right\|^2 \right]$$
$$\overset{(48)}{=} \left\| (x^{t+1} - x^t) - p_P(w^t - x^t) + (w^t - z^t) \right\|^2 + p_P(1 - p_P)^2 \left\| w^t - x^t \right\|^2$$
$$+ (1 - p_P)\mathbb{E}_t \left[ \left\| \frac{1}{n}\sum_{i=1}^n \mathcal{C}_i^t(x^{t+1} - x^t) - (x^{t+1} - x^t) \right\|^2 \right] + p_P^2(1 - p_P) \left\| w^t - x^t \right\|^2$$
$$\overset{\text{Def.}A.3}{\leq} \left\| (x^{t+1} - x^t) - p_P(w^t - x^t) + (w^t - z^t) \right\|^2 + p_P \left\| w^t - x^t \right\|^2 + \theta \left\| x^{t+1} - x^t \right\|^2 .$$

$\square$

**Lemma E.6.** *Let Assumption 1.5 hold. Furthermore, suppose that the compressors $\mathcal{Q}_i^t \in \mathbb{U}(\omega_D)$ satisfy Assumption 1.6 and that $\mathcal{C}_i^t \in \mathbb{U}(\omega_P)$ for $i \in [n]$. Then*

$$\mathbb{E}\left[ \left\| g^{t+1} - \frac{1}{n}\sum_{i=1}^n \nabla f_i(z_i^{t+1}) \right\|^2 \right]$$
$$\leq \frac{\omega_D L_{\max}^2}{n} \left( 4p_P\beta^2 \mathbb{E}\left[ \frac{1}{n}\sum_{i=1}^n \left\| w_i^t - x^t \right\|^2 \right] + 3\beta^2 \mathbb{E}\left[ \frac{1}{n}\sum_{i=1}^n \left\| z_i^t - w_i^t \right\|^2 \right] + 3(\omega_P + 1)\beta^2 \mathbb{E}\left[ \left\| x^{t+1} - x^t \right\|^2 \right] \right)$$
$$+ (1 - p_D)\mathbb{E}\left[ \left\| g^t - \frac{1}{n}\sum_{i=1}^n \nabla f_i(z_i^t) \right\|^2 \right] .$$

*Proof.* First, from the definition of $g_i^{t+1}$, we get

$$\mathbb{E}\left[\left\|g^{t+1} - \frac{1}{n}\sum_{i=1}^{n}\nabla f_i(z_i^{t+1})\right\|^2\right]$$

$$= (1-p_D)\mathbb{E}\left[\left\|\frac{1}{n}\sum_{i=1}^{n}\left(g_i^t + \mathcal{Q}_i^t(\nabla f_i(z_i^{t+1}) - \nabla f_i(z_i^t)) - \nabla f_i(z_i^{t+1})\right)\right\|^2\right]$$

and hence

$$\mathbb{E}\left[\left\|g^{t+1} - \frac{1}{n}\sum_{i=1}^{n}\nabla f_i(z_i^{t+1})\right\|^2\right]$$

$$\overset{(48)}{=} (1-p_D)\mathbb{E}\left[\left\|\frac{1}{n}\sum_{i=1}^{n}\mathcal{Q}_i^t\left(\nabla f_i(z_i^{t+1}) - \nabla f_i(z_i^t)\right) - \frac{1}{n}\sum_{i=1}^{n}\left(\nabla f_i(z_i^{t+1}) - \nabla f_i(z_i^t)\right)\right\|^2\right]$$

$$+ (1-p_D)\mathbb{E}\left[\left\|g^t - \frac{1}{n}\sum_{i=1}^{n}\nabla f_i(z_i^t)\right\|^2\right]$$

$$\overset{\text{Def.1.4,(1.6)}}{\leq} \frac{\omega_D}{n}\mathbb{E}\left[\frac{1}{n}\sum_{i=1}^{n}\left\|\nabla f_i(z_i^{t+1}) - \nabla f_i(z_i^t)\right\|^2\right] + (1-p_D)\mathbb{E}\left[\left\|g^t - \frac{1}{n}\sum_{i=1}^{n}\nabla f_i(z_i^t)\right\|^2\right]$$

$$\overset{\text{Ass.1.5}}{\leq} \frac{\omega_D L_{\max}^2}{n}\mathbb{E}\left[\frac{1}{n}\sum_{i=1}^{n}\left\|z_i^{t+1} - z_i^t\right\|^2\right] + (1-p_D)\mathbb{E}\left[\left\|g^t - \frac{1}{n}\sum_{i=1}^{n}\nabla f_i(z_i^t)\right\|^2\right]. \tag{30}$$

Let us consider the first term separately:

$$\mathbb{E}\left[\frac{1}{n}\sum_{i=1}^{n}\left\|z_i^{t+1} - z_i^t\right\|^2\right] = \mathbb{E}\left[\frac{1}{n}\sum_{i=1}^{n}\left\|\beta w_i^{t+1} + (1-\beta)z_i^t - z_i^t\right\|^2\right]$$

$$= \beta^2\mathbb{E}\left[\frac{1}{n}\sum_{i=1}^{n}\left\|w_i^{t+1} - z_i^t\right\|^2\right].$$

Using the result from Lemma E.5, we have

$$\mathbb{E}\left[\frac{1}{n}\sum_{i=1}^{n}\left\|z_i^{t+1} - z_i^t\right\|^2\right]$$

$$\leq \beta^2\mathbb{E}\left[\frac{1}{n}\sum_{i=1}^{n}\left\|(x^{t+1} - x^t) - p_P(w_i^t - x^t) + (w_i^t - z_i^t)\right\|^2\right]$$

$$+ \beta^2\mathbb{E}\left[p_P\frac{1}{n}\sum_{i=1}^{n}\left\|w_i^t - x^t\right\|^2 + \omega_P\left\|x^{t+1} - x^t\right\|^2\right]$$

$$\overset{(45)}{\leq} \beta^2\mathbb{E}\left[\frac{3}{n}\sum_{i=1}^{n}\left\|w_i^t - z_i^t\right\|^2 + 4p_P\frac{1}{n}\sum_{i=1}^{n}\left\|w_i^t - x^t\right\|^2 + (\omega_P + 3)\left\|x^{t+1} - x^t\right\|^2\right],$$

where in the last line we use the fact that $p_P \leq 1$. It remains to substitute the above inequality in (30). $\square$

**Lemma E.7.** *Let $\mathcal{C}_i^t \in \mathbb{U}(\omega_P)$ for all $i \in [n]$ and $\{\mathcal{C}_i^t\}_{i=1}^n \in \mathbb{P}(\theta)$. Then*

$$\mathbb{E}\left[\left\|z_i^{t+1} - w_i^{t+1}\right\|^2\right] \leq \left(1 - \frac{\beta}{2}\right)\mathbb{E}\left[\left\|z_i^t - w_i^t\right\|^2\right] + 4\left(\frac{1}{\beta} + \omega_P\right)\mathbb{E}\left[\left\|x^{t+1} - x^t\right\|^2\right]$$

$$+ 4p_P\left(1 + \frac{p_P}{\beta}\right)\mathbb{E}\left[\left\|w_i^t - x^t\right\|^2\right]$$

*for all $i \in [n]$, and*

$$\mathbb{E}\left[\left\|z^{t+1} - w^{t+1}\right\|^2\right] \leq \left(1 - \frac{\beta}{2}\right)\mathbb{E}\left[\left\|z^t - w^t\right\|^2\right] + 4\left(\frac{1}{\beta} + \theta\right)\mathbb{E}\left[\left\|x^{t+1} - x^t\right\|^2\right]$$
$$+ 4p_P\left(1 + \frac{p_P}{\beta}\right)\mathbb{E}\left[\left\|w^t - x^t\right\|^2\right].$$

*Proof.* From the definition of $z_i^{t+1}$, we get

$$\mathbb{E}\left[\left\|z_i^{t+1} - w_i^{t+1}\right\|^2\right] = \mathbb{E}\left[\left\|\beta w_i^{t+1} + (1-\beta)z_i^t - w_i^{t+1}\right\|^2\right] = (1-\beta)^2\mathbb{E}\left[\left\|w_i^{t+1} - z_i^t\right\|^2\right].$$

Then, Lemma E.5 gives

$$\mathbb{E}\left[\left\|z_i^{t+1} - w_i^{t+1}\right\|^2\right]$$
$$\leq (1-\beta)^2\mathbb{E}\left[\left\|(x^{t+1} - x^t) - p_P(w_i^t - x^t) + (w_i^t - z_i^t)\right\|^2\right]$$
$$+ (1-\beta)^2\mathbb{E}\left[p_P\left\|w_i^t - x^t\right\|^2 + \omega_P\left\|x^{t+1} - x^t\right\|^2\right]$$
$$\overset{(44),(46),(47)}{\leq} \left(1 - \frac{\beta}{2}\right)\mathbb{E}\left[\left\|w_i^t - z_i^t\right\|^2\right] + \frac{2}{\beta}\mathbb{E}\left[\left\|(x^{t+1} - x^t) - p_P(w_i^t - x^t)\right\|^2\right]$$
$$+ \mathbb{E}\left[p_P\left\|w_i^t - x^t\right\|^2 + \omega_P\left\|x^{t+1} - x^t\right\|^2\right]$$
$$\overset{(45)}{\leq} \left(1 - \frac{\beta}{2}\right)\mathbb{E}\left[\left\|w_i^t - z_i^t\right\|^2\right] + \frac{4}{\beta}\mathbb{E}\left[\left\|x^{t+1} - x^t\right\|^2\right] + \frac{4p_P^2}{\beta}\mathbb{E}\left[\left\|w_i^t - x^t\right\|^2\right]$$
$$+ \mathbb{E}\left[p_P\left\|w_i^t - x^t\right\|^2 + \omega_P\left\|x^{t+1} - x^t\right\|^2\right]$$
$$\leq \left(1 - \frac{\beta}{2}\right)\mathbb{E}\left[\left\|w_i^t - z_i^t\right\|^2\right] + 4\left(\frac{1}{\beta} + \omega_P\right)\mathbb{E}\left[\left\|x^{t+1} - x^t\right\|^2\right]$$
$$+ 4p_P\left(1 + \frac{p_P}{\beta}\right)\mathbb{E}\left[\left\|w_i^t - x^t\right\|^2\right].$$

The second inequality is proved almost in the same way. First,

$$\mathbb{E}\left[\left\|z^{t+1} - w^{t+1}\right\|^2\right] = (1-\beta)^2\mathbb{E}\left[\left\|w^{t+1} - z^t\right\|^2\right],$$

and using Lemma E.5, we obtain

$$\mathbb{E}\left[\left\|z^{t+1} - w^{t+1}\right\|^2\right]$$
$$\leq (1-\beta)^2\mathbb{E}\left[\left\|(x^{t+1} - x^t) - p_P(w^t - x^t) + (w^t - z^t)\right\|^2 + p_P\left\|w^t - x^t\right\|^2 + \theta\left\|x^{t+1} - x^t\right\|^2\right]$$
$$\overset{(44)}{\leq} \left(1 - \frac{\beta}{2}\right)\mathbb{E}\left[\left\|w^t - z^t\right\|^2\right] + \frac{2}{\beta}\mathbb{E}\left[\left\|(x^{t+1} - x^t) - p_P(w^t - x^t)\right\|^2\right]$$
$$+ \mathbb{E}\left[p_P\left\|w^t - x^t\right\|^2 + \theta\left\|x^{t+1} - x^t\right\|^2\right]$$
$$\overset{(45)}{\leq} \left(1 - \frac{\beta}{2}\right)\mathbb{E}\left[\left\|w^t - z^t\right\|^2\right] + 4\left(\frac{1}{\beta} + \theta\right)\mathbb{E}\left[\left\|x^{t+1} - x^t\right\|^2\right]$$
$$+ \frac{4p_P^2}{\beta}\mathbb{E}\left[\left\|w^t - x^t\right\|^2\right] + \mathbb{E}\left[p_P\left\|w^t - x^t\right\|^2\right]$$
$$\leq \left(1 - \frac{\beta}{2}\right)\mathbb{E}\left[\left\|w^t - z^t\right\|^2\right] + 4\left(\frac{1}{\beta} + \theta\right)\mathbb{E}\left[\left\|x^{t+1} - x^t\right\|^2\right]$$
$$+ 4p_P\left(1 + \frac{p_P}{\beta}\right)\mathbb{E}\left[\left\|w^t - x^t\right\|^2\right].$$

$\square$

**Theorem E.1.** *Let Assumptions 1.1, 1.2, 1.5 and 4.2 hold and suppose that the compressors $\mathcal{Q}_i^t \in \mathbb{U}(\omega_D)$ satisfy Assumption 1.6, $\{\mathcal{C}_i^t\}_{i=1}^n \in \mathbb{P}(\theta)$ and $\mathcal{C}_i^t \in \mathbb{U}(\omega_P)$ for all $i \in [n]$. Let $\gamma > 0$ be such that*

$$\gamma \leq \left( L + \sqrt{288 \left( \left( \tfrac{\theta}{p_P} + \tfrac{1+\theta p_P}{\beta^2} \right) L_B^2 + \left( \tfrac{\omega_P}{p_P} + \tfrac{1+\omega_P p_P}{\beta^2} \right) L_A^2 + \left( \tfrac{\omega_D \omega_P \beta}{n p_D} + \tfrac{\omega_D (1+\omega_P p_P)}{n p_D} \right) L_{\max}^2 \right)} \right)^{-1}.$$

*Letting*

$$\Psi^t = \delta^t + \kappa \left\| g^t - \frac{1}{n} \sum_{i=1}^n \nabla f_i(z_i^t) \right\|^2 + \eta \left\| z^t - w^t \right\|^2 + \nu \frac{1}{n} \sum_{i=1}^n \left\| z_i^t - w_i^t \right\|^2$$

$$+ \rho \left\| w^t - x^t \right\|^2 + \mu \frac{1}{n} \sum_{i=1}^n \left\| w_i^t - x^t \right\|^2,$$

*where* $\kappa = \frac{\gamma}{p_D}$, $\eta = \frac{4\gamma L_B^2}{\beta}$, $\nu = \frac{4\gamma L_A^2}{\beta} + \frac{6\gamma \omega_D \beta L_{\max}^2}{n p_D}$, $\rho = 32\gamma L_B^2 \left( \frac{1}{p_P} + \frac{p_P}{\beta^2} \right)$ *and* $\mu = 32\gamma L_A^2 \left( \frac{1}{p_P} + \frac{p_P}{\beta^2} \right) + \frac{48\gamma \omega_D L_{\max}^2}{n p_D} (\beta + p_P)$, M3 *ensures that*

$$\frac{1}{T} \sum_{t=0}^{T-1} \mathbb{E} \left[ \left\| \nabla f(x^t) \right\|^2 \right] = \mathcal{O} \left( \frac{\Psi^0}{\gamma T} \right).$$

*Proof.* Lemma H.1 gives

$\mathbb{E} \left[ \delta^{t+1} \right]$

$$\leq \mathbb{E} \left[ \delta^t \right] - \frac{\gamma}{2} \mathbb{E} \left[ \left\| \nabla f(x^t) \right\|^2 \right] - \left( \frac{1}{2\gamma} - \frac{L}{2} \right) \mathbb{E} \left[ \left\| x^{t+1} - x^t \right\|^2 \right] + \frac{\gamma}{2} \mathbb{E} \left[ \left\| g^t - \nabla f(x^t) \right\|^2 \right]$$

$$\overset{(44)}{\leq} \mathbb{E} \left[ \delta^t \right] - \frac{\gamma}{2} \mathbb{E} \left[ \left\| \nabla f(x^t) \right\|^2 \right] - \left( \frac{1}{2\gamma} - \frac{L}{2} \right) \mathbb{E} \left[ \left\| x^{t+1} - x^t \right\|^2 \right]$$

$$+ \gamma \mathbb{E} \left[ \left\| g^t - \frac{1}{n} \sum_{i=1}^n \nabla f_i(z_i^t) \right\|^2 \right] + \gamma \mathbb{E} \left[ \left\| \frac{1}{n} \sum_{i=1}^n \left( \nabla f_i(z_i^t) - \nabla f_i(x^t) \right) \right\|^2 \right]$$

$$\overset{(4.2)}{\leq} \mathbb{E} \left[ \delta^t \right] - \frac{\gamma}{2} \mathbb{E} \left[ \left\| \nabla f(x^t) \right\|^2 \right] - \left( \frac{1}{2\gamma} - \frac{L}{2} \right) \mathbb{E} \left[ \left\| x^{t+1} - x^t \right\|^2 \right]$$

$$+ \gamma \mathbb{E} \left[ \left\| g^t - \frac{1}{n} \sum_{i=1}^n \nabla f_i(z_i^t) \right\|^2 \right] + \gamma L_A^2 \frac{1}{n} \sum_{i=1}^n \mathbb{E} \left[ \left\| z_i^t - x^t \right\|^2 \right] + \gamma L_B^2 \mathbb{E} \left[ \left\| z^t - x^t \right\|^2 \right]$$

$$\overset{(44)}{\leq} \mathbb{E} \left[ \delta^t \right] - \frac{\gamma}{2} \mathbb{E} \left[ \left\| \nabla f(x^t) \right\|^2 \right] - \left( \frac{1}{2\gamma} - \frac{L}{2} \right) \mathbb{E} \left[ \left\| x^{t+1} - x^t \right\|^2 \right]$$

$$+ \gamma \mathbb{E} \left[ \left\| g^t - \frac{1}{n} \sum_{i=1}^n \nabla f_i(z_i^t) \right\|^2 \right] + 2\gamma L_A^2 \frac{1}{n} \sum_{i=1}^n \left( \mathbb{E} \left[ \left\| z_i^t - w_i^t \right\|^2 \right] + \mathbb{E} \left[ \left\| w_i^t - x^t \right\|^2 \right] \right)$$

$$+ 2\gamma L_B^2 \left( \mathbb{E} \left[ \left\| z^t - w^t \right\|^2 \right] + \mathbb{E} \left[ \left\| w^t - x^t \right\|^2 \right] \right).$$

Let $\kappa, \eta, \nu, \rho, \mu \geq 0$ be some non-negative numbers that we define later. Using Lemmas D.6, D.7, E.6 and E.7, we get

$$\mathbb{E} \left[ \delta^{t+1} \right] + \kappa \mathbb{E} \left[ \left\| g^{t+1} - \frac{1}{n} \sum_{i=1}^n \nabla f_i(z_i^{t+1}) \right\|^2 \right] + \eta \mathbb{E} \left[ \left\| z^{t+1} - w^{t+1} \right\|^2 \right]$$

$$+ \nu \mathbb{E} \left[ \frac{1}{n} \sum_{i=1}^n \left\| z_i^{t+1} - w_i^{t+1} \right\|^2 \right] + \rho \mathbb{E} \left[ \left\| w^{t+1} - x^{t+1} \right\|^2 \right] + \mu \mathbb{E} \left[ \frac{1}{n} \sum_{i=1}^n \left\| w_i^{t+1} - x^{t+1} \right\|^2 \right]$$

$$\leq \mathbb{E}\left[\delta^t\right] - \frac{\gamma}{2}\mathbb{E}\left[\|\nabla f(x^t)\|^2\right] - \left(\frac{1}{2\gamma} - \frac{L}{2}\right)\mathbb{E}\left[\|x^{t+1} - x^t\|^2\right] + \gamma\mathbb{E}\left[\left\|g^t - \frac{1}{n}\sum_{i=1}^n \nabla f_i(z_i^t)\right\|^2\right]$$

$$+ 2\gamma L_A^2 \frac{1}{n}\sum_{i=1}^n \left(\mathbb{E}\left[\|z_i^t - w_i^t\|^2\right] + \mathbb{E}\left[\|w_i^t - x^t\|^2\right]\right)$$

$$+ 2\gamma L_B^2 \left(\mathbb{E}\left[\|z^t - w^t\|^2\right] + \mathbb{E}\left[\|w^t - x^t\|^2\right]\right)$$

$$+ \kappa\left(\frac{\omega_D L_{\max}^2}{n}\left(4p_P\beta^2\mathbb{E}\left[\frac{1}{n}\sum_{i=1}^n \|w_i^t - x^t\|^2\right] + 3\beta^2\mathbb{E}\left[\frac{1}{n}\sum_{i=1}^n \|z_i^t - w_i^t\|^2\right]\right.\right.$$

$$\left.\left. + 3(\omega_P + 1)\beta^2\mathbb{E}\left[\|x^{t+1} - x^t\|^2\right]\right)\right)$$

$$+ \kappa(1 - p_D)\mathbb{E}\left[\left\|g^t - \frac{1}{n}\sum_{i=1}^n \nabla f_i(z_i^t)\right\|^2\right]$$

$$+ \eta\left(\left(1 - \frac{\beta}{2}\right)\mathbb{E}\left[\|z^t - w^t\|^2\right] + 4\left(\frac{1}{\beta} + \theta\right)\mathbb{E}\left[\|x^{t+1} - x^t\|^2\right]\right.$$

$$\left. + 4p_P\left(1 + \frac{p_P}{\beta}\right)\mathbb{E}\left[\|w^t - x^t\|^2\right]\right)$$

$$+ \nu\left(\left(1 - \frac{\beta}{2}\right)\mathbb{E}\left[\frac{1}{n}\sum_{i=1}^n \|z_i^t - w_i^t\|^2\right] + 4\left(\frac{1}{\beta} + \omega_P\right)\mathbb{E}\left[\|x^{t+1} - x^t\|^2\right]\right.$$

$$\left. + 4p_P\left(1 + \frac{p_P}{\beta}\right)\mathbb{E}\left[\frac{1}{n}\sum_{i=1}^n \|w_i^t - x^t\|^2\right]\right)$$

$$+ \rho(1 - p_P)\left(\mathbb{E}\left[\|w^t - x^t\|^2\right] + \theta\mathbb{E}\left[\|x^{t+1} - x^t\|^2\right]\right)$$

$$+ \mu(1 - p_P)\left(\mathbb{E}\left[\frac{1}{n}\sum_{i=1}^n \|w_i^t - x^t\|^2\right] + \omega_P\mathbb{E}\left[\|x^{t+1} - x^t\|^2\right]\right).$$

Taking $\kappa = \frac{\gamma}{p_D}$ and $\eta = \frac{4\gamma L_B^2}{\beta}$, we get $\gamma + \kappa(1 - p_D) = \kappa$ and $2\gamma L_B^2 + \eta(1 - \beta/2) = \eta$, which gives

$$\mathbb{E}\left[\delta^{t+1}\right] + \kappa\mathbb{E}\left[\left\|g^{t+1} - \frac{1}{n}\sum_{i=1}^n \nabla f_i(z_i^{t+1})\right\|^2\right] + \eta\mathbb{E}\left[\|z^{t+1} - w^{t+1}\|^2\right]$$

$$+ \nu\mathbb{E}\left[\frac{1}{n}\sum_{i=1}^n \|z_i^{t+1} - w_i^{t+1}\|^2\right] + \rho\mathbb{E}\left[\|w^{t+1} - x^{t+1}\|^2\right] + \mu\mathbb{E}\left[\frac{1}{n}\sum_{i=1}^n \|w_i^{t+1} - x^{t+1}\|^2\right]$$

$$\leq \mathbb{E}\left[\delta^t\right] - \frac{\gamma}{2}\mathbb{E}\left[\|\nabla f(x^t)\|^2\right] - \left(\frac{1}{2\gamma} - \frac{L}{2}\right)\mathbb{E}\left[\|x^{t+1} - x^t\|^2\right] + \kappa\mathbb{E}\left[\left\|g^t - \frac{1}{n}\sum_{i=1}^n \nabla f_i(z_i^t)\right\|^2\right]$$

$$+ \eta\mathbb{E}\left[\|z^t - w^t\|^2\right] + 2\gamma L_A^2 \frac{1}{n}\sum_{i=1}^n \left(\mathbb{E}\left[\|z_i^t - w_i^t\|^2\right] + \mathbb{E}\left[\|w_i^t - x^t\|^2\right]\right) + 2\gamma L_B^2\mathbb{E}\left[\|w^t - x^t\|^2\right]$$

$$+ \frac{\gamma}{p_D}\left(\frac{\omega_D L_{\max}^2}{n}\left(4p_P\beta^2\mathbb{E}\left[\frac{1}{n}\sum_{i=1}^n \|w_i^t - x^t\|^2\right] + 3\beta^2\mathbb{E}\left[\frac{1}{n}\sum_{i=1}^n \|z_i^t - w_i^t\|^2\right]\right.\right.$$

$$\left.\left. + 3(\omega_P + 1)\beta^2\mathbb{E}\left[\|x^{t+1} - x^t\|^2\right]\right)\right)$$

$$+ \frac{4\gamma L_B^2}{\beta}\left(4\left(\frac{1}{\beta} + \theta\right)\mathbb{E}\left[\|x^{t+1} - x^t\|^2\right] + 4p_P\left(1 + \frac{p_P}{\beta}\right)\mathbb{E}\left[\|w^t - x^t\|^2\right]\right)$$

$$+ \nu \left( \left( 1 - \frac{\beta}{2} \right) \mathbb{E} \left[ \frac{1}{n} \sum_{i=1}^{n} \| z_i^t - w_i^t \|^2 \right] + 4 \left( \frac{1}{\beta} + \omega_P \right) \mathbb{E} \left[ \| x^{t+1} - x^t \|^2 \right] \right.$$

$$\left. + 4 p_P \left( 1 + \frac{p_P}{\beta} \right) \mathbb{E} \left[ \frac{1}{n} \sum_{i=1}^{n} \| w_i^t - x^t \|^2 \right] \right)$$

$$+ \rho \left( (1 - p_P) \mathbb{E} \left[ \| w^t - x^t \|^2 \right] + \theta \mathbb{E} \left[ \| x^{t+1} - x^t \|^2 \right] \right)$$

$$+ \mu \left( (1 - p_P) \mathbb{E} \left[ \frac{1}{n} \sum_{i=1}^{n} \| w_i^t - x^t \|^2 \right] + \omega_P \mathbb{E} \left[ \| x^{t+1} - x^t \|^2 \right] \right).$$

We rearrange the terms to obtain

$$\mathbb{E} \left[ \delta^{t+1} \right] + \kappa \mathbb{E} \left[ \left\| g^{t+1} - \frac{1}{n} \sum_{i=1}^{n} \nabla f_i(z_i^{t+1}) \right\|^2 \right] + \eta \mathbb{E} \left[ \| z^{t+1} - w^{t+1} \|^2 \right]$$

$$+ \nu \mathbb{E} \left[ \frac{1}{n} \sum_{i=1}^{n} \| z_i^{t+1} - w_i^{t+1} \|^2 \right] + \rho \mathbb{E} \left[ \| w^{t+1} - x^{t+1} \|^2 \right] + \mu \mathbb{E} \left[ \frac{1}{n} \sum_{i=1}^{n} \| w_i^{t+1} - x^{t+1} \|^2 \right]$$

$$\leq \mathbb{E} \left[ \delta^t \right] - \frac{\gamma}{2} \mathbb{E} \left[ \| \nabla f(x^t) \|^2 \right] - \left( \frac{1}{2\gamma} - \frac{L}{2} \right) \mathbb{E} \left[ \| x^{t+1} - x^t \|^2 \right] + \kappa \mathbb{E} \left[ \left\| g^t - \frac{1}{n} \sum_{i=1}^{n} \nabla f_i(z_i^t) \right\|^2 \right]$$

$$+ \eta \mathbb{E} \left[ \| z^t - w^t \|^2 \right] + \left( \nu \left( 1 - \frac{\beta}{2} \right) + 2\gamma L_A^2 + \frac{3\gamma \omega_D \beta^2 L_{\max}^2}{n p_D} \right) \mathbb{E} \left[ \frac{1}{n} \sum_{i=1}^{n} \| z_i^t - w_i^t \|^2 \right]$$

$$+ \left( \frac{3\gamma \omega_D (\omega_P + 1) \beta^2 L_{\max}^2}{n p_D} + \rho \theta + \frac{16 \gamma L_B^2}{\beta} \left( \frac{1}{\beta} + \theta \right) \right.$$

$$\left. + 4\nu \left( \frac{1}{\beta} + \omega_P \right) + \mu \omega_P \right) \mathbb{E} \left[ \| x^{t+1} - x^t \|^2 \right]$$

$$+ \left( \rho(1 - p_P) + 2\gamma L_B^2 + \frac{16 \gamma L_B^2 p_P}{\beta} \left( 1 + \frac{p_P}{\beta} \right) \right) \mathbb{E} \left[ \| w^t - x^t \|^2 \right]$$

$$+ \left( \mu(1 - p_P) + 2\gamma L_A^2 + 4\nu p_P \left( 1 + \frac{p_P}{\beta} \right) + \frac{4\gamma \omega_D p_P \beta^2 L_{\max}^2}{n p_D} \right) \mathbb{E} \left[ \frac{1}{n} \sum_{i=1}^{n} \| w_i^t - x^t \|^2 \right].$$

We now consider the coefficient of the term $\mathbb{E} \left[ \| w^t - x^t \|^2 \right]$. Using the inequality $xy \leq \frac{x^2 + y^2}{2}$ for all $x, y \geq 0$, we get

$$\rho(1 - p_P) + 2\gamma L_B^2 + \frac{16 \gamma L_B^2 p_P}{\beta} \left( 1 + \frac{p_P}{\beta} \right) \leq \rho(1 - p_P) + 16 \gamma L_B^2 \left( 1 + \frac{p_P}{\beta} + \frac{p_P^2}{\beta^2} \right)$$

$$\leq \rho(1 - p_P) + 32 \gamma L_B^2 \left( 1 + \frac{p_P^2}{\beta^2} \right)$$

$$= \rho$$

for $\rho = 32 \gamma L_B^2 \left( \frac{1}{p_P} + \frac{p_P}{\beta^2} \right)$. With this choice of $\rho$, we obtain

$$\mathbb{E} \left[ \delta^{t+1} \right] + \kappa \mathbb{E} \left[ \left\| g^{t+1} - \frac{1}{n} \sum_{i=1}^{n} \nabla f_i(z_i^{t+1}) \right\|^2 \right] + \eta \mathbb{E} \left[ \| z^{t+1} - w^{t+1} \|^2 \right]$$

$$+ \nu \mathbb{E} \left[ \frac{1}{n} \sum_{i=1}^{n} \| z_i^{t+1} - w_i^{t+1} \|^2 \right] + \rho \mathbb{E} \left[ \| w^{t+1} - x^{t+1} \|^2 \right] + \mu \mathbb{E} \left[ \frac{1}{n} \sum_{i=1}^{n} \| w_i^{t+1} - x^{t+1} \|^2 \right]$$

$$\leq \mathbb{E} \left[ \delta^t \right] - \frac{\gamma}{2} \mathbb{E} \left[ \| \nabla f(x^t) \|^2 \right] - \left( \frac{1}{2\gamma} - \frac{L}{2} \right) \mathbb{E} \left[ \| x^{t+1} - x^t \|^2 \right] + \kappa \mathbb{E} \left[ \left\| g^t - \frac{1}{n} \sum_{i=1}^{n} \nabla f_i(z_i^t) \right\|^2 \right]$$

$$+ \eta \mathbb{E}\left[\left\|z^t - w^t\right\|^2\right] + \rho \mathbb{E}\left[\left\|w^t - x^t\right\|^2\right]$$

$$+ \left(\nu\left(1 - \frac{\beta}{2}\right) + 2\gamma L_A^2 + \frac{3\gamma\omega_D\beta^2 L_{\max}^2}{np_D}\right) \mathbb{E}\left[\frac{1}{n}\sum_{i=1}^n \left\|z_i^t - w_i^t\right\|^2\right]$$

$$+ \left(\frac{3\gamma\omega_D(\omega_P + 1)\beta^2 L_{\max}^2}{np_D} + 32\gamma L_B^2\left(\frac{1}{p_P} + \frac{p_P}{\beta^2}\right)\theta + \frac{16\gamma L_B^2}{\beta}\left(\frac{1}{\beta} + \theta\right)\right.$$

$$\left. + 4\nu\left(\frac{1}{\beta} + \omega_P\right) + \mu\omega_P\right) \mathbb{E}\left[\left\|x^{t+1} - x^t\right\|^2\right]$$

$$+ \left(\mu(1 - p_P) + 2\gamma L_A^2 + 4\nu p_P\left(1 + \frac{p_P}{\beta}\right) + \frac{4\gamma\omega_D p_P \beta^2 L_{\max}^2}{np_D}\right) \mathbb{E}\left[\frac{1}{n}\sum_{i=1}^n \left\|w_i^t - x^t\right\|^2\right].$$

Next, taking $\nu = \frac{4\gamma L_A^2}{\beta} + \frac{6\gamma\omega_D\beta L_{\max}^2}{np_D}$ gives

$$\mathbb{E}\left[\delta^{t+1}\right] + \kappa\mathbb{E}\left[\left\|g^{t+1} - \frac{1}{n}\sum_{i=1}^n \nabla f_i(z_i^{t+1})\right\|^2\right] + \eta\mathbb{E}\left[\left\|z^{t+1} - w^{t+1}\right\|^2\right]$$

$$+ \nu\mathbb{E}\left[\frac{1}{n}\sum_{i=1}^n \left\|z_i^{t+1} - w_i^{t+1}\right\|^2\right] + \rho\mathbb{E}\left[\left\|w^{t+1} - x^{t+1}\right\|^2\right] + \mu\mathbb{E}\left[\frac{1}{n}\sum_{i=1}^n \left\|w_i^{t+1} - x^{t+1}\right\|^2\right]$$

$$\leq \mathbb{E}\left[\delta^t\right] - \frac{\gamma}{2}\mathbb{E}\left[\left\|\nabla f(x^t)\right\|^2\right] - \left(\frac{1}{2\gamma} - \frac{L}{2}\right)\mathbb{E}\left[\left\|x^{t+1} - x^t\right\|^2\right] + \kappa\mathbb{E}\left[\left\|g^t - \frac{1}{n}\sum_{i=1}^n \nabla f_i(z_i^t)\right\|^2\right]$$

$$+ \eta\mathbb{E}\left[\left\|z^t - w^t\right\|^2\right] + \rho\mathbb{E}\left[\left\|w^t - x^t\right\|^2\right] + \nu\mathbb{E}\left[\frac{1}{n}\sum_{i=1}^n \left\|z_i^t - w_i^t\right\|^2\right]$$

$$+ \left(\frac{3\gamma\omega_D(\omega_P + 1)\beta^2 L_{\max}^2}{np_D} + 32\gamma L_B^2\left(\frac{1}{p_P} + \frac{p_P}{\beta^2}\right)\theta + \frac{16\gamma L_B^2}{\beta}\left(\frac{1}{\beta} + \theta\right)\right.$$

$$\left. + 4\left(\frac{4\gamma L_A^2}{\beta} + \frac{6\gamma\omega_D\beta L_{\max}^2}{np_D}\right)\left(\frac{1}{\beta} + \omega_P\right) + \mu\omega_P\right)\mathbb{E}\left[\left\|x^{t+1} - x^t\right\|^2\right]$$

$$+ \left(\mu(1 - p_P) + 2\gamma L_A^2 + 4\left(\frac{4\gamma L_A^2}{\beta} + \frac{6\gamma\omega_D\beta L_{\max}^2}{np_D}\right)p_P\left(1 + \frac{p_P}{\beta}\right)\right.$$

$$\left. + \frac{4\gamma\omega_D p_P \beta^2 L_{\max}^2}{np_D}\right)\mathbb{E}\left[\frac{1}{n}\sum_{i=1}^n \left\|w_i^t - x^t\right\|^2\right].$$

Let us consider the last bracket:

$$\mu(1 - p_P) + 2\gamma L_A^2 + 4\left(\frac{4\gamma L_A^2}{\beta} + \frac{6\gamma\omega_D\beta L_{\max}^2}{np_D}\right)p_P\left(1 + \frac{p_P}{\beta}\right) + \frac{4\gamma\omega_D p_P \beta^2 L_{\max}^2}{np_D}$$

$$= \mu(1 - p_P) + 2\gamma L_A^2 + \frac{16\gamma p_P L_A^2}{\beta} + \frac{24\gamma\omega_D p_P \beta L_{\max}^2}{np_D} + \frac{16\gamma p_P^2 L_A^2}{\beta^2}$$

$$+ \frac{24\gamma\omega_D p_P^2 L_{\max}^2}{np_D} + \frac{4\gamma\omega_D p_P \beta^2 L_{\max}^2}{np_D}$$

$$\leq \mu(1 - p_P) + 16\gamma L_A^2\left(1 + \frac{p_P}{\beta} + \frac{p_P^2}{\beta^2}\right) + \frac{24\gamma\omega_D p_P L_{\max}^2}{np_D}\left(\beta + p_P + \beta^2\right)$$

$$\leq \mu(1 - p_P) + 32\gamma L_A^2\left(1 + \frac{p_P^2}{\beta^2}\right) + \frac{48\gamma\omega_D p_P L_{\max}^2}{np_D}\left(\beta + p_P\right)$$

$$= \mu$$

for $\mu = 32\gamma L_A^2 \left(\frac{1}{p_P} + \frac{p_P}{\beta^2}\right) + \frac{48\gamma\omega_D L_{\max}^2}{np_D}(\beta + p_P)$. For this choice, we get

$$
\mathbb{E}\left[\delta^{t+1}\right] + \kappa\mathbb{E}\left[\left\|g^{t+1} - \frac{1}{n}\sum_{i=1}^n \nabla f_i(z_i^{t+1})\right\|^2\right] + \eta\mathbb{E}\left[\left\|z^{t+1} - w^{t+1}\right\|^2\right]
$$

$$
+ \nu\mathbb{E}\left[\frac{1}{n}\sum_{i=1}^n \left\|z_i^{t+1} - w_i^{t+1}\right\|^2\right] + \rho\mathbb{E}\left[\left\|w^{t+1} - x^{t+1}\right\|^2\right] + \mu\mathbb{E}\left[\frac{1}{n}\sum_{i=1}^n \left\|w_i^{t+1} - x^{t+1}\right\|^2\right]
$$

$$
\leq \mathbb{E}\left[\delta^t\right] - \frac{\gamma}{2}\mathbb{E}\left[\left\|\nabla f(x^t)\right\|^2\right] - \left(\frac{1}{2\gamma} - \frac{L}{2}\right)\mathbb{E}\left[\left\|x^{t+1} - x^t\right\|^2\right] + \kappa\mathbb{E}\left[\left\|g^t - \frac{1}{n}\sum_{i=1}^n \nabla f_i(z_i^t)\right\|^2\right]
$$

$$
+ \eta\mathbb{E}\left[\left\|z^t - w^t\right\|^2\right] + \nu\mathbb{E}\left[\frac{1}{n}\sum_{i=1}^n \left\|z_i^t - w_i^t\right\|^2\right] + \rho\mathbb{E}\left[\left\|w^t - x^t\right\|^2\right] + \mu\mathbb{E}\left[\frac{1}{n}\sum_{i=1}^n \left\|w_i^t - x^t\right\|^2\right]
$$

$$
+ \left(\frac{3\gamma\omega_D(\omega_P + 1)\beta^2 L_{\max}^2}{np_D} + 32\gamma L_B^2\left(\frac{1}{p_P} + \frac{p_P}{\beta^2}\right)\theta + \frac{16\gamma L_B^2}{\beta}\left(\frac{1}{\beta} + \theta\right)\right.
$$

$$
+ 4\left(\frac{4\gamma L_A^2}{\beta} + \frac{6\gamma\omega_D\beta L_{\max}^2}{np_D}\right)\left(\frac{1}{\beta} + \omega_P\right)
$$

$$
\left. + 32\gamma\omega_P L_A^2\left(\frac{1}{p_P} + \frac{p_P}{\beta^2}\right) + \frac{48\gamma\omega_D\omega_P L_{\max}^2}{np_D}(\beta + p_P)\right)\mathbb{E}\left[\left\|x^{t+1} - x^t\right\|^2\right].
$$

(31)

Let us simplify the last bracket.

$$
I := \frac{3\gamma\omega_D(\omega_P + 1)\beta^2 L_{\max}^2}{np_D} + 32\gamma L_B^2\left(\frac{1}{p_P} + \frac{p_P}{\beta^2}\right)\theta + \frac{16\gamma L_B^2}{\beta}\left(\frac{1}{\beta} + \theta\right)
$$

$$
+ 4\left(\frac{4\gamma L_A^2}{\beta} + \frac{6\gamma\omega_D\beta L_{\max}^2}{np_D}\right)\left(\frac{1}{\beta} + \omega_P\right)
$$

$$
+ 32\gamma\omega_P L_A^2\left(\frac{1}{p_P} + \frac{p_P}{\beta^2}\right) + \frac{48\gamma\omega_D\omega_P L_{\max}^2}{np_D}(\beta + p_P)
$$

$$
\leq \left(\frac{3\gamma\omega_D(\omega_P + 1)\beta^2}{np_D} + \frac{24\gamma\omega_D}{np_D} + \frac{24\gamma\omega_D\omega_P\beta}{np_D} + \frac{48\gamma\omega_D\omega_P\beta}{np_D} + \frac{48\gamma\omega_D\omega_P p_P}{np_D}\right)L_{\max}^2
$$

$$
+ \left(\frac{16\gamma}{\beta^2} + \frac{16\gamma\omega_P}{\beta} + \frac{32\gamma\omega_P}{p_P} + \frac{32\gamma\omega_P p_P}{\beta^2}\right)L_A^2 + \left(\frac{32\gamma\theta}{p_P} + \frac{32\gamma p_P\theta}{\beta^2} + \frac{16\gamma}{\beta^2} + \frac{16\gamma\theta}{\beta}\right)L_B^2.
$$

(32)

We next consider the coefficients of $L_B^2$, $L_A^2$ and $L_{\max}^2$. First, for $L_B^2$, we have

$$
\frac{32\gamma\theta}{p_P} + \frac{32\gamma p_P\theta}{\beta^2} + \frac{16\gamma}{\beta^2} + \frac{16\gamma\theta}{\beta} \leq 32\gamma\left(\frac{\theta}{p_P}\left(1 + \frac{p_P}{\beta} + \frac{p_P^2}{\beta^2}\right) + \frac{1}{\beta^2}\right)
$$

$$
\leq 64\gamma\left(\frac{\theta}{p_P} + \frac{\theta p_P}{\beta^2} + \frac{1}{\beta^2}\right)
$$

$$
= 64\gamma\left(\frac{\theta}{p_P} + \frac{1 + \theta p_P}{\beta^2}\right).
$$

Next, the coefficient of $L_A^2$ can be bounded as

$$
\frac{16\gamma}{\beta^2} + \frac{16\gamma\omega_P}{\beta} + \frac{32\gamma\omega_P}{p_P} + \frac{32\gamma\omega_P p_P}{\beta^2} \leq 32\gamma\left(\frac{1}{\beta^2} + \frac{\omega_P}{p_P}\left(1 + \frac{p_P}{\beta} + \frac{p_P^2}{\beta^2}\right)\right)
$$

$$
\leq 64\gamma\left(\frac{\omega_P}{p_P} + \frac{\omega_P p_P}{\beta^2} + \frac{1}{\beta^2}\right)
$$

$$
\leq 64\gamma\left(\frac{\omega_P}{p_P} + \frac{1 + \omega_P p_P}{\beta^2}\right),
$$

and for $L_{\max}^2$ we obtain

$$\frac{3\gamma\omega_D(\omega_P+1)\beta^2}{np_D} + \frac{24\gamma\omega_D}{np_D} + \frac{24\gamma\omega_D\omega_P\beta}{np_D} + \frac{48\gamma\omega_D\omega_P\beta}{np_D} + \frac{48\gamma\omega_D\omega_Pp_P}{np_D}$$

$$\leq 72\gamma\omega_D\left(\frac{(\omega_P+1)\beta^2}{np_D} + \frac{1}{np_D} + \frac{\omega_P\beta}{np_D} + \frac{\omega_Pp_P}{np_D}\right)$$

$$\leq 144\gamma\omega_D\left(\frac{1}{np_D} + \frac{\omega_P\beta}{np_D} + \frac{\omega_Pp_P}{np_D}\right)$$

$$= 144\gamma\left(\frac{\omega_D\omega_P\beta}{np_D} + \frac{\omega_D(1+\omega_Pp_P)}{np_D}\right)$$

since $\frac{(\omega_P+1)\beta^2}{np_D} \leq \frac{1}{np_D} + \frac{\omega_P\beta}{np_D}$. Substituting these inequalities to (31) and (32), we get

$$\mathbb{E}\left[\delta^{t+1}\right] + \kappa\mathbb{E}\left[\left\|g^{t+1} - \frac{1}{n}\sum_{i=1}^n \nabla f_i(z_i^{t+1})\right\|^2\right] + \eta\mathbb{E}\left[\left\|z^{t+1} - w^{t+1}\right\|^2\right]$$

$$+ \nu\mathbb{E}\left[\frac{1}{n}\sum_{i=1}^n \left\|z_i^{t+1} - w_i^{t+1}\right\|^2\right] + \rho\mathbb{E}\left[\left\|w^{t+1} - x^{t+1}\right\|^2\right] + \mu\mathbb{E}\left[\frac{1}{n}\sum_{i=1}^n \left\|w_i^{t+1} - x^{t+1}\right\|^2\right]$$

$$\leq \mathbb{E}\left[\delta^t\right] - \frac{\gamma}{2}\mathbb{E}\left[\left\|\nabla f(x^t)\right\|^2\right] - \left(\frac{1}{2\gamma} - \frac{L}{2}\right)\mathbb{E}\left[\left\|x^{t+1} - x^t\right\|^2\right] + \kappa\mathbb{E}\left[\left\|g^t - \frac{1}{n}\sum_{i=1}^n \nabla f_i(z_i^t)\right\|^2\right]$$

$$+ \eta\mathbb{E}\left[\left\|z^t - w^t\right\|^2\right] + \nu\mathbb{E}\left[\frac{1}{n}\sum_{i=1}^n \left\|z_i^t - w_i^t\right\|^2\right] + \rho\mathbb{E}\left[\left\|w^t - x^t\right\|^2\right] + \mu\mathbb{E}\left[\frac{1}{n}\sum_{i=1}^n \left\|w_i^t - x^t\right\|^2\right]$$

$$+ 144\gamma\left(\left(\frac{\theta}{p_P} + \frac{1+\theta p_P}{\beta^2}\right)L_B^2 + \left(\frac{\omega_P}{p_P} + \frac{1+\omega_Pp_P}{\beta^2}\right)L_A^2\right.$$

$$\left. + \left(\frac{\omega_D\omega_P\beta}{np_D} + \frac{\omega_D(1+\omega_Pp_P)}{np_D}\right)L_{\max}^2\right)\mathbb{E}\left[\left\|x^{t+1} - x^t\right\|^2\right].$$

By collecting all the terms w.r.t. $\mathbb{E}\left[\left\|x^{t+1} - x^t\right\|^2\right]$, using the step size $\gamma$ from the theorem and Lemma H.2, we obtain

$$\mathbb{E}\left[\Psi^{t+1}\right] = \mathbb{E}\left[\delta^{t+1}\right] + \kappa\mathbb{E}\left[\left\|g^{t+1} - \frac{1}{n}\sum_{i=1}^n \nabla f_i(z_i^{t+1})\right\|^2\right] + \eta\mathbb{E}\left[\left\|z^{t+1} - w^{t+1}\right\|^2\right]$$

$$+ \nu\mathbb{E}\left[\frac{1}{n}\sum_{i=1}^n \left\|z_i^{t+1} - w_i^{t+1}\right\|^2\right] + \rho\mathbb{E}\left[\left\|w^{t+1} - x^{t+1}\right\|^2\right]$$

$$+ \mu\mathbb{E}\left[\frac{1}{n}\sum_{i=1}^n \left\|w_i^{t+1} - x^{t+1}\right\|^2\right]$$

$$\leq \mathbb{E}\left[\delta^t\right] - \frac{\gamma}{2}\mathbb{E}\left[\left\|\nabla f(x^t)\right\|^2\right] + \kappa\mathbb{E}\left[\left\|g^t - \frac{1}{n}\sum_{i=1}^n \nabla f_i(z_i^t)\right\|^2\right] + \eta\mathbb{E}\left[\left\|z^t - w^t\right\|^2\right]$$

$$+ \nu\mathbb{E}\left[\frac{1}{n}\sum_{i=1}^n \left\|z_i^t - w_i^t\right\|^2\right] + \rho\mathbb{E}\left[\left\|w^t - x^t\right\|^2\right] + \mu\mathbb{E}\left[\frac{1}{n}\sum_{i=1}^n \left\|w_i^t - x^t\right\|^2\right]$$

$$= \mathbb{E}\left[\Psi^t\right] - \frac{\gamma}{2}\mathbb{E}\left[\left\|\nabla f(x^t)\right\|^2\right].$$

It remains to rearrange and sum the last inequality for $t = 0, \ldots, T-1$. $\qquad\square$

**Corollary E.2.** *Let* $\mathcal{C}_i^t \in \mathbb{P}(0)$ *for all* $i \in [n]$ *(e.g. PermK), choose* $p_P = 1/(\omega_P + 1)$, $p_D = 1/(\omega_D + 1)$ *and*

$$\beta = \min\left\{\left(\frac{n}{\omega_D\omega_P(\omega_D + 1)}\right)^{1/3}, 1\right\}.$$

*Then, in the view of Theorem E.1, the iteration complexity is*

$$\mathcal{O}\left(\frac{\Psi^0}{\varepsilon}\left(L_{\max} + \left(\frac{\omega_D\omega_P(\omega_D + 1)}{n}\right)^{1/3}L_{\max} + \sqrt{\frac{\omega_D(\omega_D + 1)}{n}}L_{\max} + \sqrt{\omega_P(\omega_P + 1)}L_A\right)\right).$$

*Proof.* By Theorem E.1, up to a constant factor, the algorithm converges after

$$\bar{T} := \frac{\Psi^0}{\varepsilon}\left(L + \sqrt{\left(\frac{\theta}{p_P} + \frac{1 + \theta p_P}{\beta^2}\right)L_B^2 + \left(\frac{\omega_P}{p_P} + \frac{1 + \omega_P p_P}{\beta^2}\right)L_A^2 + \left(\frac{\omega_D\omega_P\beta}{np_D} + \frac{\omega_D(1 + \omega_P p_P)}{np_D}\right)L_{\max}^2}\right)$$

$$= \frac{\Psi^0}{\varepsilon}\left(L + \sqrt{\frac{1}{\beta^2}L_B^2 + \left(\frac{\omega_P}{p_P} + \frac{1 + \omega_P p_P}{\beta^2}\right)L_A^2 + \left(\frac{\omega_D\omega_P\beta}{np_D} + \frac{\omega_D(1 + \omega_P p_P)}{np_D}\right)L_{\max}^2}\right)$$

$$\leq \frac{\Psi^0}{\varepsilon}\left(L + \sqrt{\frac{L_B^2}{\beta^2} + \left(\frac{\omega_P}{p_P} + \frac{2}{\beta^2}\right)L_A^2 + \left(\frac{\omega_D\omega_P\beta}{np_D} + \frac{2\omega_D}{np_D}\right)L_{\max}^2}\right)$$

$$\leq \frac{\Psi^0}{\varepsilon}\left(L + \sqrt{\frac{L_B^2}{\beta^2} + \left(\omega_P(\omega_P + 1) + \frac{2}{\beta^2}\right)L_A^2 + \left(\frac{\omega_D\omega_P(\omega_D + 1)\beta}{n} + \frac{2\omega_D(\omega_D + 1)}{n}\right)L_{\max}^2}\right)$$

$$\leq \frac{2\Psi^0}{\varepsilon}\left(L + \sqrt{\frac{L_A^2 + L_B^2}{\beta^2} + \frac{\omega_D\omega_P(\omega_D + 1)\beta}{n}L_{\max}^2 + \frac{\omega_D(\omega_D + 1)}{n}L_{\max}^2 + \omega_P(\omega_P + 1)L_A^2}\right)$$

iterations, where we use the choice of $p_P$ and $p_D$. Using Lemma C.1, we have $L_A^2 + L_B^2 \leq L_{\max}^2$ and hence

$$\bar{T} \leq \frac{2\Psi^0}{\varepsilon}\left(L + \sqrt{\left(\frac{1}{\beta^2} + \frac{\omega_D\omega_P(\omega_D + 1)\beta}{n}\right)L_{\max}^2 + \frac{\omega_D(\omega_D + 1)}{n}L_{\max}^2 + \omega_P(\omega_P + 1)L_A^2}\right)$$

$$\leq \frac{4\Psi^0}{\varepsilon}\left(L + \sqrt{\left(1 + \left(\frac{\omega_D\omega_P(\omega_D + 1)}{n}\right)^{2/3}\right)L_{\max}^2 + \frac{\omega_D(\omega_D + 1)}{n}L_{\max}^2 + \omega_P(\omega_P + 1)L_A^2}\right)$$

$$\leq \frac{8\Psi^0}{\varepsilon}\left(L_{\max} + \left(\frac{\omega_D\omega_P(\omega_D + 1)}{n}\right)^{1/3}L_{\max} + \sqrt{\frac{\omega_D(\omega_D + 1)}{n}}L_{\max} + \sqrt{\omega_P(\omega_P + 1)}L_A\right),$$

where we substitute our choice of $\beta$. □

**Corollary E.3.** *Let* $\mathcal{C}_i^t$ *be the PermK compressors and* $\mathcal{Q}_i^t$ *be the independent (Assumption 1.6) RandK compressors, both with* $K = d/n$. *Then, in the view of Corollary E.2, the iteration complexity is*

$$\mathcal{O}\left(\frac{\Psi^0}{\varepsilon}\left(n^{2/3}L_{\max} + nL_A\right)\right),$$

*and the total communication complexity is*

$$\mathcal{O}\left(\frac{\Psi^0}{\varepsilon}\left(\frac{dL_{\max}}{n^{1/3}} + dL_A\right)\right).$$

*Proof.* The choice of compressors and parameters ensures that $\omega_P = \omega_D = n - 1$ (Lemma A.6). Thus, the iteration complexity is

$$\mathcal{O}\left(\frac{\Psi^0}{\varepsilon}\left(L_{\max} + \left(\frac{\omega_D\omega_P(\omega_D + 1)}{n}\right)^{1/3}L_{\max} + \sqrt{\frac{\omega_D(\omega_D + 1)}{n}}L_{\max} + \sqrt{\omega_P(\omega_P + 1)}L_A\right)\right)$$

$$= \mathcal{O}\left( \frac{\Psi^0}{\varepsilon} \left( L_{\max} + n^{2/3} L_{\max} + \sqrt{n} L_{\max} + n L_A \right) \right) = \mathcal{O}\left( \frac{\Psi^0}{\varepsilon} \left( n^{2/3} L_{\max} + n L_A \right) \right).$$

Since $p_P = p_D = 1/n$ and $K = d/n$, on average, the algorithm sends $\leq \frac{2d}{n}$ coordinates in both directions. Therefore, the total communication complexity is

$$\mathcal{O}\left( \frac{d}{n} \times \frac{\Psi^0}{\varepsilon} \left( n^{2/3} L_{\max} + n L_A \right) \right) = \mathcal{O}\left( \frac{\Psi^0}{\varepsilon} \left( \frac{d}{n^{1/3}} L_{\max} + d L_A \right) \right).$$

$\square$

### E.3 Polyak-Łojasiewicz condition

#### E.3.1 Main Results

As with MARINA-P, we provide the analysis of M3 under the Polyak-Łojasiewicz condition.

**Theorem E.8.** *Let Assumptions 1.1, 1.2, 1.5, 4.2 and D.9 be satisfied and suppose that the compressors $\mathcal{Q}_i^t \in \mathbb{U}(\omega_D)$ satisfy Assumption 1.6, $\{\mathcal{C}_i^t\}_{i=1}^n \in \mathbb{P}(\theta)$ and $\mathcal{C}_i^t \in \mathbb{U}(\omega_P)$ for all $i \in [n]$. Let $\gamma > 0$ be such that*

$$\gamma = \min \left\{ \left( L + \sqrt{ 1536 \left( \left( \frac{\theta}{p_P} + \frac{1 + \theta p_P}{\beta^2} \right) L_B^2 + \left( \frac{\omega_P}{p_P} + \frac{1 + \omega_P p_P}{\beta^2} \right) L_A^2 + \left( \frac{\omega_D \omega_P \beta}{n p_D} + \frac{\omega_D (1 + \omega_P p_P)}{n p_D} \right) L_{\max}^2 \right) } \right)^{-1}, \right.$$

$$\left. \frac{p_P}{2\mu}, \frac{p_D}{2\mu}, \frac{\beta}{4\mu} \right\}. \tag{33}$$

*Letting*

$$\Psi^t = \delta^t + \kappa \left\| g^t - \frac{1}{n} \sum_{i=1}^n \nabla f_i(z_i^t) \right\|^2 + \eta \left\| z^t - w^t \right\|^2 + \nu \frac{1}{n} \sum_{i=1}^n \left\| z_i^t - w_i^t \right\|^2$$

$$+ \rho \left\| w^t - x^t \right\|^2 + \tau \frac{1}{n} \sum_{i=1}^n \left\| w_i^t - x^t \right\|^2,$$

*where $\kappa = \frac{2\gamma}{p_D}$, $\eta = \frac{8\gamma L_B^2}{\beta}$, $\nu = \frac{8\gamma L_A^2}{\beta} + \frac{24\gamma \omega_D \beta L_{\max}^2}{n p_D}$, $\rho = 128\gamma L_B^2 \left( \frac{1}{p_P} + \frac{p_P}{\beta^2} \right)$ and $\tau = 128\gamma L_A^2 \left( \frac{1}{p_P} + \frac{p_P}{\beta^2} \right) + \frac{384\gamma \omega_D L_{\max}^2}{n p_D} (\beta + p_P)$, M3 ensures that for each $T \geq 1$*

$$\mathbb{E}\left[ \Psi^T \right] \leq (1 - \gamma\mu)^T \Psi^0.$$

**Corollary E.9.** *Let $\mathcal{C}_i^t \in \mathbb{P}(0)$ for all $i \in [n]$ (e.g. PermK), choose $p_P = 1/(\omega_P + 1)$, $p_D = 1/(\omega_D + 1)$ and*

$$\beta = \min \left\{ \left( \frac{n}{\omega_D \omega_P (\omega_D + 1)} \right)^{1/3}, 1 \right\}.$$

*Then, in the view of Theorem E.8, Algorithm 2 ensures that $\mathbb{E}\left[ f(x^T) - f^* \right] \leq \varepsilon$ after*

$$\mathcal{O}\left( \max \left\{ \frac{\left( 1 + \left( \frac{\omega_D \omega_P (\omega_D + 1)}{n} \right)^{1/3} + \sqrt{\frac{\omega_D (\omega_D + 1)}{n}} \right) L_{\max} + \sqrt{\omega_P (\omega_P + 1)} L_A}{\mu}, \omega_P + 1, \omega_D + 1, \left( \frac{\omega_D \omega_P (\omega_D + 1)}{n} \right)^{1/3} \right\} \log \frac{\Psi^0}{\varepsilon} \right)$$

*iterations.*

**Corollary E.10.** *Let $\mathcal{C}_i^t$ be the PermK compressors and $\mathcal{Q}_i^t$ be the independent (Assumption 1.6) RandK compressors, both with $K = d/n$. Then, in the view of Corollary E.9, the total communication complexity is*

$$\mathcal{O}\left( \left( \frac{d L_{\max}}{n^{1/3} \mu} + \frac{d L_A}{\mu} + d \right) \log \frac{\Psi^0}{\varepsilon} \right).$$

### E.3.2 Proofs

**Theorem E.8.** *Let Assumptions 1.1, 1.2, 1.5, 4.2 and D.9 be satisfied and suppose that the compressors $\mathcal{Q}_i^t \in \mathbb{U}(\omega_D)$ satisfy Assumption 1.6, $\{\mathcal{C}_i^t\}_{i=1}^n \in \mathbb{P}(\theta)$ and $\mathcal{C}_i^t \in \mathbb{U}(\omega_P)$ for all $i \in [n]$. Let $\gamma > 0$ be such that*

$$\gamma = \min\left\{ \left( L + \sqrt{1536\left( \left(\tfrac{\theta}{p_P} + \tfrac{1+\theta p_P}{\beta^2}\right) L_B^2 + \left(\tfrac{\omega_P}{p_P} + \tfrac{1+\omega_P p_P}{\beta^2}\right) L_A^2 + \left(\tfrac{\omega_D \omega_P \beta}{n p_D} + \tfrac{\omega_D(1+\omega_P p_P)}{n p_D}\right) L_{\max}^2 \right)} \right)^{-1}, \right.$$

$$\left. \tfrac{p_P}{2\mu}, \tfrac{p_D}{2\mu}, \tfrac{\beta}{4\mu} \right\}. \tag{33}$$

*Letting*

$$\Psi^t = \delta^t + \kappa \left\| g^t - \frac{1}{n}\sum_{i=1}^n \nabla f_i(z_i^t) \right\|^2 + \eta \left\| z^t - w^t \right\|^2 + \nu \frac{1}{n}\sum_{i=1}^n \left\| z_i^t - w_i^t \right\|^2$$

$$+ \rho \left\| w^t - x^t \right\|^2 + \tau \frac{1}{n}\sum_{i=1}^n \left\| w_i^t - x^t \right\|^2,$$

*where $\kappa = \frac{2\gamma}{p_D}$, $\eta = \frac{8\gamma L_B^2}{\beta}$, $\nu = \frac{8\gamma L_A^2}{\beta} + \frac{24\gamma \omega_D \beta L_{\max}^2}{n p_D}$, $\rho = 128\gamma L_B^2\left(\frac{1}{p_P} + \frac{p_P}{\beta^2}\right)$ and $\tau = 128\gamma L_A^2\left(\frac{1}{p_P} + \frac{p_P}{\beta^2}\right) + \frac{384\gamma \omega_D L_{\max}^2}{n p_D}(\beta + p_P)$, M3 ensures that for each $T \geq 1$*

$$\mathbb{E}\left[\Psi^T\right] \leq (1 - \gamma\mu)^T \Psi^0.$$

*Proof.* Starting as in the proof of Theorem E.1, we have

$$\mathbb{E}\left[\delta^{t+1}\right] + \kappa\mathbb{E}\left[\left\| g^{t+1} - \frac{1}{n}\sum_{i=1}^n \nabla f_i(z_i^{t+1}) \right\|^2\right] + \eta\mathbb{E}\left[\left\| z^{t+1} - w^{t+1} \right\|^2\right]$$

$$+ \nu\mathbb{E}\left[\frac{1}{n}\sum_{i=1}^n \left\| z_i^{t+1} - w_i^{t+1} \right\|^2\right] + \rho\mathbb{E}\left[\left\| w^{t+1} - x^{t+1} \right\|^2\right] + \tau\mathbb{E}\left[\frac{1}{n}\sum_{i=1}^n \left\| w_i^{t+1} - x^{t+1} \right\|^2\right]$$

$$\leq \mathbb{E}\left[\delta^t\right] - \frac{\gamma}{2}\mathbb{E}\left[\left\| \nabla f(x^t) \right\|^2\right] - \left(\frac{1}{2\gamma} - \frac{L}{2}\right)\mathbb{E}\left[\left\| x^{t+1} - x^t \right\|^2\right] + \gamma\mathbb{E}\left[\left\| g^t - \frac{1}{n}\sum_{i=1}^n \nabla f_i(z_i^t) \right\|^2\right]$$

$$+ 2\gamma L_A^2 \frac{1}{n}\sum_{i=1}^n \left( \mathbb{E}\left[\left\| z_i^t - w_i^t \right\|^2\right] + \mathbb{E}\left[\left\| w_i^t - x^t \right\|^2\right] \right)$$

$$+ 2\gamma L_B^2 \left( \mathbb{E}\left[\left\| z^t - w^t \right\|^2\right] + \mathbb{E}\left[\left\| w^t - x^t \right\|^2\right] \right)$$

$$+ \kappa\left( \frac{\omega_D L_{\max}^2}{n}\left( 4p_P\beta^2\mathbb{E}\left[\frac{1}{n}\sum_{i=1}^n \left\| w_i^t - x^t \right\|^2\right] + 3\beta^2\mathbb{E}\left[\frac{1}{n}\sum_{i=1}^n \left\| z_i^t - w_i^t \right\|^2\right]\right.\right.$$

$$\left.\left. + 3(\omega_P + 1)\beta^2\mathbb{E}\left[\left\| x^{t+1} - x^t \right\|^2\right] \right)\right)$$

$$+ \kappa(1 - p_D)\mathbb{E}\left[\left\| g^t - \frac{1}{n}\sum_{i=1}^n \nabla f_i(z_i^t) \right\|^2\right]$$

$$+ \eta\left( \left(1 - \frac{\beta}{2}\right)\mathbb{E}\left[\left\| z^t - w^t \right\|^2\right] + 4\left(\frac{1}{\beta} + \theta\right)\mathbb{E}\left[\left\| x^{t+1} - x^t \right\|^2\right] + 4p_P\left(1 + \frac{p_P}{\beta}\right)\mathbb{E}\left[\left\| w^t - x^t \right\|^2\right] \right)$$

$$+ \nu\left( \left(1 - \frac{\beta}{2}\right)\mathbb{E}\left[\frac{1}{n}\sum_{i=1}^n \left\| z_i^t - w_i^t \right\|^2\right] + 4\left(\frac{1}{\beta} + \omega_P\right)\mathbb{E}\left[\left\| x^{t+1} - x^t \right\|^2\right] \right)$$

$$+ 4p_P\left(1 + \frac{p_P}{\beta}\right)\mathbb{E}\left[\frac{1}{n}\sum_{i=1}^{n}\left\|w_i^t - x^t\right\|^2\right]\right)$$

$$+ \rho(1 - p_P)\left(\mathbb{E}\left[\left\|w^t - x^t\right\|^2\right] + \theta\mathbb{E}\left[\left\|x^{t+1} - x^t\right\|^2\right]\right)$$

$$+ \tau(1 - p_P)\left(\mathbb{E}\left[\frac{1}{n}\sum_{i=1}^{n}\left\|w_i^t - x^t\right\|^2\right] + \omega_P\mathbb{E}\left[\left\|x^{t+1} - x^t\right\|^2\right]\right)$$

for some $\kappa, \eta, \nu, \rho, \tau \geq 0$. This time, we let $\kappa = \frac{2\gamma}{p_D}$ and $\eta = \frac{8\gamma L_B^2}{\beta}$, which gives $\gamma + \kappa(1 - p_D) = \kappa\left(1 - \frac{p_D}{2}\right)$ and $2\gamma L_B^2 + \eta(1 - \beta/2) = \eta\left(1 - \frac{\beta}{4}\right)$. Hence

$$\mathbb{E}\left[\delta^{t+1}\right] + \kappa\mathbb{E}\left[\left\|g^{t+1} - \frac{1}{n}\sum_{i=1}^{n}\nabla f_i(z_i^{t+1})\right\|^2\right] + \eta\mathbb{E}\left[\left\|z^{t+1} - w^{t+1}\right\|^2\right]$$

$$+ \nu\mathbb{E}\left[\frac{1}{n}\sum_{i=1}^{n}\left\|z_i^{t+1} - w_i^{t+1}\right\|^2\right] + \rho\mathbb{E}\left[\left\|w^{t+1} - x^{t+1}\right\|^2\right] + \tau\mathbb{E}\left[\frac{1}{n}\sum_{i=1}^{n}\left\|w_i^{t+1} - x^{t+1}\right\|^2\right]$$

$$\leq \mathbb{E}\left[\delta^t\right] - \frac{\gamma}{2}\mathbb{E}\left[\left\|\nabla f(x^t)\right\|^2\right] - \left(\frac{1}{2\gamma} - \frac{L}{2}\right)\mathbb{E}\left[\left\|x^{t+1} - x^t\right\|^2\right]$$

$$+ \kappa\left(1 - \frac{p_D}{2}\right)\mathbb{E}\left[\left\|g^t - \frac{1}{n}\sum_{i=1}^{n}\nabla f_i(z_i^t)\right\|^2\right] + \eta\left(1 - \frac{\beta}{4}\right)\mathbb{E}\left[\left\|z^t - w^t\right\|^2\right]$$

$$+ 2\gamma L_A^2\frac{1}{n}\sum_{i=1}^{n}\left(\mathbb{E}\left[\left\|z_i^t - w_i^t\right\|^2\right] + \mathbb{E}\left[\left\|w_i^t - x^t\right\|^2\right]\right) + 2\gamma L_B^2\mathbb{E}\left[\left\|w^t - x^t\right\|^2\right]$$

$$+ \frac{2\gamma}{p_D}\left(\frac{\omega_D L_{\max}^2}{n}\left(4p_P\beta^2\mathbb{E}\left[\frac{1}{n}\sum_{i=1}^{n}\left\|w_i^t - x^t\right\|^2\right] + 3\beta^2\mathbb{E}\left[\frac{1}{n}\sum_{i=1}^{n}\left\|z_i^t - w_i^t\right\|^2\right]\right.\right.$$

$$\left.\left. + 3(\omega_P + 1)\beta^2\mathbb{E}\left[\left\|x^{t+1} - x^t\right\|^2\right]\right)\right)$$

$$+ \frac{8\gamma L_B^2}{\beta}\left(4\left(\frac{1}{\beta} + \theta\right)\mathbb{E}\left[\left\|x^{t+1} - x^t\right\|^2\right] + 4p_P\left(1 + \frac{p_P}{\beta}\right)\mathbb{E}\left[\left\|w^t - x^t\right\|^2\right]\right)$$

$$+ \nu\left(\left(1 - \frac{\beta}{2}\right)\mathbb{E}\left[\frac{1}{n}\sum_{i=1}^{n}\left\|z_i^t - w_i^t\right\|^2\right] + 4\left(\frac{1}{\beta} + \omega_P\right)\mathbb{E}\left[\left\|x^{t+1} - x^t\right\|^2\right]\right.$$

$$\left. + 4p_P\left(1 + \frac{p_P}{\beta}\right)\mathbb{E}\left[\frac{1}{n}\sum_{i=1}^{n}\left\|w_i^t - x^t\right\|^2\right]\right)$$

$$+ \rho\left((1 - p_P)\mathbb{E}\left[\left\|w^t - x^t\right\|^2\right] + \theta\mathbb{E}\left[\left\|x^{t+1} - x^t\right\|^2\right]\right)$$

$$+ \tau\left((1 - p_P)\mathbb{E}\left[\frac{1}{n}\sum_{i=1}^{n}\left\|w_i^t - x^t\right\|^2\right] + \omega_P\mathbb{E}\left[\left\|x^{t+1} - x^t\right\|^2\right]\right).$$

Rearranging the terms

$$\mathbb{E}\left[\delta^{t+1}\right] + \kappa\mathbb{E}\left[\left\|g^{t+1} - \frac{1}{n}\sum_{i=1}^{n}\nabla f_i(z_i^{t+1})\right\|^2\right] + \eta\mathbb{E}\left[\left\|z^{t+1} - w^{t+1}\right\|^2\right]$$

$$+ \nu\mathbb{E}\left[\frac{1}{n}\sum_{i=1}^{n}\left\|z_i^{t+1} - w_i^{t+1}\right\|^2\right] + \rho\mathbb{E}\left[\left\|w^{t+1} - x^{t+1}\right\|^2\right] + \tau\mathbb{E}\left[\frac{1}{n}\sum_{i=1}^{n}\left\|w_i^{t+1} - x^{t+1}\right\|^2\right]$$

$$\leq \mathbb{E}\left[\delta^t\right] - \frac{\gamma}{2}\mathbb{E}\left[\left\|\nabla f(x^t)\right\|^2\right] - \left(\frac{1}{2\gamma} - \frac{L}{2}\right)\mathbb{E}\left[\left\|x^{t+1} - x^t\right\|^2\right]$$

$$+ \kappa \left(1 - \frac{p_D}{2}\right) \mathbb{E}\left[\left\|g^t - \frac{1}{n}\sum_{i=1}^n \nabla f_i(z_i^t)\right\|^2\right] + \eta\left(1 - \frac{\beta}{4}\right)\mathbb{E}\left[\left\|z^t - w^t\right\|^2\right]$$

$$+ \left(\nu\left(1 - \frac{\beta}{2}\right) + 2\gamma L_A^2 + \frac{6\gamma\omega_D\beta^2 L_{\max}^2}{np_D}\right)\mathbb{E}\left[\frac{1}{n}\sum_{i=1}^n \left\|z_i^t - w_i^t\right\|^2\right]$$

$$+ \left(\frac{6\gamma\omega_D(\omega_P + 1)\beta^2 L_{\max}^2}{np_D} + \rho\theta + \frac{32\gamma L_B^2}{\beta}\left(\frac{1}{\beta} + \theta\right) + 4\nu\left(\frac{1}{\beta} + \omega_P\right) + \tau\omega_P\right)\mathbb{E}\left[\left\|x^{t+1} - x^t\right\|^2\right]$$

$$+ \left(\rho(1 - p_P) + 2\gamma L_B^2 + \frac{32\gamma L_B^2 p_P}{\beta}\left(1 + \frac{p_P}{\beta}\right)\right)\mathbb{E}\left[\left\|w^t - x^t\right\|^2\right]$$

$$+ \left(\tau(1 - p_P) + 2\gamma L_A^2 + 4\nu p_P\left(1 + \frac{p_P}{\beta}\right) + \frac{8\gamma\omega_D p_P\beta^2 L_{\max}^2}{np_D}\right)\mathbb{E}\left[\frac{1}{n}\sum_{i=1}^n \left\|w_i^t - x^t\right\|^2\right].$$

Considering the coefficient of $\mathbb{E}\left[\left\|w^t - x^t\right\|^2\right]$ and using the inequality $xy \le \frac{x^2 + y^2}{2}$ for all $x, y \ge 0$, we get

$$\rho(1 - p_P) + 2\gamma L_B^2 + \frac{32\gamma L_B^2 p_P}{\beta}\left(1 + \frac{p_P}{\beta}\right) \le \rho(1 - p_P) + 32\gamma L_B^2\left(1 + \frac{p_P}{\beta} + \frac{p_P^2}{\beta^2}\right)$$

$$\le \rho(1 - p_P) + 64\gamma L_B^2\left(1 + \frac{p_P^2}{\beta^2}\right)$$

$$= \rho\left(1 - \frac{p_P}{2}\right),$$

where we define $\rho = 128\gamma L_B^2\left(\frac{1}{p_P} + \frac{p_P}{\beta^2}\right)$. Substituting this choice of $\rho$, we obtain

$$\mathbb{E}\left[\delta^{t+1}\right] + \kappa\mathbb{E}\left[\left\|g^{t+1} - \frac{1}{n}\sum_{i=1}^n \nabla f_i(z_i^{t+1})\right\|^2\right] + \eta\mathbb{E}\left[\left\|z^{t+1} - w^{t+1}\right\|^2\right]$$

$$+ \nu\mathbb{E}\left[\frac{1}{n}\sum_{i=1}^n \left\|z_i^{t+1} - w_i^{t+1}\right\|^2\right] + \rho\mathbb{E}\left[\left\|w^{t+1} - x^{t+1}\right\|^2\right] + \tau\mathbb{E}\left[\frac{1}{n}\sum_{i=1}^n \left\|w_i^{t+1} - x^{t+1}\right\|^2\right]$$

$$\le \mathbb{E}\left[\delta^t\right] - \frac{\gamma}{2}\mathbb{E}\left[\left\|\nabla f(x^t)\right\|^2\right] - \left(\frac{1}{2\gamma} - \frac{L}{2}\right)\mathbb{E}\left[\left\|x^{t+1} - x^t\right\|^2\right]$$

$$+ \kappa\left(1 - \frac{p_D}{2}\right)\mathbb{E}\left[\left\|g^t - \frac{1}{n}\sum_{i=1}^n \nabla f_i(z_i^t)\right\|^2\right] + \eta\left(1 - \frac{\beta}{4}\right)\mathbb{E}\left[\left\|z^t - w^t\right\|^2\right]$$

$$+ \rho\left(1 - \frac{p_P}{2}\right)\mathbb{E}\left[\left\|w^t - x^t\right\|^2\right] + \left(\nu\left(1 - \frac{\beta}{2}\right) + 2\gamma L_A^2 + \frac{6\gamma\omega_D\beta^2 L_{\max}^2}{np_D}\right)\mathbb{E}\left[\frac{1}{n}\sum_{i=1}^n \left\|z_i^t - w_i^t\right\|^2\right]$$

$$+ \left(\frac{6\gamma\omega_D(\omega_P + 1)\beta^2 L_{\max}^2}{np_D} + 128\gamma L_B^2\left(\frac{1}{p_P} + \frac{p_P}{\beta^2}\right)\theta + \frac{32\gamma L_B^2}{\beta}\left(\frac{1}{\beta} + \theta\right)\right.$$

$$\left. + 4\nu\left(\frac{1}{\beta} + \omega_P\right) + \tau\omega_P\right)\mathbb{E}\left[\left\|x^{t+1} - x^t\right\|^2\right]$$

$$+ \left(\tau(1 - p_P) + 2\gamma L_A^2 + 4\nu p_P\left(1 + \frac{p_P}{\beta}\right) + \frac{8\gamma\omega_D p_P\beta^2 L_{\max}^2}{np_D}\right)\mathbb{E}\left[\frac{1}{n}\sum_{i=1}^n \left\|w_i^t - x^t\right\|^2\right].$$

Similarly, taking $\nu = \frac{8\gamma L_A^2}{\beta} + \frac{24\gamma\omega_D\beta L_{\max}^2}{np_D}$ gives $\nu\left(1 - \frac{\beta}{2}\right) + 2\gamma L_A^2 + \frac{6\gamma\omega_D\beta^2 L_{\max}^2}{np_D} = \nu\left(1 - \frac{\beta}{4}\right)$, so

$$\mathbb{E}\left[\delta^{t+1}\right] + \kappa\mathbb{E}\left[\left\|g^{t+1} - \frac{1}{n}\sum_{i=1}^n \nabla f_i(z_i^{t+1})\right\|^2\right] + \eta\mathbb{E}\left[\left\|z^{t+1} - w^{t+1}\right\|^2\right]$$

$$+ \nu \mathbb{E}\left[\frac{1}{n}\sum_{i=1}^{n}\left\|z_i^{t+1} - w_i^{t+1}\right\|^2\right] + \rho \mathbb{E}\left[\left\|w^{t+1} - x^{t+1}\right\|^2\right] + \tau \mathbb{E}\left[\frac{1}{n}\sum_{i=1}^{n}\left\|w_i^{t+1} - x^{t+1}\right\|^2\right]$$

$$\leq \mathbb{E}\left[\delta^t\right] - \frac{\gamma}{2}\mathbb{E}\left[\left\|\nabla f(x^t)\right\|^2\right] - \left(\frac{1}{2\gamma} - \frac{L}{2}\right)\mathbb{E}\left[\left\|x^{t+1} - x^t\right\|^2\right]$$

$$+ \kappa\left(1 - \frac{p_D}{2}\right)\mathbb{E}\left[\left\|g^t - \frac{1}{n}\sum_{i=1}^{n}\nabla f_i(z_i^t)\right\|^2\right] + \eta\left(1 - \frac{\beta}{4}\right)\mathbb{E}\left[\left\|z^t - w^t\right\|^2\right]$$

$$+ \rho\left(1 - \frac{p_P}{2}\right)\mathbb{E}\left[\left\|w^t - x^t\right\|^2\right] + \nu\left(1 - \frac{\beta}{4}\right)\mathbb{E}\left[\frac{1}{n}\sum_{i=1}^{n}\left\|z_i^t - w_i^t\right\|^2\right]$$

$$+ \left(\frac{6\gamma\omega_D(\omega_P + 1)\beta^2 L_{\max}^2}{np_D} + 128\gamma L_B^2\left(\frac{1}{p_P} + \frac{p_P}{\beta^2}\right)\theta + \frac{32\gamma L_B^2}{\beta}\left(\frac{1}{\beta} + \theta\right)\right.$$

$$\left. + 4\left(\frac{8\gamma L_A^2}{\beta} + \frac{24\gamma\omega_D\beta L_{\max}^2}{np_D}\right)\left(\frac{1}{\beta} + \omega_P\right) + \tau\omega_P\right)\mathbb{E}\left[\left\|x^{t+1} - x^t\right\|^2\right]$$

$$+ \left(\tau(1 - p_P) + 2\gamma L_A^2 + 4\left(\frac{8\gamma L_A^2}{\beta} + \frac{24\gamma\omega_D\beta L_{\max}^2}{np_D}\right)p_P\left(1 + \frac{p_P}{\beta}\right)\right.$$

$$\left. + \frac{8\gamma\omega_D p_P\beta^2 L_{\max}^2}{np_D}\right)\mathbb{E}\left[\frac{1}{n}\sum_{i=1}^{n}\left\|w_i^t - x^t\right\|^2\right].$$

Considering the last bracket, we have

$$\tau(1 - p_P) + 2\gamma L_A^2 + 4\left(\frac{8\gamma L_A^2}{\beta} + \frac{24\gamma\omega_D\beta L_{\max}^2}{np_D}\right)p_P\left(1 + \frac{p_P}{\beta}\right) + \frac{8\gamma\omega_D p_P\beta^2 L_{\max}^2}{np_D}$$

$$= \tau(1 - p_P) + 2\gamma L_A^2 + \frac{32\gamma p_P L_A^2}{\beta} + \frac{32\gamma p_P^2 L_A^2}{\beta^2} + \frac{96\gamma\omega_D p_P\beta L_{\max}^2}{np_D} + \frac{96\gamma\omega_D p_P^2 L_{\max}^2}{np_D}$$

$$+ \frac{8\gamma\omega_D p_P\beta^2 L_{\max}^2}{np_D}$$

$$\leq \tau(1 - p_P) + 32\gamma L_A^2\left(1 + \frac{p_P}{\beta} + \frac{p_P^2}{\beta^2}\right) + \frac{96\gamma\omega_D p_P L_{\max}^2}{np_D}(\beta + p_P + \beta^2)$$

$$\leq \tau(1 - p_P) + 64\gamma L_A^2\left(1 + \frac{p_P^2}{\beta^2}\right) + \frac{192\gamma\omega_D p_P L_{\max}^2}{np_D}(\beta + p_P)$$

$$= \tau\left(1 - \frac{p_P}{2}\right)$$

for $\tau = 128\gamma L_A^2\left(\frac{1}{p_P} + \frac{p_P}{\beta^2}\right) + \frac{384\gamma\omega_D L_{\max}^2}{np_D}(\beta + p_P)$. Then

$$\mathbb{E}\left[\delta^{t+1}\right] + \kappa\mathbb{E}\left[\left\|g^{t+1} - \frac{1}{n}\sum_{i=1}^{n}\nabla f_i(z_i^{t+1})\right\|^2\right] + \eta\mathbb{E}\left[\left\|z^{t+1} - w^{t+1}\right\|^2\right]$$

$$+ \nu\mathbb{E}\left[\frac{1}{n}\sum_{i=1}^{n}\left\|z_i^{t+1} - w_i^{t+1}\right\|^2\right] + \rho\mathbb{E}\left[\left\|w^{t+1} - x^{t+1}\right\|^2\right] + \tau\mathbb{E}\left[\frac{1}{n}\sum_{i=1}^{n}\left\|w_i^{t+1} - x^{t+1}\right\|^2\right]$$

$$\leq \mathbb{E}\left[\delta^t\right] - \frac{\gamma}{2}\mathbb{E}\left[\left\|\nabla f(x^t)\right\|^2\right] - \left(\frac{1}{2\gamma} - \frac{L}{2}\right)\mathbb{E}\left[\left\|x^{t+1} - x^t\right\|^2\right]$$

$$+ \kappa\left(1 - \frac{p_D}{2}\right)\mathbb{E}\left[\left\|g^t - \frac{1}{n}\sum_{i=1}^{n}\nabla f_i(z_i^t)\right\|^2\right] + \eta\left(1 - \frac{\beta}{4}\right)\mathbb{E}\left[\left\|z^t - w^t\right\|^2\right]$$

$$+ \rho\left(1 - \frac{p_P}{2}\right)\mathbb{E}\left[\left\|w^t - x^t\right\|^2\right] + \nu\left(1 - \frac{\beta}{4}\right)\mathbb{E}\left[\frac{1}{n}\sum_{i=1}^{n}\left\|z_i^t - w_i^t\right\|^2\right]$$

$$+ \tau \left(1 - \frac{p_P}{2}\right) \mathbb{E}\left[\frac{1}{n}\sum_{i=1}^{n} \|w_i^t - x^t\|^2\right]$$

$$+ \left(\frac{6\gamma\omega_D(\omega_P + 1)\beta^2 L_{\max}^2}{np_D} + 128\gamma L_B^2\left(\frac{1}{p_P} + \frac{p_P}{\beta^2}\right)\theta + \frac{32\gamma L_B^2}{\beta}\left(\frac{1}{\beta} + \theta\right)\right.$$

$$+ 4\left(\frac{8\gamma L_A^2}{\beta} + \frac{24\gamma\omega_D\beta L_{\max}^2}{np_D}\right)\left(\frac{1}{\beta} + \omega_P\right)$$

$$\left.+ 128\gamma\omega_P L_A^2\left(\frac{1}{p_P} + \frac{p_P}{\beta^2}\right) + \frac{384\gamma\omega_D\omega_P L_{\max}^2}{np_D}(\beta + p_P)\right)\mathbb{E}\left[\|x^{t+1} - x^t\|^2\right], \quad (34)$$

where the last bracket can be bounded as

$$I := \frac{6\gamma\omega_D(\omega_P + 1)\beta^2 L_{\max}^2}{np_D} + 128\gamma L_B^2\left(\frac{1}{p_P} + \frac{p_P}{\beta^2}\right)\theta + \frac{32\gamma L_B^2}{\beta}\left(\frac{1}{\beta} + \theta\right)$$

$$+ 4\left(\frac{8\gamma L_A^2}{\beta} + \frac{24\gamma\omega_D\beta L_{\max}^2}{np_D}\right)\left(\frac{1}{\beta} + \omega_P\right) + 128\gamma\omega_P L_A^2\left(\frac{1}{p_P} + \frac{p_P}{\beta^2}\right)$$

$$+ \frac{384\gamma\omega_D\omega_P L_{\max}^2}{np_D}(\beta + p_P)$$

$$= \left(\frac{6\gamma\omega_D(\omega_P + 1)\beta^2}{np_D} + \frac{96\gamma\omega_D\beta}{np_D}\left(\frac{1}{\beta} + \omega_P\right) + \frac{384\gamma\omega_D\omega_P}{np_D}(\beta + p_P)\right)L_{\max}^2$$

$$+ \left(\frac{32\gamma}{\beta}\left(\frac{1}{\beta} + \omega_P\right) + 128\gamma\omega_P\left(\frac{1}{p_P} + \frac{p_P}{\beta^2}\right)\right)L_A^2$$

$$+ \left(128\gamma\left(\frac{1}{p_P} + \frac{p_P}{\beta^2}\right)\theta + \frac{32\gamma}{\beta}\left(\frac{1}{\beta} + \theta\right)\right)L_B^2$$

$$= \left(\frac{6\gamma\omega_D(\omega_P + 1)\beta^2}{np_D} + \frac{96\gamma\omega_D}{np_D} + \frac{96\gamma\omega_D\omega_P\beta}{np_D} + \frac{384\gamma\omega_D\omega_P\beta}{np_D} + \frac{384\gamma\omega_D\omega_P p_P}{np_D}\right)L_{\max}^2$$

$$+ \left(\frac{32\gamma}{\beta^2} + \frac{32\gamma\omega_P}{\beta} + \frac{128\gamma\omega_P}{p_P} + \frac{128\gamma\omega_P p_P}{\beta^2}\right)L_A^2$$

$$+ \left(\frac{128\gamma\theta}{p_P} + \frac{128\gamma p_P\theta}{\beta^2} + \frac{32\gamma}{\beta^2} + \frac{32\gamma\theta}{\beta}\right)L_B^2.$$

$$(35)$$

We next consider the coefficients of $L_B^2$, $L_A^2$ and $L_{\max}^2$. First, for $L_B^2$, we have

$$\frac{128\gamma\theta}{p_P} + \frac{128\gamma p_P\theta}{\beta^2} + \frac{32\gamma}{\beta^2} + \frac{32\gamma\theta}{\beta} \leq 128\gamma\left(\frac{\theta}{p_P}\left(1 + \frac{p_P}{\beta} + \frac{p_P^2}{\beta^2}\right) + \frac{1}{\beta^2}\right)$$

$$\leq 256\gamma\left(\frac{\theta}{p_P} + \frac{\theta p_P}{\beta^2} + \frac{1}{\beta^2}\right)$$

$$= 256\gamma\left(\frac{\theta}{p_P} + \frac{1 + \theta p_P}{\beta^2}\right).$$

Next, the coefficient of $L_A^2$ can be bounded as

$$\frac{32\gamma}{\beta^2} + \frac{32\gamma\omega_P}{\beta} + \frac{128}{p_P} + \frac{128\gamma\omega_P p_P}{\beta^2} \leq 128\gamma\left(\frac{1}{\beta^2} + \frac{\omega_P}{p_P}\left(1 + \frac{p_P}{\beta} + \frac{p_P^2}{\beta^2}\right)\right)$$

$$\leq 256\gamma\left(\frac{\omega_P}{p_P} + \frac{\omega_P p_P}{\beta^2} + \frac{1}{\beta^2}\right)$$

$$\leq 256\gamma\left(\frac{\omega_P}{p_P} + \frac{1 + \omega_P p_P}{\beta^2}\right),$$

and for $L_{\max}^2$ we obtain

$$\frac{6\gamma\omega_D(\omega_P + 1)\beta^2}{np_D} + \frac{96\gamma\omega_D}{np_D} + \frac{96\gamma\omega_D\omega_P\beta}{np_D} + \frac{384\gamma\omega_D\omega_P\beta}{np_D} + \frac{384\gamma\omega_D\omega_P p_P}{np_D}$$

$$\leq 384\gamma\omega_D\left(\frac{(\omega_P+1)\beta^2}{np_D}+\frac{1}{np_D}+\frac{\omega_P\beta}{np_D}+\frac{\omega_Pp_P}{np_D}\right)$$

$$\leq 768\gamma\omega_D\left(\frac{1}{np_D}+\frac{\omega_P\beta}{np_D}+\frac{\omega_Pp_P}{np_D}\right)$$

$$= 768\gamma\left(\frac{\omega_D\omega_P\beta}{np_D}+\frac{\omega_D(1+\omega_Pp_P)}{np_D}\right)$$

since $\frac{(\omega_P+1)\beta^2}{np_D}\leq\frac{1}{np_D}+\frac{\omega_P\beta}{np_D}$. Substituting these inequalities to (34) and (35), we get

$$\mathbb{E}\left[\delta^{t+1}\right]+\kappa\mathbb{E}\left[\left\|g^{t+1}-\frac{1}{n}\sum_{i=1}^{n}\nabla f_i(z_i^{t+1})\right\|^2\right]+\eta\mathbb{E}\left[\left\|z^{t+1}-w^{t+1}\right\|^2\right]$$

$$+\nu\mathbb{E}\left[\frac{1}{n}\sum_{i=1}^{n}\left\|z_i^{t+1}-w_i^{t+1}\right\|^2\right]+\rho\mathbb{E}\left[\left\|w^{t+1}-x^{t+1}\right\|^2\right]+\tau\mathbb{E}\left[\frac{1}{n}\sum_{i=1}^{n}\left\|w_i^{t+1}-x^{t+1}\right\|^2\right]$$

$$\leq\mathbb{E}\left[\delta^t\right]-\frac{\gamma}{2}\mathbb{E}\left[\left\|\nabla f(x^t)\right\|^2\right]-\left(\frac{1}{2\gamma}-\frac{L}{2}\right)\mathbb{E}\left[\left\|x^{t+1}-x^t\right\|^2\right]$$

$$+\kappa\left(1-\frac{p_D}{2}\right)\mathbb{E}\left[\left\|g^t-\frac{1}{n}\sum_{i=1}^{n}\nabla f_i(z_i^t)\right\|^2\right]+\eta\left(1-\frac{\beta}{4}\right)\mathbb{E}\left[\left\|z^t-w^t\right\|^2\right]$$

$$+\rho\left(1-\frac{p_P}{2}\right)\mathbb{E}\left[\left\|w^t-x^t\right\|^2\right]+\nu\left(1-\frac{\beta}{4}\right)\mathbb{E}\left[\frac{1}{n}\sum_{i=1}^{n}\left\|z_i^t-w_i^t\right\|^2\right]$$

$$+\tau\left(1-\frac{p_P}{2}\right)\mathbb{E}\left[\frac{1}{n}\sum_{i=1}^{n}\left\|w_i^t-x^t\right\|^2\right]$$

$$+768\gamma\Bigg(\left(\frac{\theta}{p_P}+\frac{1+\theta p_P}{\beta^2}\right)L_B^2+\left(\frac{\omega_P}{p_P}+\frac{1+\omega_Pp_P}{\beta^2}\right)L_A^2$$

$$+\left(\frac{\omega_D\omega_P\beta}{np_D}+\frac{\omega_D(1+\omega_Pp_P)}{np_D}\right)L_{\max}^2\Bigg)\mathbb{E}\left[\left\|x^{t+1}-x^t\right\|^2\right].$$

By collecting all the terms w.r.t. $\mathbb{E}\left[\left\|x^{t+1}-x^t\right\|^2\right]$, using the step size $\gamma$ from the theorem and Lemma H.2, we obtain

$$\mathbb{E}\left[\Psi^{t+1}\right]=\mathbb{E}\left[\delta^{t+1}\right]+\kappa\mathbb{E}\left[\left\|g^{t+1}-\frac{1}{n}\sum_{i=1}^{n}\nabla f_i(z_i^{t+1})\right\|^2\right]+\eta\mathbb{E}\left[\left\|z^{t+1}-w^{t+1}\right\|^2\right]$$

$$+\nu\mathbb{E}\left[\frac{1}{n}\sum_{i=1}^{n}\left\|z_i^{t+1}-w_i^{t+1}\right\|^2\right]+\rho\mathbb{E}\left[\left\|w^{t+1}-x^{t+1}\right\|^2\right]$$

$$+\tau\mathbb{E}\left[\frac{1}{n}\sum_{i=1}^{n}\left\|w_i^{t+1}-x^{t+1}\right\|^2\right]$$

$$\leq\mathbb{E}\left[\delta^t\right]-\frac{\gamma}{2}\mathbb{E}\left[\left\|\nabla f(x^t)\right\|^2\right]+\kappa\left(1-\frac{p_D}{2}\right)\mathbb{E}\left[\left\|g^t-\frac{1}{n}\sum_{i=1}^{n}\nabla f_i(z_i^t)\right\|^2\right]$$

$$+\eta\left(1-\frac{\beta}{4}\right)\mathbb{E}\left[\left\|z^t-w^t\right\|^2\right]+\rho\left(1-\frac{p_P}{2}\right)\mathbb{E}\left[\left\|w^t-x^t\right\|^2\right]$$

$$+\nu\left(1-\frac{\beta}{4}\right)\mathbb{E}\left[\frac{1}{n}\sum_{i=1}^{n}\left\|z_i^t-w_i^t\right\|^2\right]+\tau\left(1-\frac{p_P}{2}\right)\mathbb{E}\left[\frac{1}{n}\sum_{i=1}^{n}\left\|w_i^t-x^t\right\|^2\right].$$

Lastly, Assumption D.9 gives

$$
\begin{aligned}
\mathbb{E}\left[\Psi^{t+1}\right] \quad = \quad & \mathbb{E}\left[\delta^{t+1}\right] + \kappa\mathbb{E}\left[\left\|g^{t+1} - \frac{1}{n}\sum_{i=1}^{n}\nabla f_i(z_i^{t+1})\right\|^2\right] + \eta\mathbb{E}\left[\left\|z^{t+1} - w^{t+1}\right\|^2\right] \\
& + \nu\mathbb{E}\left[\frac{1}{n}\sum_{i=1}^{n}\left\|z_i^{t+1} - w_i^{t+1}\right\|^2\right] + \rho\mathbb{E}\left[\left\|w^{t+1} - x^{t+1}\right\|^2\right] \\
& + \tau\mathbb{E}\left[\frac{1}{n}\sum_{i=1}^{n}\left\|w_i^{t+1} - x^{t+1}\right\|^2\right] \\
\overset{\text{Ass.}D.9,(33)}{\leq} \quad & (1-\gamma\mu)\,\mathbb{E}\left[\delta^t\right] + \kappa\,(1-\gamma\mu)\,\mathbb{E}\left[\left\|g^t - \frac{1}{n}\sum_{i=1}^{n}\nabla f_i(z_i^t)\right\|^2\right] \\
& + \eta\,(1-\gamma\mu)\,\mathbb{E}\left[\left\|z^t - w^t\right\|^2\right] + \nu\,(1-\gamma\mu)\,\mathbb{E}\left[\frac{1}{n}\sum_{i=1}^{n}\left\|z_i^t - w_i^t\right\|^2\right] \\
& + \rho\,(1-\gamma\mu)\,\mathbb{E}\left[\left\|w^t - x^t\right\|^2\right] + \tau\,(1-\gamma\mu)\,\mathbb{E}\left[\frac{1}{n}\sum_{i=1}^{n}\left\|w_i^t - x^t\right\|^2\right] \\
= \quad & (1-\gamma\mu)\,\mathbb{E}\left[\Psi^t\right]
\end{aligned}
$$

It remains to apply the last inequality iteratively to finish the proof. $\qquad\square$

**Corollary E.9.** *Let $\mathcal{C}_i^t \in \mathbb{P}(0)$ for all $i \in [n]$ (e.g. PermK), choose $p_P = 1/(\omega_P + 1)$, $p_D = 1/(\omega_D + 1)$ and*

$$
\beta = \min\left\{\left(\frac{n}{\omega_D\omega_P(\omega_D + 1)}\right)^{1/3}, 1\right\}.
$$

*Then, in the view of Theorem E.8, Algorithm 2 ensures that $\mathbb{E}\left[f(x^T) - f^*\right] \leq \varepsilon$ after*

$$
\mathcal{O}\left(\max\left\{\frac{\left(1 + \left(\frac{\omega_D\omega_P(\omega_D+1)}{n}\right)^{1/3} + \sqrt{\frac{\omega_D(\omega_D+1)}{n}}\right)L_{\max} + \sqrt{\omega_P(\omega_P+1)}L_A}{\mu}, \omega_P + 1, \omega_D + 1, \left(\frac{\omega_D\omega_P(\omega_D+1)}{n}\right)^{1/3}\right\}\log\frac{\Psi^0}{\varepsilon}\right)
$$

*iterations.*

*Proof.* Note that $\Psi^T \geq f(x^T) - f^*$. In view of condition (33) from Theorem E.8, the step size satisfies

$$
\gamma = \Theta\left(\min\left\{\left(L + \sqrt{\left(\frac{\theta}{p_P} + \frac{1+\theta p_P}{\beta^2}\right)L_B^2 + \left(\frac{\omega_P}{p_P} + \frac{1+\omega_P p_P}{\beta^2}\right)L_A^2 + \left(\frac{\omega_D\omega_P\beta}{np_D} + \frac{\omega_D(1+\omega_P p_P)}{np_D}\right)L_{\max}^2}\right)^{-1},\right.\right.
$$
$$
\left.\left.\frac{p_P}{2\mu}, \frac{p_D}{2\mu}, \frac{\beta}{4\mu}\right\}\right).
$$

Therefore, since $\theta = 0$, the algorithm converges after

$$
\bar{T} = \mathcal{O}\left(\max\left\{\frac{L + \sqrt{\frac{1}{\beta^2}L_B^2 + \left(\frac{\omega_P}{p_P} + \frac{1+\omega_P p_P}{\beta^2}\right)L_A^2 + \left(\frac{\omega_D\omega_P\beta}{np_D} + \frac{\omega_D(1+\omega_P p_P)}{np_D}\right)L_{\max}^2}}{\mu}, \frac{1}{p_P}, \frac{1}{p_D}, \frac{1}{\beta}\right\}\log\frac{\Psi^0}{\varepsilon}\right)
$$

iterations. Using the choice of $p_P$ and $p_D$, we have

$$
\bar{T} = \mathcal{O}\left(\max\left\{\frac{L + \sqrt{\frac{1}{\beta^2}(L_B^2 + L_A^2) + \omega_P(\omega_P+1)L_A^2 + \left(\frac{\omega_D(\omega_D+1)\omega_P\beta}{n} + \frac{\omega_D(\omega_D+1)}{n}\right)L_{\max}^2}}{\mu}, \omega_P + 1, \omega_D + 1, \frac{1}{\beta}\right\}\log\frac{\Psi^0}{\varepsilon}\right).
$$

Due to Lemma C.1, we get

$$\bar{T} = \mathcal{O}\left(\max\left\{\frac{L + \sqrt{\omega_P(\omega_P+1)L_A^2 + \left(\frac{1}{\beta^2} + \frac{\omega_D(\omega_D+1)\omega_P\beta}{n} + \frac{\omega_D(\omega_D+1)}{n}\right)L_{\max}^2}}{\mu}, \omega_P + 1, \omega_D + 1, \frac{1}{\beta}\right\}\log\frac{\Psi^0}{\varepsilon}\right).$$

Using the choice of $\beta$, we obtain the result of the theorem. $\qquad\square$

**Corollary E.10.** *Let $\mathcal{C}_i^t$ be the PermK compressors and $\mathcal{Q}_i^t$ be the independent (Assumption 1.6) RandK compressors, both with $K = d/n$. Then, in the view of Corollary E.9, the total communication complexity is*

$$\mathcal{O}\left(\left(\frac{dL_{\max}}{n^{1/3}\mu} + \frac{dL_A}{\mu} + d\right)\log\frac{\Psi^0}{\varepsilon}\right).$$

*Proof.* The choice of compressors and parameters ensures that $\omega_P = \omega_D = n - 1$ (Lemma A.6). Thus, the iteration complexity is

$$\mathcal{O}\left(\max\left\{\frac{\left(1 + n^{2/3} + n^{1/2}\right)L_{\max} + nL_A}{\mu}, n, n, n^{2/3}\right\}\log\frac{\Psi^0}{\varepsilon}\right)$$

$$= \mathcal{O}\left(\max\left\{\frac{n^{2/3}L_{\max} + nL_A}{\mu}, n\right\}\log\frac{\Psi^0}{\varepsilon}\right).$$

Since $p_P = p_D = 1/n$ and $K = d/n$, on average, the algorithm sends $\leq 2d/n$ coordinates in both directions. Therefore, the total communication complexity is

$$\mathcal{O}\left(\frac{d}{n} \times \max\left\{\frac{n^{2/3}L_{\max} + nL_A}{\mu}, n\right\}\log\frac{\Psi^0}{\varepsilon}\right) = \mathcal{O}\left(\max\left\{\frac{\frac{d}{n^{1/3}}L_{\max} + dL_A}{\mu}, d\right\}\log\frac{\Psi^0}{\varepsilon}\right).$$

$\qquad\square$

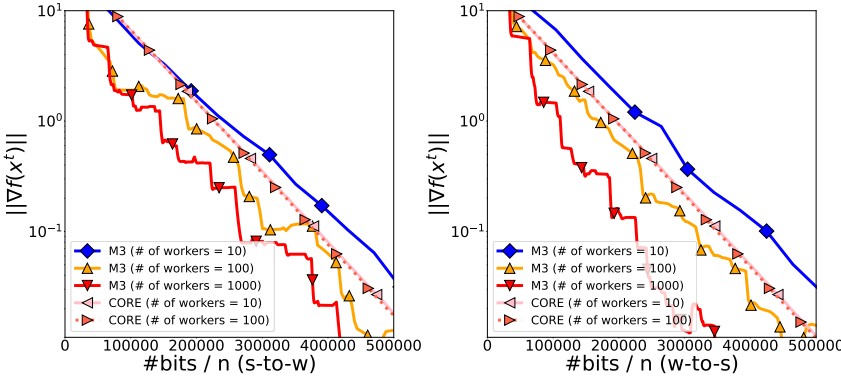

Figure 2: Experiments on the quadratic optimization problem from Section F.1. We plot the norm of the gradient w.r.t. # of coordinates sent from the server (s-to-w) and from the workers (w-to-s).

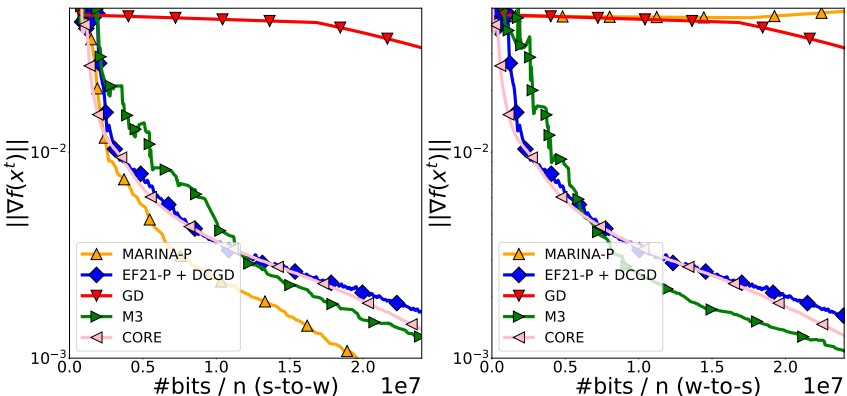

Figure 3: Experiments on the autoencoder task from Section F.2. We plot the norm of the gradient w.r.t. # of coordinates sent from the server (s-to-w) and from the workers (w-to-s).

# F  Experiments

The experiments were prepared in Python. The distributed environment was emulated on a machine with Intel(R) Xeon(R) Gold 6226R CPU @ 2.90GHz and 64 cores.

## F.1  Experiments with M3 on quadratic optimization tasks

We consider close-to-homogeneous quadratic optimization problem with $\mathbf{A}_i = (1 + \xi_i)\mathbf{I}_d$, where $\xi_i \sim \mathcal{N}(0, 0.01)$ for all $i \in [n]$, and $d = 1000$. We run two algorithms from Table 1, M3 and CORE, and check whether theory matches practice. In M3, we use PermK followed by the natural compressor $\mathcal{C}_{\text{nat}}$ (Horváth et al., 2022) (composition of two unbiased compressors) on the server's side, and RandK followed by $\mathcal{C}_{\text{nat}}$ on the workers' side. We use $K = \lfloor d/n \rfloor \in \{1, 10, 100\}$ for $n \in \{1000, 100, 10\}$. In CORE, the number of communicated coordinates is set to 10. We run each experiment 5 times with different seeds and plot the average to reduce the noise factor. Only the step size is fine-tuned for each algorithm.

The results are presented in Figure 2. As expected, CORE does not change its behavior as the number of workers increases from 10 to 100; this is expected since CORE does not depend on $n$. At the same time, M3 does improve with $n$, which supports our findings from Theorem 5.1.

## F.2  Experiments with an autoencoder and MNIST

We now compare MARINA-P, M3, CORE, EF21-P + DCGD, and GD on a non-convex autoencoder problem. We train it on the MNIST dataset (LeCun et al., 2010) with objective function $f(\mathbf{D}, \mathbf{E}) := \frac{1}{m}\sum_{i=1}^{m}\|\mathbf{D}\mathbf{E}b_i - b_i\|^2 + \frac{\lambda}{2}\|\mathbf{D}\mathbf{E} - \mathbf{I}\|_F^2$, where $\mathbf{D} \in \mathbb{R}^{d_1 \times d_2}$, $\mathbf{E} \in \mathbb{R}^{d_2 \times d_1}$, $b_i \in \mathbb{R}^{d_1}$ are samples,

$d_1 = 784$ is the number of features, $d_2 = 16$ is the size of the encoding space, $\lambda = 0.001$ is a regularizer, and $m = 60\,000$ is the number of samples. The dimension of the problem is $d = 25\,088$. We randomly split the dataset among $n = 100$ workers. For MARINA-P and M3, we take $\mathrm{Perm}K$ followed by the natural compressor $\mathcal{C}_{\mathrm{nat}}$ on the server's side. On the workers' side, M3 uses $\mathrm{Rand}K$ and $\mathcal{C}_{\mathrm{nat}}$. For EF21-P + DCGD, we take $\mathrm{Rand}K$ with $\mathcal{C}_{\mathrm{nat}}$ on both the workers' and server's sides. In each case, $K = \lfloor d/n \rfloor = 250$. For CORE, we set the number of communicated coordinates to 100. As in previous experiments, we only fine-tune the step size, repeat each experiment 5 times, and plot the average results.

The results are presented in Figure 3. All methods with bidirectional compression: M3, CORE, and EF21-P + DCGD, converge much faster than GD. MARINA-P converges fastest only in the first plot. This is expected since it compresses only from the server to the workers. M3, CORE, and EF21-P + DCGD have similar convergence rates in both metrics, with M3 performing better in the low accuracy regime.

### F.3 Extra experiments with quadratic optimization tasks

The aim of this set of experiments is to empirically test our results under Assumption 4.2. We consider the problem of quadratic minimization with varying level of heterogeneity between the $n$ functions stored on the workers. The goal is to minimize the squared norm of the gradient of $\sum_{i=1}^{n} f_i$, where the functions $f_i$ are of form

$$f_i(x) = \frac{1}{2} x^T \mathbf{A}_i x + b_i^T x.$$

Here, $\mathbf{A}_i$ are $d \times d$ matrices generated following the procedure in Algorithm 3, and $b_i$ denotes a standard normal vector in $\mathbb{R}^d$. The constants $L_A$ and $L_B$ from Assumption 4.2 (in this case, by Theorem 4.8 $L_A = \sqrt{2} \max_{i \in [n]} \|\mathbf{A}_i - \mathbf{A}\|$ and $L_B = \sqrt{2} \left( \frac{1}{n} \sum_{i=1}^{n} \|\mathbf{A}_i\| \right)$) are controlled by parameters $v_i$ and $\sigma_i^2$. In particular, for $\sigma_i^2 = 0$, all workers hold the same matrix $\mathbf{A}_i$, and hence in this case $L_A = 0$.

We compare the following algorithms:

1. MARINA-P with $\mathrm{Perm}K$ compressors,
2. MARINA-P with $\mathrm{Rand}K$ compressors,
3. MARINA-P with $\mathrm{SameRand}K$ compressor,
4. EF21-P with $\mathrm{Top}K$ compressor,
5. GD.

In all compressed methods, we set $K = d/n$ and use $p = k/d$ in MARINA-P.

The step sizes are tuned from $2^i, i \in \mathbb{Z}$ multiples of the values predicted by the theory (indicated by $\times 1, \times 2, \dots$ in the plots). We fix $d = 300$ and generate optimization tasks with $n \in \{10, 100, 900\}$. The results are presented in Figures 4, 5, 6.

The empirical results align well with the theory. Among the algorithms tested, MARINA-P with $\mathrm{Perm}K$ compressor exhibits the best performance, while MARINA-P with $\mathrm{SameRand}K$ converges the slowest and comparable to GD. MARINA-P with $\mathrm{Rand}K$ compressor and EF21-P achieve performance levels somewhere in between. Notably, the differences between the runs of MARINA-P with different compressors become more pronounced as the value of $n$ increases. As anticipated, the performance of MARINA-P with $\mathrm{Rand}K$ and $\mathrm{Perm}K$ compressors improves with an increase in the number of workers, while the performance of EF21-P does not follow the same behaviour. Specifically, for $n = 10$, EF21-P outperforms MARINA-P with $\mathrm{Rand}K$ compressor, but this pattern reverses for both $n = 100$ and $n = 1000$.

**Algorithm 3** Heterogeneous quadratic problem generation

1: **Parameters:** $v_0, \ldots, v_4 \in \mathbb{R}_+$, $\sigma_0, \ldots, \sigma_4 \in \mathbb{R}_{\geq 0}$.
2: Let

$$\mathbf{X} = \frac{1}{4} \begin{bmatrix} 2 & -1 & & 0 \\ -1 & \ddots & \ddots & \\ & \ddots & \ddots & -1 \\ 0 & & -1 & 2 \end{bmatrix} \in \mathbb{R}^{300 \times 300},$$

3: **for** $k = 0, \ldots, 4$ **do**
4:     Generate $\xi_i \sim \mathcal{N}(0, \sigma_k^2) \cap [-v_0, v_0]$ for $i \in [n]$
5:     **for** $l = 0, \ldots, 4$ **do**
6:         Set $\mathbf{A}_i^{k,l} = (v_l + \xi_i)\mathbf{X}$ for $i \in [n]$
7:         Sample $b_i^{k,l} \sim \mathcal{N}(0, \mathbb{I}_d)$ for $i \in [n]$
8:     **end for**
9:     **Output:** matrices $\mathbf{A}_i^{k,l}$, vectors $b_i^{k,l}$, $i \in [n]$, $k, l \in [4]$.
10: **end for**

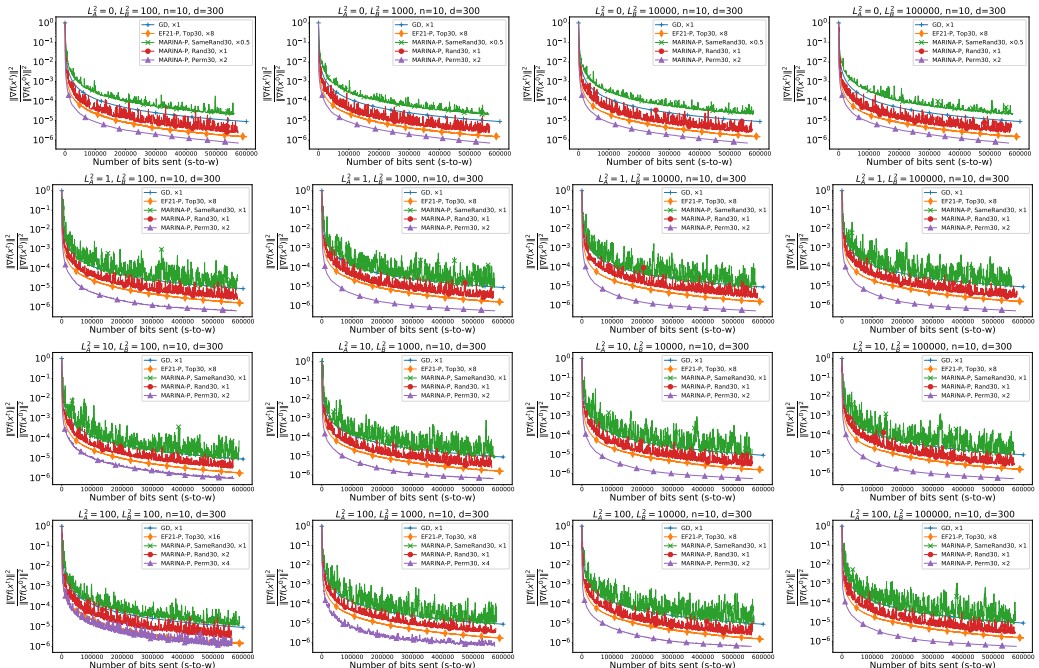

Figure 4: Experiments on the quadratic optimization problem from Section F.3 with $n = 10$ for $L_A^2 \in \{0, 1, 10, 100\}$ and $L_B^2 \in \{100, 1000, 10000, 100000\}$.

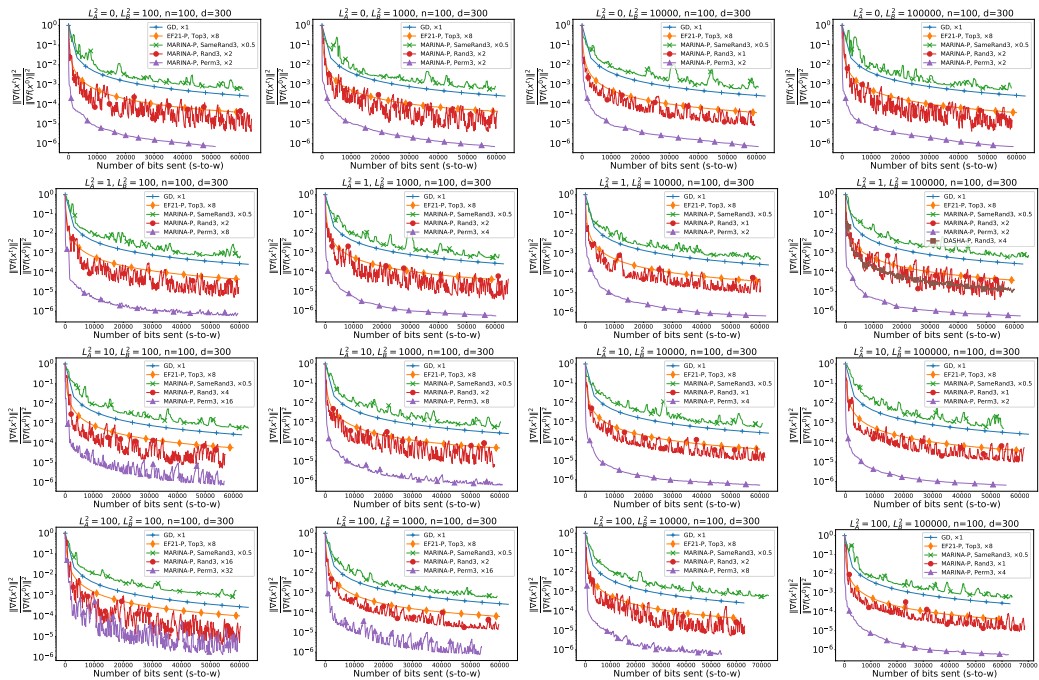

Figure 5: Experiments on the quadratic optimization problem from Section F.3 with $n = 100$ for $L_A^2 \in \{0, 1, 10, 100\}$ and $L_B^2 \in \{100, 1000, 10000, 100000\}$.

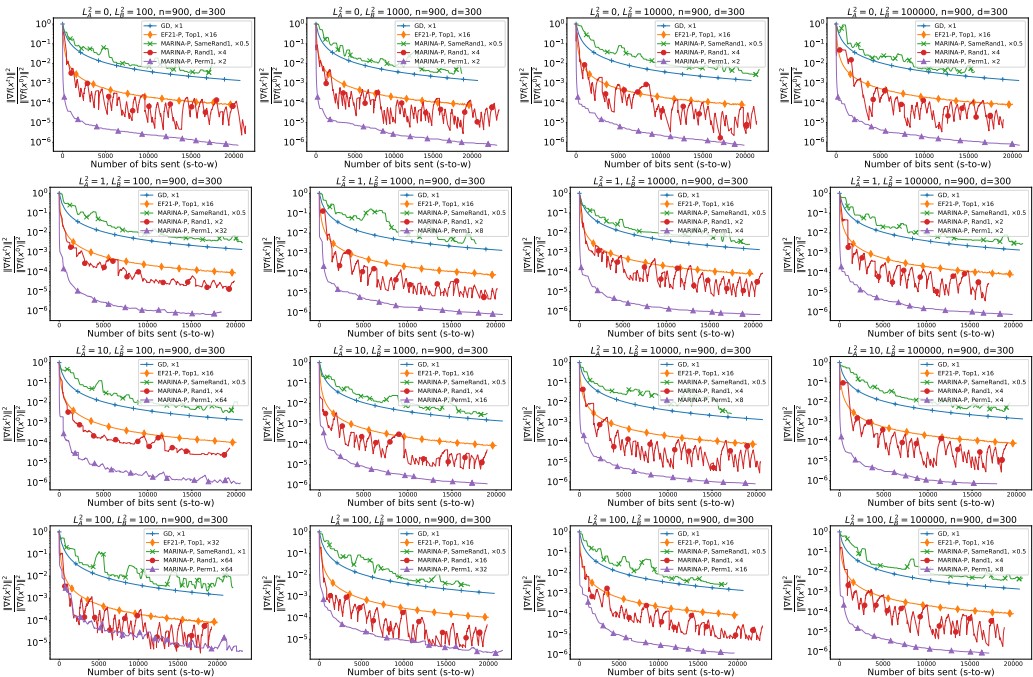

Figure 6: Experiments on the quadratic optimization problem from Section F.3 with $n = 900$ for $L_A^2 \in \{0, 1, 10, 100\}$ and $L_B^2 \in \{100, 1000, 10000, 100000\}$.

# G Proof of the Lower Bounds

## G.1 The "difficult" function from the nonconvex world

In our lower bound, we use the function from Carmon et al. (2020); Arjevani et al. (2022). For any $T \in \mathbb{N}$, let

$$F_T(x) := -\Psi(1)\Phi([x]_1) + \sum_{i=2}^{T} \left(\Psi(-[x]_{i-1})\Phi(-[x]_i) - \Psi([x]_{i-1})\Phi([x]_i)\right), \qquad (36)$$

where

$$\Psi(x) = \begin{cases} 0, & x \leq 1/2, \\ \exp\left(1 - \frac{1}{(2x-1)^2}\right), & x \geq 1/2, \end{cases} \quad \text{and} \quad \Phi(x) = \sqrt{e} \int_{-\infty}^{x} e^{-\frac{1}{2}t^2} \, dt.$$

Carmon et al. (2020); Arjevani et al. (2022) also proved the following properties of the function:

**Lemma G.1** (Carmon et al. (2020); Arjevani et al. (2022)). *The function $F_T$ satisfies:*

1. *$F_T(0) - \inf_{x \in \mathbb{R}^T} F_T(x) \leq \Delta^0 T$, where $\Delta^0 = 12$.*

2. *The function $F_T$ is $l_1$–smooth, where $l_1 = 152$.*

3. *For all $x \in \mathbb{R}^T$, $\|\nabla F_T(x)\|_\infty \leq \gamma_\infty$, where $\gamma_\infty = 23$.*

4. *For all $x \in \mathbb{R}^T$, $\text{prog}(\nabla F_T(x)) \leq \text{prog}(x) + 1$.*

5. *For all $x \in \mathbb{R}^T$, if $\text{prog}(x) < T$, then $\|\nabla F_T(x)\| > 1$,*

where $\text{prog}(x) := \max\{i \geq 0 \,|\, x_i \neq 0\}$ $(x_0 \equiv 1)$.

The function is a standard function that is used to establish lower bounds in the nonconvex world (Carmon et al., 2020; Arjevani et al., 2022; Lu and De Sa, 2021; Tyurin and Richtárik, 2023c).

## G.2 Theorems

Our lower bound applies to the family of methods with the following structure:

---
**Protocol 4** Protocol
---
1: **Input:** functions $f_1, \ldots, f_n \in \mathcal{F}$, algorithm $A$, probability $p$
2: **for** $k = 0, \ldots, \infty$ **do**
3:     Server calculates a new point: $x^k = B_1^k(g_1^1, \ldots, g_1^k, \ldots, g_n^1, \ldots, g_n^k)$
4:     Server aggregates all available information: $s_i^k = B_{2,i}^k(g_1^1, \ldots, g_1^k, \ldots, g_n^1, \ldots, g_n^k)$
5:     Server sends sparsified vectors $\bar{s}_i^k$ to the workers, where

$$[\bar{s}_i^k]_j = [s_i^k]_j \times \eta_{i,j}^k,$$

    and $\eta_{i,j}^k$ is a random variable such that $\mathbb{P}\left(\eta_{i,j}^k \neq 0\right) \leq p$ for all $j \in [d']$ and for all $i \in [n]$. We define $d' := \dim(\text{dom}(f_1))$, and $[\cdot]_j$ means the $j^{\text{th}}$ coordinate.
6:     Workers aggregate all available local information and calculate gradients: $g_i^{k+1} = L_i^k(\bar{s}_i^0, \ldots, \bar{s}_i^k)$
    ($L_i^k$ has access to the gradient oracle of $f_i$ and can call it as many times as it wants according to the rules (37) and (38))
7:     Workers send $g_i^{k+1}$ to the server
8: **end for**

---

We consider the following standard classes of functions and algorithms:

**Definition G.2.** Let the function $f : \mathbb{R}^d \to \mathbb{R}$ be differentiable, $L$-smooth (i.e., $\|\nabla f(x) - \nabla f(y)\| \leq L \|x - y\|$ for all $x, y \in \mathbb{R}^d$), and $f(0) - \inf_{x \in \mathbb{R}^d} f(x) \leq \delta^0$. We denote the family of functions that satisfy these properties by $\mathcal{F}_{\delta^0, L}$.

**Definition G.3.** Consider Protocol 4. A sequence of tuples of mappings $A = \{(B_1^k, B_{2,1}^k, \ldots, B_{2,n}^k, L_1^k, \ldots, L_n^k)\}_{k=0}^{\infty}$ is a zero-respecting algorithm, if,

1. $B_1^k : \underbrace{\mathbb{R}^d \times \cdots \times \mathbb{R}^d}_{n \times k \text{ times}} \to \mathbb{R}^d$ for all $k \geq 1$, and $B_1^0 \in \mathbb{R}^d$.

2. $B_{2,i}^k : \underbrace{\mathbb{R}^d \times \cdots \times \mathbb{R}^d}_{n \times k \text{ times}} \to \mathbb{R}^d$ for all $k \geq 1$, and $B_{2,i}^0 \in \mathbb{R}^d$ for all $i \in [n]$

3. $L_i^k : \underbrace{\mathbb{R}^d \times \cdots \times \mathbb{R}^d}_{k+1 \text{ times}} \to \mathbb{R}^d$ for all $k \geq 0$ and for all $i \in [n]$.

4. $\operatorname{supp}\left(x^k\right) \subseteq \bigcup_{j=1}^{k} \bigcup_{i=1}^{n} \operatorname{supp}\left(g_i^j\right)$, $\operatorname{supp}\left(s_i^k\right) \subseteq \bigcup_{j=1}^{k} \bigcup_{i=1}^{n} \operatorname{supp}\left(g_i^j\right)$.
For all $\hat{g}_{i,1}^{k+1}, \hat{g}_{i,2}^{k+1}, \ldots$ such that

$$\operatorname{supp}\left(\hat{g}_{i,1}^{k+1}\right) \subseteq \bigcup_{j=0}^{k} \operatorname{supp}\left(\bar{s}_i^j\right),$$

$$\operatorname{supp}\left(\hat{g}_{i,2}^{k+1}\right) \subseteq \bigcup_{j=0}^{k} \operatorname{supp}\left(\bar{s}_i^j\right) \bigcup \operatorname{supp}(\nabla f_i(\hat{g}_{i,1}^{k+1})),$$

$$\operatorname{supp}\left(\hat{g}_{i,3}^{k+1}\right) \subseteq \bigcup_{j=0}^{k} \operatorname{supp}\left(\bar{s}_i^j\right) \bigcup \operatorname{supp}(\nabla f_i(\hat{g}_{i,1}^{k+1})) \bigcup \operatorname{supp}(\nabla f_i(\hat{g}_{i,2}^{k+1})),$$

$$\ldots$$

we have

$$\operatorname{supp}\left(g_i^{k+1}\right) \subseteq \bigcup_{j=1}^{\infty} \operatorname{supp}\left(\hat{g}_{i,j}^{k+1}\right), \tag{38}$$

for all $k \in \mathbb{N}_0$ and for all $i \in [n]$, where $\operatorname{supp}(x) := \{i \in [d] \mid x_i \neq 0\}$.

We denote the set of all algorithms that satisfy these properties by $\mathcal{A}_{\text{zr}}$.

The first three properties define the domains of the mapping. The last property is a standard assumption for a zero-respecting algorithm. Assumption (38) allows the mappings $L_i^k$ to calculate gradients.

**Theorem G.4.** *Consider Protocol 4. Assume that the sets* $\{\eta_{i,j}^0\}_{i\in[n],j\in[d']}$, $\{\eta_{i,j}^1\}_{i\in[n],j\in[d']}, \ldots, \{\eta_{i,j}^k\}_{i\in[n],j\in[d']}, \ldots$ *are mutually independent (the variables within one set can be dependent). Let* $p > 0, L, \delta^0, \varepsilon > 0, n \geq 2$ *be any numbers such that* $\bar{c}\varepsilon < L\delta^0$. *Then, for any algorithm* $A \in \mathcal{A}_{\text{zr}}$, *there exists a function* $f \in \mathcal{F}_{\delta^0,L}$ *and functions* $f_1, \ldots, f_n$ *such that* $f = \frac{1}{n}\sum_{i=1}^{n} f_i$ *and* $\mathbb{E}\left[\left\|\nabla f(x^k)\right\|^2\right] > \varepsilon$ *for all*

$$k \leq \hat{c}\frac{L\delta^0}{p\varepsilon}.$$

*The quantities* $\bar{c}$ *and* $\hat{c}$ *are universal constants.*

*Proof.* The proof is conceptually the same as in Arjevani et al. (2022); Lu and De Sa (2021); Huang et al. (2022); Fang et al. (2018); Carmon et al. (2020); Tyurin and Richtárik (2023c). We fix $\lambda > 0$, and consider the following function $f : \mathbb{R}^T \to \mathbb{R}$:

$$f(x) := \frac{L\lambda^2}{l_1} F_T\left(\frac{x}{\lambda}\right).$$

One can show (Arjevani et al., 2022)[Theorem 1] that $f \in \mathcal{F}_{\delta^0,L}$ if

$$T = \left\lfloor \frac{\delta^0 l_1}{L\lambda^2 \Delta^0} \right\rfloor. \tag{39}$$

Next, we define

$$F_i(x) := \begin{cases} -\Psi(1)\Phi([x]_1) + \sum_{2 \le j \le T \text{ and } (j-1) \bmod n = 0} \left(\Psi(-[x]_{j-1})\Phi(-[x]_j) - \Psi([x]_{j-1})\Phi([x]_j)\right), & i = 1 \\ \sum_{2 \le j \le T \text{ and } (j-1) \bmod n = i-1}^{T} \left(\Psi(-[x]_{j-1})\Phi(-[x]_j) - \Psi([x]_{j-1})\Phi([x]_j)\right), & i > 1 \end{cases}$$

and

$$f_i(x) := \frac{nL\lambda^2}{l_1} F_i\left(\frac{x}{\lambda}\right).$$

The idea is that we take the first block from (36) to the first worker, the second block to the second worker, $\ldots$, $(n+1)^{\text{th}}$ block to the first worker, and so on. Then, one can show that

$$\frac{1}{n}\sum_{i=1}^{n} f_i(x) = f(x).$$

Using Lemma G.1, we obtain

$$\|\nabla f(x)\|^2 = \frac{L^2\lambda^2}{l_1^2}\left\|\nabla F_T\left(\frac{x}{\lambda}\right)\right\|^2 > \frac{L^2\lambda^2}{l_1^2}\mathbb{1}[\text{prog}(x) < T]. \tag{40}$$

The functions $f_i$ are *zero-chain* (Arjevani et al., 2022): for all $i \in [n]$, if $\text{prog}(x) = j$ and $(j \bmod n) + 1 = i$, then $\text{prog}(\nabla f_i(x)) \le j+1$, and for all $i \in [n]$, if $\text{prog}(x) = j$ and $(j \bmod n) + 1 \ne i$, then $\text{prog}(\nabla f_i(x)) \le j$. Using the zero-chain property and the fact that we consider the family of zero-respecting algorithms:

1. The first non-zero coordinate can be discovered only by the first worker.

2. Assume that $\max_{j=1}^{k} \max_{i=1}^{n} \text{prog}\left(g_i^j\right) = j \ge 1$. An algorithm can discover *one* new non-zero coordinate in the $(j+1)^{\text{th}}$ position only if the $(j \bmod n + 1)^{\text{th}}$ worker gets a *non-zero* $j^{\text{th}}$ coordinate from *the server*. This is by the construction of the functions $f_i$. Note that for $n \ge 2$, one worker cannot discover two *consecutive* coordinates.

Let us define

$$\xi^j = \mathbb{I}[\text{In the } j^{\text{th}} \text{ iteration, the coordinate with index } \bar{p} \equiv \max_{j=1}^{k} \max_{i=1}^{n} \text{prog}\left(g_i^j\right) \text{ is not zeroed out in Line 5}$$

of Protocol 4 to the worker with index $(\bar{p} \bmod n + 1)$ AND $T - 1 \ge \bar{p} \ge 1]$  $(\bar{p} = 0 \text{ if } k = 0)$.

Then, we have

$$\mathbb{P}\left(\text{prog}(x^k) \ge T\right) \le \mathbb{P}\left(\sum_{j=0}^{k-1} \xi^j \ge T - 1\right).$$

Assume that $\mathcal{G}_j$ is the $\sigma$–algebra generated by all randomness up to the $j^{\text{th}}$ iteration (inclusive). Then, $\xi^j$ is $\mathcal{G}_j$–measurable, and, by the construction of Line 5 of Protocol 4, $\mathbb{P}\left(\xi^{j+1} = 1 \big| \mathcal{G}_j\right) \le p$, where we also use the assumption of the theorem that the sets of random variables are mutually independent. Using the standard approach with the Chernoff method (Arjevani et al., 2022; Lu and De Sa, 2021; Huang et al., 2022), one can show that

$$\mathbb{P}\left(\sum_{j=0}^{k-1} \xi^j \ge T - 1\right) \le \rho$$

for all

$$k \le \frac{T - 1 - \log\frac{1}{\rho}}{2p}$$

and $\rho \in (0, 1]$. Therefore, we get

$$\mathbb{P}\left(\text{prog}(x^k) \ge T\right) \le \rho \tag{41}$$

for all

$$k \leq \frac{T - 1 - \log \frac{1}{\rho}}{2p}.$$

Using (40), we have

$$\mathbb{E}\left[\left\|\nabla f(x^k)\right\|^2\right] > 2\varepsilon \mathbb{P}\left(\left\|\nabla f(x^k)\right\|^2 > 2\varepsilon\right) \geq 2\varepsilon \mathbb{P}\left(\frac{L^2\lambda^2}{l_1^2}\mathbb{1}[\text{prog}(x) < T] \geq 2\varepsilon\right).$$

Let us take $\lambda = \frac{\sqrt{2\varepsilon}l_1}{L}$. Then

$$\mathbb{E}\left[\left\|\nabla f(x^k)\right\|^2\right] > 2\varepsilon \mathbb{P}\left(\frac{L^2\lambda^2}{l_1^2}\mathbb{1}[\text{prog}(x) < T] \geq 2\varepsilon\right) = 2\varepsilon \mathbb{P}\left(\text{prog}(x) < T\right). \qquad (42)$$

From (41) with $\rho = \frac{1}{2}$, we get

$$\mathbb{E}\left[\left\|\nabla f(x^k)\right\|^2\right] > 2\varepsilon \mathbb{P}\left(\text{prog}(x) < T\right) \geq \varepsilon \qquad (43)$$

for all

$$k \leq \frac{T - 1 - \log 2}{2p}.$$

From (39), one can conclude that

$$T = \left\lfloor \frac{L\delta^0 l_1}{2\varepsilon l_1^2 \Delta^0} \right\rfloor.$$

By the theorem's assumption, $L\delta^0 \geq \bar{c}\varepsilon$. One can choose a universal constant $\bar{c}$ such that (42) holds for

$$k \leq \Theta\left(\frac{T}{p}\right) = \Theta\left(\frac{L\delta^0}{\varepsilon p}\right),$$

where $\Theta$ hides only a universal constant. $\qquad \square$

### G.3 Compressed communication with independent compressors

Protocol 5 is exactly the same as Protocol 4 except for Line 5 and describes the family of methods that send compressed vectors from the server to the workers.

---

**Protocol 5** Protocol with Compressors

---

1: **Input:** functions $f_1, \ldots, f_n \in \mathcal{F}$, algorithm $A$, compressors $\mathcal{C}_1, \ldots, \mathcal{C}_n$
2: **for** $k = 0, \ldots, \infty$ **do**
3:     Server calculates a new point: $x^k = B_1^k(g_1^1, \ldots, g_1^k, \ldots, g_n^1, \ldots, g_n^k)$
4:     Server aggregates all available information: $s_i^k = B_{2,i}^k(g_1^1, \ldots, g_1^k, \ldots, g_n^1, \ldots, g_n^k)$
5:     Server sends compressed vectors $\bar{s}_i^k = \mathcal{C}_i(s_i^k)$ to the workers
6:     Workers aggregate all available local information and calculate gradients: $g_i^{k+1} = L_i^k(\bar{s}_i^0, \ldots, \bar{s}_i^k)$
    ($L_i^k$ has access to the gradient oracle of $f_i$ and can call it as many times as it wants according to the rules (37) and (38))
7:     Workers send $g_i^{k+1}$ to the server
8: **end for**

---

**Theorem G.5.** *Consider Protocol 5. Let $\omega \geq 0, L, \delta^0, \varepsilon > 0, n \geq 2$ be any numbers such that $\bar{c}\varepsilon < L\delta^0$. Then for any algorithm $A \in \mathcal{A}_{zr}$, there exists a function $f \in \mathcal{F}_{\delta^0, L}$, functions $f_1, \ldots, f_n$ such that $f = \frac{1}{n}\sum_{i=1}^n f_i$, and i.i.d. compressors $\mathcal{C}_1, \ldots, \mathcal{C}_n \in \mathbb{U}(\omega)$ such that $\mathbb{E}\left[\left\|\nabla f(x^k)\right\|^2\right] > \varepsilon$ for all*

$$k \leq \hat{c}\frac{(\omega + 1)L\delta^0}{\varepsilon}.$$

*The quantities $\bar{c}$ and $\hat{c}$ are universal constants.*

*Proof.* We can use the result of Theorem G.4. It is sufficient to construct an appropriate compressor. Let us define $p := 1/\omega+1$. We define the following compressor:

$$[\mathcal{C}(x)]_j := \begin{cases} \frac{1}{p}x_j, & j \in S, \\ 0, & j \notin S, \end{cases} \quad \forall j \in [T],$$

where $S$ is a random subset of $[T]$ and each element from $[T]$ appears with probability $p$ independently. Then, $\mathcal{C}$ is unbiased:

$$\mathbb{E}_S\left[[\mathcal{C}(x)]_j\right] = x_j \quad \forall j \in \mathbb{R}^T$$

and

$$\mathbb{E}_S\left[\|\mathcal{C}(x)\|^2\right] = \mathbb{E}_S\left[\sum_{j=1}^{nT} \mathbb{1}\left[j \in S\right]\frac{1}{p^2}x_j^2\right] = \sum_{j=1}^{nT}\mathbb{P}\left(j \in S\right)\frac{1}{p^2}x_j^2 = \sum_{j=1}^{nT}\frac{1}{p}x_j^2 = (\omega+1)\|x\|^2.$$

Therefore, we get $\mathcal{C} \in \mathbb{U}(\omega)$. Let $\mathcal{C}_i$ be i.i.d. instantiations of $\mathcal{C}$ for all $i \in [n]$. Since $\mathcal{C}$ is a sparsifier as in Line 5 of Protocol 4, we can use Theorem G.4 with $p = 1/\omega+1$ to finish the proof. $\qquad\square$

# H Useful Identities and Inequalities

For all $x, y, x_1, \ldots, x_m \in \mathbb{R}^d$, $s > 0$ and $\alpha \in (0, 1]$, we have:

$$\|x + y\|^2 \leq (1 + s) \|x\|^2 + (1 + s^{-1}) \|y\|^2, \tag{44}$$

$$\left\| \sum_{i=1}^m x_i \right\|^2 \leq m \left( \sum_{i=1}^m \|x_i\|^2 \right), \tag{45}$$

$$(1 - \alpha) \left( 1 + \frac{\alpha}{2} \right) \leq 1 - \frac{\alpha}{2}, \tag{46}$$

$$(1 - \alpha) \left( 1 + \frac{2}{\alpha} \right) \leq \frac{2}{\alpha}. \tag{47}$$

**Variance decomposition:** For any random vector $X \in \mathbb{R}^d$ and any non-random vector $c \in \mathbb{R}^d$, we have

$$\mathbb{E}\left[ \|X - c\|^2 \right] = \mathbb{E}\left[ \|X - \mathbb{E}[X]\|^2 \right] + \|\mathbb{E}[X] - c\|^2. \tag{48}$$

**Tower property:** For any random variables $X$ and $Y$, we have

$$\mathbb{E}[\mathbb{E}[X \mid Y]] = \mathbb{E}[X]. \tag{49}$$

**Jensen's inequality:** If $f$ is a convex function and $X$ is a random variable, then

$$\mathbb{E}[f(X)] \geq f(\mathbb{E}[X]). \tag{50}$$

**Lemma H.1** (Lemma 2 of Li et al. (2021)). *Suppose that function $f$ is L-smooth and let $x^{t+1} = x^t - \gamma g^t$. Then for any $g^t \in \mathbb{R}^d$ and $\gamma > 0$, we have*

$$f(x^{t+1}) \leq f(x^t) - \frac{\gamma}{2} \|\nabla f(x^t)\|^2 - \left( \frac{1}{2\gamma} - \frac{L}{2} \right) \|x^{t+1} - x^t\|^2 + \frac{\gamma}{2} \|g^t - \nabla f(x^t)\|^2.$$

**Lemma H.2** (Lemma 5 of Richtárik et al. (2021)). *Let $a, b > 0$. If $0 \leq \gamma \leq \frac{1}{\sqrt{a} + b}$, then $a\gamma^2 + b\gamma \leq 1$. Moreover, the bound is tight up to the factor of 2 since $\frac{1}{\sqrt{a} + b} \leq \min\left\{ \frac{1}{\sqrt{a}}, \frac{1}{b} \right\} \leq \frac{2}{\sqrt{a} + b}$.*

# I  Notation

| Algorithms | |
|---|---|
| $n$ | number of workers/nodes/clients/devices |
| $\gamma$ | stepsize |
| $\mathcal{C}_1^t, \ldots \mathcal{C}_n^t$ | Server-to-workers (primal) compressors |
| $\mathcal{Q}_1^t, \ldots \mathcal{Q}_n^t$ | Workers-to-server (dual) compressors |
| $\omega_P, \omega_D$ | Parameters of server-to-workers (primal) and workers-to-server (dual) compressors |
| $\theta$ | Correlated compressors parameter (Definition A.3) |
| $\beta$ | Momentum parameter (see Algorithm 2) |
| **Definitions** | |
| $\mathbb{U}(\omega)$ | The family of unbiased compressors with parameter $\omega$ (Definition 1.4) |
| $\mathbb{P}(\theta)$ | The family of correlated compressors with parameter $\theta$ (Definition A.3) |
| $L$ | Smoothness parameter of $f$ (Assumption 1.1 ) |
| $L_i$ | Smoothness parameter of $f_i$ (Assumption 1.5) |
| $L_A, L_B$ | Parameters from Assumption 4.2 |
| **Notation** | |
| $[k] = \{1, \ldots, k\}$ for any positive integer $k$ | |
| $\mathbb{E}_t\left[\cdot\right]$ - expectation conditioned on the first $t$ iterations | |
| $\delta^t := f(x^t) - f^*$ | |
| $\widehat{L}^2 := \frac{1}{n}\sum_{i=1}^n L_i^2$, $L_{\max} := \max_{i \in [n]} L_i$ | |
| $w^t := 1/n \sum_{i=1}^n w_i^t$ | |
| $g^t := 1/n \sum_{i=1}^n g_i^t$ | |
| $z^t := 1/n \sum_{i=1}^n z_i^t$ | |

Table 2: Frequently used notation.

