# OpenReview forum: "Improving the Worst-Case Bidirectional Communication Complexity for Nonconvex Distributed Optimization under Function Similarity"
_NeurIPS.cc/2024/Conference — NeurIPS 2024 spotlight_

### Official Review · Reviewer_K3Nf · 2024-06-19

**Soundness:** 3
**Presentation:** 4
**Contribution:** 3
**Rating:** 7
**Confidence:** 3

**Summary:**

Differently from other papers in the field of federated learning, the authors consider the problem of uplink compression - from the server to the workers. The main contributions of the paper are MARINA-P and its variants. These optimization schemes leverage PermK sparsification to reduce the communication complexity in the uplink. When combined with momentum, MARINA-P achieves a total communication complexity that, in the case of $n>>1$ and for close-to-homogeneous clients, is smaller than competing state-of-the-art alternatives.

**Strengths:**

The paper is original as it focuses on uplink compression, an aspect which is often overlooked in the federated learning community. The results are very interesting, as the authors not only show that under suitable conditions, MARINA-P has very low communication complexity, but they also demonstrate that without additional assumptions, there is an insurpassable lower bound to the communication complexity. The paper is very well-written; in fact, despite being very dense, it still introduces the problem and the proposed solution in a very clear manner.

**Weaknesses:**

- The main weakness of the paper is the assumption of data homogeneity. While I understand that this assumption is necessary, it would be interesting to include experimental evidence in the main text showing how MARINA-P performs with heterogeneous data.
- The following might be erroneous, but to me seems like that the overall number of coordinates sent in the uplink is $d$. From the discussion in Section 4.5 it seems like that the parameter vector is permuted and divided into non-overlapping chunks, and each chunk is sent to a different client. As such, it looks like that the \emph{overall} number of coordinates sent to the server is $d$.

**Questions:**

- Related to my previous point, how many coordinates are sent into the uplink at each round from the server to all users? i.e. what is the support of $\sum^n_{i=1} \mathcal{C}_i(x)$?
- Is MARINA-P compatible with client selection? Meaning, is it possible to apply MARINA-P if, at each round, the number of users varies or a different subset is chosen?
- Would it be possible to compare MARINA-P to other state-of-the-art schemes that use masking to reduce communication (e.g., SPARSE RANDOM NETWORKS FOR COMMUNICATION-EFFICIENT FEDERATED LEARNING)?"

**Limitations:**

Limitations are adequately discussed in the text.

---

> ### Author Rebuttal · Authors · 2024-08-04
>
> Thank you for your review and for acknowledging the contributions of our research. We will now address each of your comments and provide clarifications.
>
> __Weaknesses__
>
> > The main weakness of the paper is the assumption of data homogeneity. While I understand that this assumption is necessary, it would be interesting to include experimental evidence in the main text showing how MARINA-P performs with heterogeneous data.
>
> The assumption of data homogeneity is not necessary, and we do not assume it! While it is true that MARINA-P and M3 perform exceptionally well in the close-to-homogeneous regime, all our main results hold for arbitrarily heterogeneous clients. The only requirement is that the functional $(L_A,L_B)$ inequality holds. Importantly, for this to be true, the local functions do not need to be homogeneous. In fact, the functional $(L_A,L_B)$ inequality is weaker than the $L_i$-smoothness of the local functions $f_i$, which can clearly hold in heterogeneous scenarios.
>
> > The following might be erroneous, but to me seems like that the overall number of coordinates sent in the uplink is $d$. From the discussion in subsection 4.5 it seems like that the parameter vector is permuted and divided into non-overlapping chunks, and each chunk is sent to a different client. As such, it looks like that the _overall_ number of coordinates sent to the server is $d$.
>
> Yes, it is true that each client receives $\approx d/n$ coordinates, so the total number sent by the server is $d$ (as is the case for any sparsifier with the same sparsification level). Notice that without compression, the server would send $d$ coordinates to each worker, resulting in a total of $n \times d$ instead of $d$ coordinates sent.
>
> __Questions__
>
> > Related to my previous point, how many coordinates are sent into the uplink at each round from the server to all users? i.e. what is the support of $\sum \mathcal{C}_i(x)$?
>
> As mentioned previously, the server sends $K \approx d/n$ (distinct) coordinates to each client in the case of the permutation compressors, and $\sum \mathcal{C}_i(x) = x$.
>
> > Is MARINA-P compatible with client selection? Meaning, is it possible to apply MARINA-P if, at each round, the number of users varies or a different subset is chosen?
>
> Such behavior can be modeled using a specific compressor that satisfies Definition 1.4, and hence our theory applies. Specifically, let us define a compression operator $\mathcal{C}$ via $\mathcal{C}(x) = \mathcal{\bar{C}}(x)$ with probability $p$, and $0$ otherwise, where $\mathcal{\bar{C}} \in \mathbb{U}(\bar{\omega})$. It can easily be shown that $\mathcal{C} \in \mathbb{U}(p\bar{\omega}+1-p)$. In this model, the case when $\mathcal{C}(x) = 0$ can be interpreted as a worker not participating in this step.
>
> > Would it be possible to compare MARINA-P to other state-of-the-art schemes that use masking to reduce communication (e.g., SPARSE RANDOM NETWORKS FOR COMMUNICATION-EFFICIENT FEDERATED LEARNING)?"
>
> First, [5] focuses on training sparse networks, whereas our objective is the optimization of a general objective function. More importantly, the method in [5] does not address communication reduction in the downlink direction. The primary goal of the proposed FedPM algorithm is to train a probability mask to find the optimal sparse random network within the original random one, aiming to reduce the uplink communication cost. In contrast, this paper targets both downlink and uplink communication directions. MARINA-P specifically targets the downlink communication direction. Since these two algorithms address different goals, a direct comparison is not evident.
>
> [5] Isik, B., Pase, F., Gunduz, D., Weissman, T., & Zorzi, M. (2022). Sparse random networks for communication-efficient federated learning. arXiv preprint arXiv:2209.15328.
>
> ---
>
> We hope that the explanations provided adequately address the reviewer’s comments. Should any additional clarifications be required, we are happy to provide them.

---

> > ### Comment · Reviewer_K3Nf · 2024-08-10
> >
> > Thank you for your detailed response! Regarding the number of bits sent by the server to the workers: The channel from the server to the workers is broadcast, meaning that whatever the server transmits is received by all workers. Therefore, whether the server sends the same vector to all devices (as normally done) or sending different chunks of the vector to different device (as in MARINA-P), the number of bits and communication cost remain the same. Could you clarify under which communication model MARINA-P is more efficient?

---

> > > ### Author Response · Authors · 2024-08-10
> > >
> > > Thank you for the question.
> > >
> > > We agree that the number of bits that the server produces on the way out is the same in both cases. However, the number of bits that the server sends to *one worker* will be different. For MARINA-P with PermK, it is $d / n$ instead of $d.$ In Definition 1.3, we say "The server-to-worker (s2w) communication complexity of a method is the expected number of coordinates/floats the server sends to **a worker** to find an $\varepsilon$–solution." This definition is consistent with the fact that the downlink communication issue stems from the clients' download speed. Hence, the critical factor is the size of the message each client receives.
> > >
> > > To clarify further, imagine a tree (graph) with a central node (server) and $n$ child nodes (clients). Each worker is connected to the server by a unique edge. In this setup, the server-to-worker (s2w) communication complexity of a method is measured by the expected number of coordinates/floats the main node transmits through each edge to each of the connected workers.
> > >
> > > We hope this answers your question. If you have any more, please do not hesitate to ask.

---

### Official Review · Reviewer_r7n3 · 2024-07-14

**Soundness:** 3
**Presentation:** 3
**Contribution:** 3
**Rating:** 5
**Confidence:** 4

**Summary:**

This paper presents methods to improve communication efficiency in distributed optimization, particularly focusing on the server-to-worker (s2w) communication costs, which are often overlooked. The authors first present a lower bound on the required round of communication.  They used it to state that under no assumption between the individual functions stored at the distributed workers, the s2w communication loads can not be improved compared to the naive design. Then, the authors provide an algorithm and analysis showing that when the hessian of the individual functions are similar or aligned, improvements in communication complexity can be achieved.

**Strengths:**

The paper introduces a formal quantification, called functional $(L_A, L_B)$ inequality, to measure the similarity in the smoothness of the individual functions available to the workers. Novel algorithms and analysis are provided to show that the similarity can be used to improve the s2w communication load.

The paper also includes an interesting theoretical analysis, providing lower bounds on the required rounds of communication in the worst case.

**Weaknesses:**

1. The authors use the lower bound on the rounds communication to claim a lower bound on the communication load. However, there is a gap in this argument: The authors state that since the rounds of communication are lower bounded by a certain function, the overall communication load is lower bounded by d times that function, given that the full gradient contains d entries. But if a compressor is used as assumed by the authors, where the server can deliver an unbiased function that has fewer entries, it is totally possible that the communication load is smaller than d times the rounds of communication, violating this claimed bound.

Therefore, I have to suggest a weak rejection. I'm happy to raise my score if this issue can be appropriately fixed.

2. The concept of functional $(L_A, L_B)$ inequality can be simplified since only L_A is used in the main results. It can be stated such as the functions satisfy an $L_A$ condition, if we can find $L_B$ such that ineq. (9) is satisfied.

**Questions:**

Is there any connection between $L_A$ and the Hessians of the individual functions, similar to the smoothness parameter $L$, which can be interpreted as the largest eigenvalue of the Hessian of f?

**Limitations:**

See weekness

---

> ### Author Rebuttal · Authors · 2024-08-04
>
> Thank you for your review and appreciating the strengths of our work. We would like to address your concerns and provide additional explanations.
>
> __Weaknesses__
>
> > The authors use the lower bound on the rounds communication to claim a lower bound on the communication load....
>
> As far as we understand, the reviewer is pointing to a possible gap in our first contribution from Section 3. We believe that there may have been a misunderstanding, and there are no gaps in our argument.
>
> First of all, we are careful with our claims and only provide the lower bound $\Omega\left( \frac{(\omega + 1) L \delta_0}{\varepsilon} \right)$ for the number of **iterations/rounds.** We have never provided _a lower bound on the communication load_, and never claimed that _the overall communication load is lower bounded by $d$ times that function, given that the full gradient contains $d$ entries_. Once again, we only proved the lower bound $\Omega\left( \frac{(\omega + 1) L \delta_0}{\varepsilon} \right)$ for the number of iterations/rounds, where $\omega$ is the variance from Definition 1.4.
>
> We are on the same page with the reviewer that in order to determine the communication complexity, we have to multiply the number of iterations/rounds by the number of entries/coordinates of a compressor. For simplicity, let us consider the Rand$K$ compressor, which sends $K$ random coordinates and has variance $\omega = \frac{d}{K} - 1.$ Assuming that the server uses Rand$K$ in every iteration/round, the lower bound on the communication complexity is
> $$K \times \Omega\left( \frac{(\omega + 1) L \delta_0}{\varepsilon} \right) = K \times \Omega\left( \frac{d L \delta_0}{K \varepsilon} \right) = \Omega\left( \frac{d L \delta_0}{\varepsilon} \right),$$
> because, as the reviewer noted, we have to multiply by $K$ (not $d,$ as both we and the paper agree upon). Hence, we get the communication complexity $\Omega\left( \frac{d L \delta_0}{\varepsilon} \right)$ of GD. This argument leaves us no hope to improve the $\Omega\left( \frac{d L \delta_0}{\varepsilon} \right)$ communication complexity without an extra assumption.
>
> It is worth noting that we do not prove a lower bound for the communication complexity $\Omega\left( \frac{d L \delta_0}{\varepsilon} \right)$ in general. Instead, we establish a lower bound $\Omega\left( \frac{(\omega + 1) L \delta_0}{\varepsilon} \right)$ for the number of iterations/rounds, and then, assuming that the server uses, for instance, Rand$K$ in every iteration, we get the $\Omega\left( \frac{d L \delta_0}{\varepsilon} \right)$ communication complexity. Even this result provides strong evidence that we need an extra assumption to get further theoretical improvement.
>
> We believe that this clarification addresses the reviewer's concerns and explains that there are no mathematical gaps in our argument. We hope that the reviewer will reconsider the score. In case of any additional questions or if further clarification is needed, please let us know.
>
> > The concept of functional ($L_A, L_B$) inequality can be simplified since only $L_A$ is used in the main results. It can be stated such as the functions satisfy an $L_A$ condition, if we can find $L_B$ such that ineq. (9) is satisfied.
>
> It is true that $L_B$ does not appear in the result from Theorem 4.6 and Corollary 4.7. This is a particular strength of MARINA-P with permutation compressors.
>
> Let us look at the general results in Theorem D.1. (plus Theorem D.2 and Corollary D.4). The upper bound on the stepsize depends on $L_B^2\theta$, where $\theta$ is a compression parameter from Definition A.3. The reason why this term does not appear in Theorem 4.6 and Corollary 4.7 is that for permutation compressors, very conveniently one obtains $\theta=0$, which ultimately translates into superior complexity bound. However, the $L_B$ term is necessary for comparing this result with different compression schemes where $\theta>0$. Moreover, both $L_A$ and $L_B$ appear in the complexity bounds of M3 (Theorem 5.1), so their introduction in the main part of the paper is necessary.
>
> Overall, the functional $(L_A,L_B)$ inequality enables us to capture the intricate structure of the objective function and obtain superior complexity results for certain objective functions (those satisfying the assumption with small $L_A$). This would not be possible without introducing both the $L_A$ and $L_B$ terms. While the second term on the right-hand side of (9) (the $L_B$ term) can be bounded by the first term using Jensen's inequality, reallocating mass from $L_B$ to $L_A$ results in worse theoretical guarantees. Therefore, keeping $L_A$ as small as possible is beneficial, as smaller values yield better complexity results. Hence, the possibility that $L_B > 0$ is critical for obtaining the enhancements we present. On the other hand, since in general $L_A\neq0$, requiring that the $(L_A,L_B)$ holds with $L_A=0$ would limit the applicability of our results.
>
> We hope this clarifies why both terms are needed. Please let us know if there are any further questions.
>
> __Questions__
>
> > Is there any connection between  and the Hessians of the individual functions, similar to the smoothness parameter $L$, which can be interpreted as the largest eigenvalue of the Hessian of $f$?
>
> Yes, there is such a connection! In fact, this result is already included in the paper. Please, see Theorem 4.8, where we derive $L_A$ and $L_B$ based on the Hessians of the functions $f_i$.
>
> ---
> We hope that the explanations provided adequately address the reviewer's comments, and we kindly ask for a reconsideration of the score. Please let us know if more details or clarifications are needed.

---

> > ### Comment · Reviewer_r7n3 · 2024-08-10
> >
> > I have read the reviewer's response. Let me first clarify that my comment on the gap is due to the statements in the contributions in Section 2 rather than Section 3. The first contribution states: "This result gives no hope for improving the communication complexity (2) in the worst case." but the worst case in equation (2) refers to the communication cost instead of rounds of communication, i.e., $d\delta^0L/\epsilon$ vs the $(\omega+1)\delta^0L/\epsilon$ in the manuscript. This difference is precisely where the gap happens and the only way to make sense of it is to have a lower bound on communication cost rather than the rounds of communication.
> >
> > Similarly, the second main contribution in Section 2 uses the $(\omega+1)\delta^0L/\epsilon$ lower bound to provide a motivation to introduce extra assumptions, but the third main contribution again shows an upper bound in communication complexity, so the same gap happens.
> >
> > As explained by the author, the attempt to bridge this gap is different but similar to the spirit of the example provided by the reviewer, which is to take some "worst cases" over the designs rather than the problem scenarios. I believe we can agree that this is not a convincing argument, as the author's response clarified this is not proof as well. However, the issue of the manuscript is that by reading Section 2, it hints at the opposite, as explained above. For clarity and readability, the reviewer recommends the authors revise related statements to ensure that all claims in the paper exactly reflect the technical contributions.

---

> > > ### Author Response · Authors · 2024-08-10
> > >
> > > First, we would like to thank the reviewer for a quick response!
> > >
> > > We still believe that there is no mathematical gap in our assertions, as the sentence _This result gives no hope for improving the communication complexity (2) in the worst case_ is not a strong mathematical statement. What we intended to say is that _This result **indicates** it is not possible to improve the communication complexity (2) in the worst case_, which means that while we cannot be certain the bound is impossible to break with *any technique*, it is definitely not possible with approaches like the Rand$K$ compressor due the lower bound theorem.
> > >
> > > Nevertheless, we agree that this can lead to misunderstandings. Therefore, we propose to rewrite the contributions section in the following way (changes are in **bold**):
> > >
> > > 1. We start by proving the impossibility of devising a method where the server communicates with the workers using unbiased compressors $\mathbb{U}(\omega)$ (or biased compressors from Section B) and achieves an iteration rate faster than $\Omega\left(\frac{(\omega + 1) L \delta^0}{\varepsilon}\right)$ (Theorem 3.1) under Assumptions 1.1, 1.2, and 1.6.
> > > ~~**This result gives no hope for improving the communication complexity (2) in the worst case**~~.
> > > **This result provides a lower bound for any method that applies such compressors to vectors sent from the server to the workers in every iteration. Moreover, we prove a more general iteration lower bound of
> > > $\Omega\left(\frac{(\omega + 1)L \delta^0}{\varepsilon}\right)$ for all methods where the server zeroes out a coordinate with probability $1 / (\omega + 1)$ (see Remark 3.2).**
> > > 2. In view of this results, it is clear that an extra assumption is needed to break the lower bound $\Omega\left(\frac{(\omega + 1) L \delta^0}{\varepsilon}\right).$ In response, we introduce a novel assumption termed "Functional $(L_A, L_B)$ Inequality" (see Assumption 4.2). We prove that this assumption is relatively weak and holds, for instance, under the local smoothness of the functions $f_i$ (see Assumption 1.5).
> > > 3. We develop a new method for downlink compression, MARINA-P, and show that, under our new assumption, **along with Assumptions 1.1, 1.2 and 1.6, it can achieve the iteration rate of**
> > > \begin{align*}
> > >     \textstyle O\left(\frac{\delta^0 L}{\varepsilon} + \frac{\delta^0 L_A (\omega + 1)}{\varepsilon} + \frac{\delta^0 L_B (\omega + 1)}{\varepsilon \sqrt{n}}\right).
> > > \end{align*}
> > > **(see Theorem D.1 with $p = 1 / (\omega + 1)$ + Lemma A.5)**.
> > > Notably, when $L_A$ is small and $n \gg 1,$ this **iteration** complexity is provably superior to **$\Theta(\frac{\delta^0 L (\omega + 1)}{\varepsilon})$** and the complexities of the previous compressed methods. In this context, $L_A$ serves as a measure of the similarity between the functions $f_i$, and can be bounded by the ``variance'' of the Hessians of the functions $f_i$ (see Theorem 4.8). Thus, MARINA-P is the first method whose **iteration** complexity can provably improve with the number of workers $n$.
> > > 4. **Moreover, MARINA-P can achieve the s2w communication complexity of**
> > > \begin{align*}
> > >     \textstyle O\left(\frac{d \delta^0 L}{n \varepsilon} + \frac{d \delta^0 L_A}{\varepsilon}\right).
> > > \end{align*}
> > > **When $L_A$ is small and $n \gg 1,$ this communication complexity is provably superior to (2) and the communication complexities of the previous compressed methods.**
> > > 5. Our theoretical improvements can be combined with techniques enhancing the w2s communication complexities. In particular, by combining MARINA-P with MARINA (Gorbunov et al., 2021) and adding the crucial momentum step, we develop a new method, M3, that guarantees a total communication complexity (s2w + w2s) of
> > > \begin{align*}
> > >     \textstyle O\left(\frac{d \delta^0 L_{\max}}{n^{1/3} \varepsilon} + \frac{d \delta^0 L_A}{\varepsilon}\right).
> > > \end{align*}
> > > When $n > 1$ and in the close-to-homogeneous regime, i.e., when $L_A$ is small, this **communication** complexity is better than (2) and the complexities of the previous bidirectionally compressed methods.
> > > 6. Our theoretical results are supported by numerical experiments (see Section F).
> > >
> > > Additionally, we will change Lines 122-123 to "Let us first investigate the possibility of improving the iteration complexity $O\left(\frac{(\omega + 1)\delta^0 L}{\varepsilon}\right)$ under Assumptions 1.1, 1.2, and 1.6."
> > >
> > > These extra 4-5 sentences fully resolve any potential misunderstandings and clarify our contributions. These are very minor adjustments that can be easily incorporated into the camera-ready version of the paper.
> > >
> > > We believe that we have resolved the concerns and will happily answer any additional questions.

---

> > > > ### Comment · Reviewer_r7n3 · 2024-08-14
> > > >
> > > > I have read the authors’s response and the proposed revision would address my earlier concern. I have raised my score for the manuscript.

---

### Official Review · Reviewer_vddK · 2024-07-23

**Soundness:** 4
**Presentation:** 4
**Contribution:** 3
**Rating:** 6
**Confidence:** 4

**Summary:**

This paper addresses optimizing server-to-worker communication in distributed optimization by introducing MARINA-P, a novel method using correlated compressors for downlink compression. The study identifies inefficiencies in current downlink compression approaches and shows that MARINA-P can reduce communication complexity as the number of workers increases. Additionally, MARINA-P serves as a foundation for bidirectional compression methods. The paper also presents M3, which combines MARINA-P with uplink compression and a momentum step, further improving total communication complexity. Both theoretical analyses and empirical experiments validate the efficiency of the proposed algorithms.

**Strengths:**

The paper is exceptionally well written, with technical results rigorously proved. One of the main novelties is the use of correlated compressors to achieve a server-to-worker communication complexity that improves with the number of workers.

**Weaknesses:**

The compressor model (Definition 1.4), while standard in the federated learning literature, is somewhat contrived. It is primarily defined in its current form to facilitate analysis. It remains unclear whether the main conclusions derived under this model can be directly translated to the information-theoretic setting (quantization plus entropy coding), where compression efficiency is measured in terms of bits per sample.

**Questions:**

The role of L_B is not completely clear, as it does not prominently affect communication complexities compared to L_A. Is it that by isolating L_B from L_A, one can effectively reduce L_A  and, consequently, the communication complexities as well? Moreover, the class of functions for which L_B is known to be positive is quite restricted (Theorem 4.5). It would be beneficial to find more general examples.

**Limitations:**

The paper does not explicitly address its limitations. The only related point is that the proposed method does not consistently outperform the CORE method by Yue et al. (2023) in terms of performance (p. 8).

---

> ### Author Rebuttal · Authors · 2024-08-04
>
> We appreciate your comments and are grateful for highlighting the positive aspects of our work. We will now proceed to address the concerns you raised and provide clarifications.
>
> __Weaknesses__
>
> > The compressor model (Definition 1.4), while standard in the federated learning literature, is somewhat contrived. It is primarily defined in its current form to facilitate analysis.
>
> As in all mathematical theories, we need some assumptions about compressors to prove theorems and develop theory. Definition 1.4 is particularly useful because it not only facilitates analysis but also captures a wide variety of compressors.
>
> > It remains unclear whether the main conclusions derived under this model can be directly translated to the information-theoretic setting (quantization plus entropy coding), where compression efficiency is measured in terms of bits per sample.
>
> In Tables 1 and 2, we compare the compression efficiencies of the algorithms based on the total number of coordinates sent to find a $\varepsilon$-stationary point. The formulas we compare there do not involve parameters and notations from Definition 1.4. Following the convention in virtually all papers in computer science published at NeurIPS, we ignore the fact that all computations use float32/64 instead of real numbers. With this simplification, sending one coordinate is equivalent to sending 32/64 bits. We acknowledge that addressing the errors arising from floating-point arithmetic is an important problem.
>
> __Questions__
>
> > The role of $L_B$ is not completely clear, as it does not prominently affect communication complexities compared to $L_A$. Is it that by isolating $L_B$ from $L_A$, one can effectively reduce $L_A$ and, consequently, the communication complexities as well? Moreover, the class of functions for which $L_B$ is known to be positive is quite restricted (Theorem 4.5). It would be beneficial to find more general examples.
>
> Thank you for your comment. We agree that a more detailed discussion of this point (currently in the appendix), would benefit the paper, and we are happy to include it in the revised version. Let us clarify the results.
>
> Theorem 4.5 serves as an illustration of a case when $L_A=0$. However, please note that we do have much more general results with $L_B>0$ - see, for example, Theorem 4.8 (where we derive the constants $L_A$ and $L_B$ in terms of Hessians of the local functions $f_i$).
>
> It is true that $L_B$ does not appear in the result from Theorem 4.6 and Corollary 4.7. However, this is not a disadvantage but rather a particular strength of MARINA-P with permutation compressors. To explain this, let us refer to a more general result in Theorem D.1. (along with Theorem D.2 and Corollary D.4). The upper bound on the stepsize depends on $L_B^2\theta$, where $\theta$ is a compression parameter from Definition A.3. The reason why this term does not appear in Theorem 4.6 and Corollary 4.7 is that for permutation compressors, we very conveniently obtain $\theta=0$, which ultimately translates into a superior complexity bound.
>
> The functional  $(L_A,L_B)$ inequality enables us to capture the intricate structure of the objective function and achieve superior complexity results for certain objective functions (particularly those satisfying the assumption with small $L_A$). This would not be possible without introducing both the $L_A$ and $L_B$ terms. While the second term on the right-hand side of (9) (the $L_B$ term) can be bounded by the first term using Jensen's inequality, reallocating mass from $L_B$ to $L_A$ results in worse theoretical guarantees. Therefore, keeping $L_A$ as small as possible is beneficial, as smaller values yield better complexity results, and hence the possibility that $L_B > 0$ is critical for obtaining the enhancements we present. Furthermore, in general $L_A\neq0$, so requiring that the $(L_A,L_B)$ inequality holds with $L_A=0$ would limit the applicability of our results.
>
> Lastly, both $L_A$ and $L_B$ appear in the complexity bounds of M3 (Theorem 5.1), making their introduction in the main part of the paper necessary.
>
> We hope this clarifies the topic. Please let us know if there are any further questions.
>
> ---
>
> We appreciate your constructive feedback and trust that our responses have satisfactorily addressed the questions. Should any additional clarifications be required, we are happy to provide them.

---

> > ### Comment · Reviewer_vddK · 2024-08-13
> >
> > I am happy with the authors's response and will keep my original score.

---

### Official Review · Reviewer_1G9G · 2024-07-30

**Soundness:** 3
**Presentation:** 3
**Contribution:** 4
**Rating:** 7
**Confidence:** 4

**Summary:**

This paper deals with the downlink (server-to-client) communication cost in federated learning (FL). The main motivation of the paper is to provide a theoretical analysis of the total communication cost with downlink compression. This is achieved considering unbiased or a certain class of biased lossy compression algorithms at the server. First, the authors present an impossibility result (i.e., a converse theorem), showing that faster than certain rate is not possible if the server employs an unbiased compressor (or biased compressor as specified in the paper). Then, they proposed a downlink compression method called MARINA-P, which achieves better iteration than those in the literature. This method, basically, transmits an uncompressed model with a small probability p to all the clients, while the compressed model update is transmitted otherwise. The main twist in the proposed scheme is to transmit different compressed versions to the clients, rather than sending the same compression. In particular, an orthogonal sparse subset of the parameters are sent to each client, so that they together can recover all the parameters. This way, each parameter is considered at least by one of the workers.  Here, the compressed versions transmitted to the users become correlated. The authors show that this correlated compression approach achieves the best convergence rate. Finally, they extend this framework by combining downlink compression with uplink compression using MARINA together with a momentum step.

**Strengths:**

The paper looks at an important yet under explored aspect of communication in FL. The authors propose a novel server-to-client compression method, and provide upper and lower bounds on the convergence rate under various assumptions. Overall, the assumptions seem reasonable, and are shown to be satisfied by common compressors such as randK (and the introduced permK).

The proposed compressor is rather simple, but the observation that the best convergence rate is achieved by this particular correlated compression method is interesting.

**Weaknesses:**

I would say that the lack of any empirical results is the main weakness of the paper. While it is interesting to obtain the theoretical convergence guarantees, one also wonders how these three different compression schemes would perform in practice (for scenarios that satisfy the assumptions, as well as those that do not).

Also, only sparsification type compression methods are considered. What about quantization? Can the arguments be easily extended to the case with quantization?

**Questions:**

- While the authors have covered most of the relevant literature, as far as I'm aware, one missing paper that I am aware of is the following:

M. Mohammadi Amiri, D. Gündüz, S. R. Kulkarni, and H. V. Poor, "Federated Learning With Quantized Global Model Updates" arXiv preprint arXiv:2006.10672, Jun 2020.

Here, not only the authors consider downlink communication, but they actually transmit different updates to different clients. Instead of compressing the difference with rest to the previous global model, here the authors compress the difference with respect to the model updated by the client. This way the gap to the new global model is potentially lower. I can't say how the convergence rates compare, but I would like the authors to comment on the pros/cons of this approach compared to their own.

- How limiting is the assumption that the calls to the compressor are independent? There are some works in the literature that consider time-correlated sparsification, and show improved results (which also makes sense as the sparsity pattern should not change drastically from one iteration to the next). Please provide some comments in the paper if this may be a significant limitation on the practical performance of the considered class of compressors.

**Limitations:**

As mentioned above:
- lack of numerical comparisons,
- one missing reference

---

> ### Author Rebuttal · Authors · 2024-08-04
>
> Thank you for your feedback and for recognizing the strengths of our work. We will now address each of your comments in detail.
>
> __Weaknesses__
>
> > The lack of any empirical results is the main weakness of the paper.
>
> We thank the reviewer for the suggestion. We would like to highlight that our empirical results are presented in Section F.
>
> Our primary focus in this paper was on the theoretical aspects, rather than extensive experimental validation, which is why we placed the empirical section in the appendix. The experiments included are intended to demonstrate that our theoretical results closely align with practical observations, highlighting the robustness of our theoretical framework. In the camera-ready version, having an extra page, we are happy to include the highlights of experiments in the main part of the paper.
>
> > Only sparsification type compression methods are considered. What about quantization? Can the arguments be easily extended to the case with quantization?
>
> Yes, of course, our theory already covers quantization. Our general theoretical results (Theorems D.1, D.2, E.1) rely on Definition 1.4, which encompasses a wide range of compression operators. These operators include not only sparsification, but also, for example, natural compression and natural dithering [1], general exponential dithering (a generalization of ternary quantization), general unbiased rounding, and unbiased exponential rounding [2]. Various combinations of these compression methods are also covered. Naturally, our theory also applies to quantization. More information can be found in the references in Section 1.1 of the paper.
>
> In some theoretical results, we specialize in sparsification to highlight the superiority of permutation compressors. However, the aforementioned theorems demonstrate that our approach is applicable to a variety of other compression schemes as well (all compression techniques satisfying Definition 1.4).
>
> We agree that we can clarify this point further in the paper and will make the necessary revisions to ensure this.
>
> [1] Horvath, S., Ho, C. Y., Horvath, L., Sahu, A. N., Canini, M., & Richtárik, P. (2022, September). Natural compression for distributed deep learning. In Mathematical and Scientific Machine Learning (pp. 129-141). PMLR.
>
> [2] Beznosikov, A., Horváth, S., Richtárik, P., & Safaryan, M. (2023). On biased compression for distributed learning. Journal of Machine Learning Research, 24(276), 1-50.
>
> __Questions__
>
> > While the authors have covered most of the relevant literature, as far as I'm aware, one missing paper that I am aware of is the following:
> M. Mohammadi Amiri, D. Gündüz, S. R. Kulkarni, and H. V. Poor, "Federated Learning With Quantized Global Model Updates" arXiv preprint arXiv:2006.10672, Jun 2020.
> Here, not only the authors consider downlink communication, but they actually transmit different updates to different clients. Instead of compressing the difference with rest to the previous global model, here the authors compress the difference with respect to the model updated by the client. This way the gap to the new global model is potentially lower. I can't say how the convergence rates compare, but I would like the authors to comment on the pros/cons of this approach compared to their own.
>
> Thank you for pointing out this reference, we will add this work to the references. We would like to provide our comments on it.
>
> First, the statement that _they actually transmit different updates to different clients_ is incorrect. While it is true that the updates transmitted from the devices are different, the server sends the same message $Q(\theta(t) - \hat{\theta}(t-1), q_1)$ to all clients (line 2 of Algorithm 1).
>
> Regarding the theoretical comparison between the proposed LFL and our MARINA-P/M3, a crucial difference lies in the underlying assumptions. Most importantly, the results of [3] rely on the strong convexity of the local loss functions, whereas MARINA-P and M3 operate in the non-convex setting. Additionally, the theory in [3] assumes the boundedness of the stochastic gradients, which is a very restrictive assumption and, in fact, contradicts the strong convexity assumption (see, e.g., [4]). Therefore, the constant $G$ in Theorem 1 can be arbitrarily large, potentially making the upper bound infinite.
>
> We hope this clarifies the matter.
>
> [4] Nguyen, L., Nguyen, P. H., Dijk, M., Richtárik, P., Scheinberg, K., \& Takác, M. (2018, July). SGD and Hogwild! convergence without the bounded gradients assumption. In International Conference on Machine Learning (pp. 3750-3758). PMLR.
>
> > How limiting is the assumption that the calls to the compressor are independent? There are some works in the literature that consider time-correlated sparsification, and show improved results (which also makes sense as the sparsity pattern should not change drastically from one iteration to the next). Please provide some comments in the paper if this may be a significant limitation on the practical performance of the considered class of compressors.
>
> There exist various classes of compressors, each of them coming with some benefits and potentially some drawbacks. The assumption of time-independence is not a limitation. The class of compressors we consider is very wide, as a result of which the theoretical framework we developed covers a broad range of compression operators. Most importantly, the use of permutation compressors enables us to obtain superior complexity results, resulting in the first downlink compression algorithms whose complexities can provably improve with the number of workers.
>
> We are not aware of any work with theoretical evidence that _the sparsity pattern should not change drastically from one iteration to the next_. We would be grateful if the reviewer could provide such references.
> ___
> We are grateful for the feedback and believe that we have resolved the issues identified. If further information or explanations are required, please do not hesitate to ask.

---

> > ### Comment · Reviewer_1G9G · 2024-08-13
> >
> > I have read all the reviews and the authors' responses. I thank the authors for the detailed explanations. I believe it is a solid contribution on a relatively less explored aspect of distributed learning. The techniques are not necessarily new, but I think their application is rigorous and seems correct. I retain my original score.

---

### Decision · Program_Chairs · 2024-09-25

**Decision:**

Accept (spotlight)

**Comment:**

Unlike most papers in federated learning/distributed optimization, this paper studies the downlink communication cost (server to users). The paper presents a theoretical analysis on the required number communication rounds and shows that, under no assumption on the functions stored at the distributed workers, the communication cost cannot be improved with respect to a naive approach. But, if the hessian of the individual functions are "aligned", improvements in communication complexity can be achieved. In particular, they propose a downlink compression scheme called MARINA-P, which uses the interesting idea of sending correlated compression versions to the users. The reviewers agree that the paper provides a meaningful contribution in a relatively unexplored aspect of distributed optimization.